# Modeling Microenvironment Trajectories on Spatial Transcriptomics with NicheFlow

**Kristiyan Sakalyan**[*,1], **Alessandro Palma**[*,1,2,3], **Filippo Guerranti**[*,1,2,4],
**Fabian J. Theis**[1,2,3,4,5], **Stephan Günnemann**[1,2,4]

[1]School of Computation, Information and Technology, Technical University of Munich
[2]Munich Centre for Machine Learning (MCML)
[3]Institute of Computational Biology, Helmholtz Munich
[4]Munich Data Science Institute (MDSI), Technical University of Munich
[5]TUM School of Life Sciences Weihenstephan, Technical University of Munich
Correspondence to: {k.sakalyan, a.palma, f.guerranti}@tum.de

## Abstract

Understanding the evolution of cellular microenvironments in spatiotemporal data is essential for deciphering tissue development and disease progression. While experimental techniques like spatial transcriptomics now enable high-resolution mapping of tissue organization across space and time, current methods that model cellular evolution operate at the single-cell level, overlooking the coordinated development of cellular states in a tissue. We introduce NicheFlow, a flow-based generative model that infers the temporal trajectory of cellular microenvironments across sequential spatial slides. By representing local cell neighborhoods as point clouds, NicheFlow jointly models the evolution of cell states and spatial coordinates using optimal transport and Variational Flow Matching. Our approach successfully recovers both global spatial architecture and local microenvironment composition across diverse spatiotemporal datasets, from embryonic to brain development[1].

## 1 Introduction

Uncovering the principles governing tissue organization across space and time remains one of the most fundamental challenges in biology, with profound implications for evolutionary and developmental studies [1, 2]. While individual cells form the basic units of biological systems, they operate not in isolation but as integral parts of spatially organized *microenvironments*, functionally distinct neighborhoods, or *niches*, shaped by cell-to-cell interactions and extracellular components [3–5]. These spatial microenvironments influence crucial biological processes, from tumor progression to immune infiltration and tissue regeneration [6–8].

Spatial transcriptomics (ST) has transformed our ability to investigate these tissue architectures by providing single-cell resolution mapping of gene expression while preserving spatial context [9–12]. This technological breakthrough has enabled researchers to examine the molecular underpinnings of tissue organization with unprecedented detail. However, ST provides only static snapshots of inherently dynamic biological systems. *Time-resolved* spatial analysis extends ST by capturing how gene expression patterns and cellular arrangements evolve across developmental stages or experimental time [13–15]. This temporal dimension offers critical insights into the development of tissue organization in health and disease [16] that cannot be inferred from static observations alone.

---

[*]Equal contribution
[1]Project page: https://www.cs.cit.tum.de/daml/nicheflow

39th Conference on Neural Information Processing Systems (NeurIPS 2025).

Despite these technological advances, modeling trajectories on time-resolved spatial slides is complicated, as no direct correspondence exists between cells across slides due to the destructive nature of acquisition practices. Moreover, current computational methods fall short in modeling the evolution of tissue organization at the level of cellular microenvironments. Most approaches infer trajectories by modeling single-cell dynamics using velocity-based models [17–19] or optimal transport between individual cells [20, 21]. While effective at capturing cell evolution, these cell-centric methods fundamentally miss the coordinated evolution of structured niches within tissues.

This limitation presents a critical research gap that we address with the following question:

*How can we model the spatiotemporal evolution of cellular microenvironments preserving both local neighborhood relationships and cellular state transitions?*

To address this question, we directly model the dynamics of cellular neighborhoods as cohesive units rather than focusing on isolated cell trajectories. This approach aligns naturally with tissue-scale biological processes and enables principled learning of dynamics in structured, high-dimensional, and variably sized spatial domains.

We introduce **Niche Flow** Matching (**NicheFlow**) (Fig. 1), a generative model for learning spatiotemporal dynamics of cellular niches from time-resolved spatial transcriptomics data. NicheFlow builds on recent advances in Flow Matching (FM) and Optimal Transport (OT) to operate over distributions of microenvironments, which we represent as point clouds. NicheFlow enables accurate modeling of global spatial architecture and local microenvironment composition within evolving tissues.

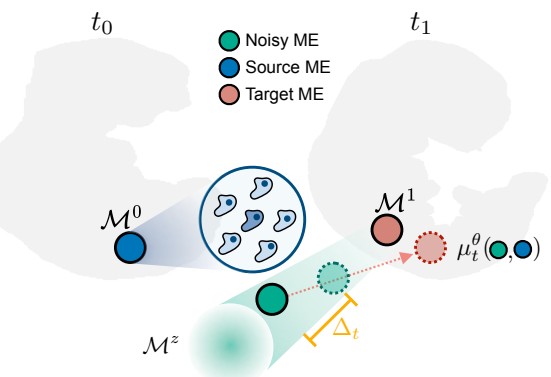

Figure 1: **Overview of NicheFlow.** At time $t_1$, we generate a target microenvironment $\mathcal{M}^1$ by transforming Gaussian noise $\mathcal{M}^z$ using a Variational Flow Matching model with a posterior $\mu_t^\theta$ conditioned on a source microenvironment $\mathcal{M}^0$ at $t_0$. Source-target pairs are identified via entropic OT over pooled microenvironment coordinates and gene expression profiles.

Our contributions include:

- **A microenvironment-centered trajectory inference paradigm** that shifts from modeling individual cells in time to modeling niches as point clouds, enabling simultaneous prediction of spatial coordinates and gene expression profiles while preserving local tissue context.

- **A factorized Variational Flow Matching (VFM) approach** with distributional families (Laplace for spatial coordinates, Gaussian for gene expression) that jointly trains on spatial and cell state dynamics using a factorized loss, modeling spatial reconstruction and biological fidelity with tailored distributional assumptions.

- **A spatially-aware sampling strategy** using OT between niche representations, enabling scalable training on large tissue sections while ensuring comprehensive coverage of heterogeneous regions.

Our approach consistently outperforms baselines in recovering cell-type organization and spatial structure across embryonic, brain development, and ageing datasets. NicheFlow enables principled learning of dynamics in structured, high-dimensional, and variably sized spatial domains, a challenge with parallels in other spatiotemporal modeling domains beyond biology.

## 2 Related Work

NicheFlow is at the interface between generative models and spatiotemporal transcriptomic data.

**FM and single-cell transcriptomics.** We propose a model based on FM, a framework introduced by several seminal works [22–24]. Specifically, we adopt a variational view of the FM objective, following Eijkelboom et al. [25], but extend it to mixed-factorized distributions for point cloud generation. Our method, NicheFlow, combines FM with OT, a pairing that has proven effective in modeling cellular data [26–29]. Unlike these models, however, we focus on point clouds of

spatially-resolved transcriptomic profiles. Closest to our approach is Wasserstein FM for point cloud generation [30], applied to reconstruct cellular niches. Yet, that work does not address joint generation of spatial coordinates and cellular states, nor OT-based temporal trajectory prediction, both of which are central to our contribution.

**Generative models for spatial transcriptomics.** Generative models have been key to spatial tasks such as gene expression prediction from histology slides [31, 32], integration with dissociated single-cell data [32], spatial imputation [33, 34], and perturbation [35]. More recently, LUNA [36] demonstrated strong performance in predicting single-cell spatial coordinates using diffusion models [37] conditioned on transcription data. While related, our model addresses the distinct task of inferring niche trajectories, enabling the simultaneous generation of coordinates and cellular states.

**Trajectory inference for spatial transcriptomics.** Previous work has explored learning trajectories from spatial slides. Pham et al. [38] proposed a graph-based spatiotemporal algorithm for pseudotime inference, while others leveraged tissue-resolved transcriptomics to estimate cell velocity [17–19]. Closer to our approach, Klein et al. [20] and Bryan et al. [21] use discrete OT to link cells across time and infer the evolution of cell states from spatially-resolved gene expression. Similarly, DeST-OT [39] aligns spatial slides with semi-relaxed OT couplings, preserving transcriptomic and spatial proximity between ancestor and descendant cells, while SpaTrack [19] uses Fused Gromov-Wasserstein OT [40], balancing transcriptomic and spatial differences based on spatial autocorrelation of features. Unlike our mini-batch deep learning model, these methods do not operate on entire microenvironments and rely on exact OT at the single-cell level, resulting in limitations in scalability and generalization.

## 3 Background

### 3.1 Optimal Transport with FM

FM [23] is a generative model that transforms a source density $p_0$ into a target density $p_1$. It operates by learning a time-dependent velocity field $u_t(\boldsymbol{x})$ for $t \in [0, 1]$ and $\boldsymbol{x} \in \mathbb{R}^D$, which generates a probability path $\{p_t\}_{t \in [0,1]}$. This path is constructed such that the marginals at time $t = 0$ and $t = 1$ match the source and target distributions, i.e., $p_0$ and $p_1$, respectively. The velocity field induces an Ordinary Differential Equation (ODE), whose solution $\phi_t(\boldsymbol{x})$ defines a *flow map* that transports samples from the source to the target distribution.

In practice, FM approximates $u_t(\boldsymbol{x})$ with a time-conditioned neural network $v_t^\theta(\boldsymbol{x})$. While the exact marginal velocity field $u_t(\boldsymbol{x})$ is intractable, it can be expressed in terms of data-conditioned velocity fields and a joint distribution $\pi(\boldsymbol{x}_0, \boldsymbol{x}_1)$ over the source and target samples $\boldsymbol{x}_0 \sim p_0$ and $\boldsymbol{x}_1 \sim p_1$:

$$u_t(\boldsymbol{x}) = \int u_t(\boldsymbol{x} \mid \boldsymbol{x}_0, \boldsymbol{x}_1) \frac{p_t(\boldsymbol{x} \mid \boldsymbol{x}_0, \boldsymbol{x}_1)\pi(\boldsymbol{x}_0, \boldsymbol{x}_1)}{p_t(\boldsymbol{x})} \, \mathrm{d}\boldsymbol{x}_0 \, \mathrm{d}\boldsymbol{x}_1 \,, \tag{1}$$

where $p_t(\boldsymbol{x} \mid \boldsymbol{x}_0, \boldsymbol{x}_1)$ is a pre-defined interpolating probability path. Here, we consider the tractable probability path $p_t(\boldsymbol{x} \mid \boldsymbol{x}_0, \boldsymbol{x}_1) = \delta(\boldsymbol{x} - g_t(\boldsymbol{x}_0, \boldsymbol{x}_1))$, where $g_t(\boldsymbol{x}_0, \boldsymbol{x}_1) = (1-t)\boldsymbol{x}_0 + t\boldsymbol{x}_1$ is a linear interpolation and $\delta$ denotes a Dirac delta function, representing a deterministic conditional path.

Lipman et al. [23] show that regressing the conditional field $u_t(\boldsymbol{x} \mid \boldsymbol{x}_0, \boldsymbol{x}_1)$ is equivalent to learning the marginal field $u_t(\boldsymbol{x})$ in expectation. Hence, the FM objective becomes the task of learning the velocity along the conditional probability path between any pair of source and target data points. For a linear conditional probability path, the velocity $u_t(\boldsymbol{x} \mid \boldsymbol{x}_0, \boldsymbol{x}_1)$ has a closed form, and the FM loss is:

$$\mathcal{L}_{\mathrm{FM}}(\theta) = \mathbb{E}_{t\sim\mathcal{U}[0,1],\, (\boldsymbol{x}_0,\boldsymbol{x}_1)\sim\pi} \left[ \left\| v_t^\theta\left(g_t(\boldsymbol{x}_0, \boldsymbol{x}_1)\right) - \frac{\partial}{\partial t} g_t(\boldsymbol{x}_0, \boldsymbol{x}_1) \right\|_2^2 \right]. \tag{2}$$

In practice, the coupling $\pi(\boldsymbol{x}_0, \boldsymbol{x}_1)$ is instantiated using sample pairs drawn from a mini-batch estimate. When one chooses $\pi^\star$ as the OT coupling under a squared Euclidean cost between samples from $p_0$ and $p_1$, FM approximates the dynamic OT map between source and target densities [41, 26]. Thus, the solution samples $(\boldsymbol{x}_0, \boldsymbol{x}_1) \sim \pi^\star$ from the joint distribution approximately follow:

$$\boldsymbol{x}_0 \sim p_0, \; \boldsymbol{x}_1 \sim \delta\left(\boldsymbol{x}_1 - \phi_1^\theta(\boldsymbol{x}_0)\right), \tag{3}$$

where $\phi_t^\theta$ is the solution of the ODE with velocity field $v_t^\theta$.

## 3.2 Generative OT on incomparable source and target spaces

Klein et al. [29] generalize the OT FM formulation to settings where the source and target distributions are defined on incomparable spaces and propose an approach to generative entropic OT using FM. Given a standard normal noise distribution with samples $z \sim \mathcal{N}(\mathbf{0}, \mathbf{I}_D)$, the authors show that the following sampling procedure:

$$x_0 \sim p_0, \; x_1 \sim \delta \left( x_1 - \phi_1^\theta(z \mid x_0) \right), \tag{4}$$

defines a generative model that implicitly samples from an Entropic OT (EOT) coupling, where $\phi_t^\theta(z \mid x_0)$ is a FM model that maps noise to target samples, conditioned on source points. To achieve this, $\phi_t^\theta$ is trained using source-target pairs $(x_0, x_1)$ drawn from the EOT coupling $\pi_\epsilon^\star$, with $\epsilon$ denoting the entropic regularization parameter, which the model aims to approximate.

Crucially, this formulation enables OT between distinct source and target spaces, *as $x_0$ does not flow directly into $x_1$, but instead conditions the generation of target samples from noise*.

## 3.3 Source-conditioned VFM

Consider the source-conditioned FM formulation in Sec. 3.2. Given a conditioning source $x_0$ and noise-based generation, the marginal field in Eq. (1) can be written as:

$$u_t(x \mid x_0) = \mathbb{E}_{p_t(x_1 \mid x, x_0)} \left[ u_t(x \mid x_1) \right] , \tag{5}$$

where we drop the conditioning on $x_0$ in the velocity field, as $u_t$ is entirely determined by the target $x_1$ when generating from noise under linear probability paths.

Since $u_t(x \mid x_1)$ is tractable [23], one can recast FM as a variational inference problem, following Eijkelboom et al. [25], by introducing a parameterized approximation $q_t^\theta(x_1 \mid x, x_0)$ to the true posterior $p_t(x_1 \mid x, x_0)$. Integrating the expected velocity in Eq. (5) over $t \in [0, 1]$ enables the generation of target points $x_1$ from noise, conditioned on $x_0$.

During training, the source-conditioned Variational Flow Matching (VFM) loss is:

$$\mathcal{L}_{\text{SC-VFM}}(\theta) = -\mathbb{E}_{t \sim \mathcal{U}[0,1], \, (x_0, x_1) \sim \pi_\epsilon^\star, \, x \sim p_t(x \mid x_1)} \left[ \log q_t^\theta(x_1 \mid x, x_0) \right] , \tag{6}$$

where $\pi_\epsilon^\star$ is an entropic OT coupling modeling the joint distribution over source and target samples, and $p_t(x \mid x_1)$ interpolates between target samples and noise. In the generation phase, one samples $x_0 \sim p_0$ and noise $z \sim \mathcal{N}(\mathbf{0}, \mathbf{I}_D)$, then simulates the marginal field in Eq. (5) starting from $\phi_0(x) = z$ to generate a target sample $x_1$.

Crucially, under the assumption that $u_t(x \mid x_1)$ is linear in $x_1$, which holds when using straight-line interpolation paths, the marginal field in Eq. (5) only depends on the posterior's first moment on $x_1$:

$$u_t(x \mid x_0) = u_t \left( x \mid \mathbb{E}_{p_t(x_1 \mid x, x_0)} [x_1] \right) \tag{7}$$

This implies that the VFM objective reduces to matching the first moment of the approximate posterior $q_t^\theta(x_1 \mid x, x_0)$ to that of the true posterior $p_t(x_1 \mid x, x_0)$. As a result, the approximate posterior can be chosen fully factorized under a *mean field assumption*, since each dimension can be matched independently if the mean of the true posterior is preserved; see App. C.2 for more details.

# 4 NicheFlow

We introduce a flow-based generative OT model to infer the temporal evolution of spatially resolved cellular microenvironments. More specifically, given a spatial microenvironment represented as a point cloud of cell states with their coordinates, NicheFlow predicts the corresponding tissue structure at a later time point. To delineate our approach, we define a list of desiderata.

**Generative model on structured data.** Similar to prior work [30, 36], we consider a generative model over structured point cloud data representing cellular microenvironments. This approach implicitly accounts for spatial correlations between cells, in contrast to models that study the evolution of spatial trajectories at the single-cell level [20].

**Sub-regions and variable location.** Crucially, for better memory efficiency, we do not consider an entire spatial slide for trajectory inference, but instead learn the dynamics of variably located

sub-regions. This design choice enables scalability and flexibility in modeling functional regions across different parts of the screened tissue.

**Changes in the number of nodes.** To model the temporal evolution and densification of microenvironments, we allow the source and target regions to differ in the number of nodes. We adopt a formulation similar to Sec. 3.2, implementing OT FM between non-comparable spaces.

**Flexible generative models for features and coordinates.** We allow flexibility in the choice of generative models for the features and coordinates. To this end, we implement the approach described in Sec. 3.3, factorizing features and coordinates into separate posteriors from different families.

## 4.1 Data description and problem statement

We are given a sequence of time-resolved spatial transcriptomic measurements across biological processes such as development or ageing. For simplicity, we formulate our problem in terms of two consecutive time points indexed by $s$, such that $s \in \{0, 1\}$, though the model can be extended to collections of more consecutive discrete temporal measurements. Each dataset at a time point is a full tissue slide that can be represented as an *attributed point cloud*:

$$\mathcal{P}_s = \{(\boldsymbol{c}_i^s, \boldsymbol{x}_i^s) \mid i = 1, \ldots, N_s\}, \tag{8}$$

where $\boldsymbol{c}_i^s \in \mathbb{R}^2$ denotes the 2D spatial coordinate of cell $i$ at time $s$, and $\boldsymbol{x}_i^s \in \mathbb{R}^D$ denotes its associated feature vector, typically corresponding to its gene expression profile or a low-dimensional representation thereof. Thus, each dataset is an *attributed point set* in two-dimensional space. Note that each slide can have a variable number of cells, and no direct correspondence exists between single cells across subsequent slides.

To capture spatial context beyond individual cells, we define local *microenvironments* as fixed-radius neighborhoods. Specifically, for each cell $(\boldsymbol{c}_i^s, \boldsymbol{x}_i^s)$ at time $s$, we construct a neighborhood $\mathcal{M}_i^s$ consisting of all neighboring cells within a spatial radius $r$:

$$\mathcal{M}_i^s = \left\{ (\boldsymbol{c}_j^s, \boldsymbol{x}_j^s) \, \middle| \, \|\boldsymbol{c}_j^s - \boldsymbol{c}_i^s\| \leq r \right\}. \tag{9}$$

Let $\{\mathcal{M}_i^0\}_{i=1}^{N_0}$ and $\{\mathcal{M}_j^1\}_{j=1}^{N_1}$ be collections of source and target microenvironments at consecutive time points. Our goal is to train a parameterized flow model $\phi_t^\theta$, with $t \in [0, 1]$, that generates target microenvironments conditioned on source point clouds. Specifically, to sample a target microenvironment with $k$ cells conditioned on a variably sized source $\mathcal{M}^0$, we define sampling as:

$$\mathcal{M}^z = \{(\boldsymbol{c}_i^z, \boldsymbol{z}_i) \mid \boldsymbol{c}_i^z \sim \mathcal{N}(\boldsymbol{0}, \mathbf{I}_2), \, \boldsymbol{z}_i \sim \mathcal{N}(\boldsymbol{0}, \mathbf{I}_D), \, \forall i = 1, \ldots, k\}, \tag{10}$$

$$\mathcal{M}^1 = \phi_1^\theta(\mathcal{M}^z \mid \mathcal{M}^0), \tag{11}$$

where $\mathcal{M}^z$ is a point cloud composed of noisy coordinates and features, and $\mathcal{M}^1$ is a generated prediction for the evolution of $\mathcal{M}^0$ at the next time point. As explained in Sec. 3.1 and Sec. 3.2, we want our generative model to parameterize some notion of optimal entropic coupling $\pi_\epsilon^\star$ between microenvironments across slides (see Sec. 4.2).

## 4.2 OT formulation

To train the flow map $\phi_t^\theta$ to perform conditional EOT, we define a cost function that induces an optimal entropic coupling $\pi_\epsilon^\star$ between source and target point clouds. This coupling is used to sample pairs of source and target microenvironment mini-batches during training. While there is no established notion of optimal cost in this setting, we propose to compute OT using source and target microenvironment representations based on the weighted average of features and coordinates.

Specifically, we compute a pooled representation for each microenvironment in the source and target slides by averaging spatial coordinates and gene expression features, weighted by a tunable hyperparameter $\lambda \in [0, 1]$ that balances spatial versus cellular state information:

$$\bar{\boldsymbol{m}}_i^s = \left[ \frac{1-\lambda}{|\mathcal{M}_i^s|} \sum_{(\boldsymbol{c}_j^s, \boldsymbol{x}_j^s) \in \mathcal{M}_i^s} \boldsymbol{c}_j^s \;\middle\|\; \frac{\lambda}{|\mathcal{M}_i^s|} \sum_{(\boldsymbol{c}_j^s, \boldsymbol{x}_j^s) \in \mathcal{M}_i^s} \boldsymbol{x}_j^s \right], \tag{12}$$

where $\|$ denotes concatenation, $\bar{\boldsymbol{m}}_i^s \in \mathbb{R}^{2+D}$, and $s \in \{0, 1\}$. We then apply EOT on the sets $\{\bar{\boldsymbol{m}}_i^0\}_{i=1}^{N_0}$ and $\{\bar{\boldsymbol{m}}_j^1\}_{j=1}^{N_1}$ using a squared Euclidean cost and regularization parameter $\epsilon$, yielding the

coupling $\pi^\star_{\epsilon,\lambda}$. During training, we sample matched pairs $(\mathcal{M}^0, \mathcal{M}^1) \sim \pi^\star_{\epsilon,\lambda}$ computed over mini-batches, and use them as supervision for learning the conditional generative model. A higher value of $\lambda$ prioritizes feature similarity, while lower values favor proximity in coordinates (see Fig. 12).

### 4.3 Mixed-factorized VFM

Once we have established a strategy for performing mini-batch OT, we proceed to describe our approach for learning the flow model $\phi^\theta_t$ and simulating Eq. (11). To this end, we adopt a variant of VFM (Sec. 3.3), originally developed for graph generation, and adapt it to our point cloud setting.

In line with Sec. 3.2 and 3.3, we delineate an objective to train a parameterized, source-conditioned posterior over target point clouds $q^\theta_t(\mathcal{M}^1 \mid \mathcal{M}, \mathcal{M}^0)$, where $\mathcal{M}^0$ and $\mathcal{M}^1$ represent source and target niches, and $\mathcal{M}$ denotes a noisy point cloud at interpolation time $t$. Importantly, our posterior comes with the following characteristics: **(i)** the posterior is factorized across the single points in a point cloud; **(ii)** the posterior is factorized across cellular features and coordinate dimensions; and **(iii)** the family of posteriors can be chosen differently between cellular features and coordinates.

Following **(i)**, we model the variational distribution over $\mathcal{M}^1$ by factorizing it across individual points $(\boldsymbol{c}_1, \boldsymbol{x}_1) \in \mathcal{M}^1$. Moreover, we tackle **(ii)** and **(iii)** using the mean-field VFM assumption, modeling cellular state and positions separately (see App. C.1 and C.2 for theoretical justifications):

$$q^\theta_t(\mathcal{M}^1 \mid \mathcal{M}, \mathcal{M}^0) = \prod_{(\boldsymbol{c}_1, \boldsymbol{x}_1) \in \mathcal{M}^1} \left( \prod_{k=1}^{2} f^\theta_t(c_1^k \mid \mathcal{M}, \mathcal{M}^0) \prod_{d=1}^{D} r^\theta_t(x_1^d \mid \mathcal{M}, \mathcal{M}^0) \right) . \quad (13)$$

Here, $f^\theta_t$ and $r^\theta_t$ denote distinct approximate posterior families for cellular states ($x_1^d$) and spatial positions ($c_1^k$), respectively. We use a Laplace distribution for $f^\theta_t$ due to its concentration around the mean, which supports precise modeling of coordinate features, while a Gaussian distribution is used for $r^\theta_t$.

As explained in Sec. 3.3, if one uses FM with straight probability paths, only the first moment of $q^\theta_t$ is required to simulate the generative field in Eq. (5). Therefore, the posterior is replaced by a time-dependent predictor $(\bar{\boldsymbol{f}}^\theta_t, \bar{\boldsymbol{r}}^\theta_t) = \mu^\theta_t(\mathcal{M}, \mathcal{M}^0)$ of the mean features $\bar{\boldsymbol{r}}^\theta_t$ and coordinates $\bar{\boldsymbol{f}}^\theta_t$ of the target point clouds, learned minimizing the following loss:

$$\mathcal{L}_{\text{NicheFlow}}(\theta) = \mathbb{E}_{\substack{t \sim \mathcal{U}[0,1] \\ (\mathcal{M}^0, \mathcal{M}^1) \sim \pi^\star_{\epsilon,\lambda} \\ \mathcal{M} \sim p_t(\mathcal{M} \mid \mathcal{M}^1)}} \left[ \sum_{(\boldsymbol{c}_1, \boldsymbol{x}_1) \in \mathcal{M}^1} \left( \|\boldsymbol{c}_1 - \bar{\boldsymbol{f}}^\theta_t\|_1 + \frac{1}{2}\|\boldsymbol{x}_1 - \bar{\boldsymbol{r}}^\theta_t\|_2^2 \right) \right] . \quad (14)$$

We derive the objective in App. C.4 and provide algorithms in App. E. Here, $\mu^\theta_t$ inputs a noisy point cloud $\mathcal{M}$ and a source $\mathcal{M}^0$ and provides a mean prediction vector for coordinates and feature dimensions, respectively indicated as $\bar{\boldsymbol{f}}^\theta_t$ and $\bar{\boldsymbol{r}}^\theta_t$. We implement it as a point cloud transformer (see Sec. 4.4).

We highlight a crucial aspect about Eq. (14). While the single dimensions are fully factorized in the predictions from $\mu^\theta_t$, every feature's mean is a function of the whole noisy point cloud $\mathcal{M}$ as well as the target microenvironment $\mathcal{M}^0$. In other words, the predictions exploit structural information in the point cloud to predict the individual posterior mean of each dimension. Like most approaches, our method has modeling limitations that we outline in App. B.

### 4.4 Backbone architecture: Microenvironment transformer

To parameterize the conditional posterior mean $\mu^\theta_t$ from Sec. 4.3, we use a *permutation-equivariant* transformer architecture designed for point clouds with variable size.

**Encoder–decoder structure.** The model follows an encoder-decoder layout, processing the source microenvironment $\mathcal{M}^0$ via the encoder and predicting the posterior mean from a noisy target $\mathcal{M} \sim p_t(\cdot \mid \mathcal{M}^1)$ via the decoder conditioned on the encoder's output.

**Input embeddings.** Each point is represented by features $\boldsymbol{x} \in \mathbb{R}^D$ and spatial coordinates $\boldsymbol{c} \in \mathbb{R}^2$, embedded separately and concatenated. Time $t$ is encoded using sinusoidal embeddings, linearly projected and broadcast across points.

**Cross-attention conditioning.** The encoder processes the embedded source $\mathcal{M}^0$ microenvironment using self-attention. The decoder operates on the noisy target $\mathcal{M}$ using self-attention, followed by

Table 1: Performance comparison across three biological datasets for SPFlow, RPCFlow and NicheFlow. Models are trained using the Conditional Flow Matching (CFM), Gaussian VFM (GVFM), or Gaussian-Laplacian VFM (GLVFM) strategies. Results are reported as mean ± standard deviation over five evaluation runs on mouse embryonic development (MED), axolotl brain development (ABD), and mouse brain aging (MBA). For all experiments, we use a fixed value of $\lambda = 0.1$, enabling spatial location preservation across time.

| | | MED | | | ABD | | | MBA | | |
|---|---|---|---|---|---|---|---|---|---|---|
| Model | Obj. | 1NN-F1 ↑ | PSD ↓ $(10^2)$ | SPD ↓ $(10^2)$ | 1NN-F1 ↑ | PSD ↓ $(10^2)$ | SPD ↓ $(10^2)$ | 1NN-F1 ↑ | PSD ↓ $(10^2)$ | SPD ↓ $(10^2)$ |
| LUNA | — | 0.540 ± 0.004 | — | — | 0.331 ± 0.003 | — | — | 0.222 ± 0.000 | — | — |
| SPFlow | CFM | 0.272 ± 0.0011 | 1.681 ± 0.0087 | 0.602 ± 0.0013 | 0.190 ± 0.0005 | 2.494 ± 0.0051 | 1.119 ± 0.0037 | 0.205 ± 0.0003 | 1.836 ± 0.0022 | 0.824 ± 0.0006 |
| SPFlow | GVFM | 0.259 ± 0.0009 | 2.383 ± 0.0082 | 0.582 ± 0.0009 | 0.175 ± 0.0010 | 3.373 ± 0.0103 | 1.104 ± 0.0023 | 0.181 ± 0.0001 | 2.585 ± 0.0029 | 0.834 ± 0.0011 |
| SPFlow | GLVFM | 0.251 ± 0.0008 | 2.249 ± 0.0114 | 0.592 ± 0.0015 | 0.173 ± 0.0013 | 2.870 ± 0.0238 | 1.093 ± 0.0037 | 0.195 ± 0.0005 | 2.320 ± 0.0009 | 0.853 ± 0.0008 |
| RPCFlow | CFM | 0.546 ± 0.0012 | 0.981 ± 0.0024 | 0.564 ± 0.0015 | 0.524 ± 0.0020 | **2.051 ± 0.0039** | 1.015 ± 0.0036 | 0.271 ± 0.0004 | **1.543 ± 0.0016** | 0.810 ± 0.0010 |
| RPCFlow | GVFM | 0.503 ± 0.0013 | 1.155 ± 0.0044 | 0.578 ± 0.0007 | 0.477 ± 0.0008 | 2.260 ± 0.0077 | 1.036 ± 0.0031 | 0.249 ± 0.0003 | 1.753 ± 0.0020 | 0.784 ± 0.0010 |
| RPCFlow | GLVFM | 0.586 ± 0.0016 | 0.979 ± 0.0021 | 0.586 ± 0.0012 | 0.554 ± 0.0007 | 2.053 ± 0.0044 | 1.038 ± 0.0025 | 0.265 ± 0.0004 | 1.723 ± 0.0015 | 0.779 ± 0.0011 |
| NicheFlow | CFM | 0.609 ± 0.0030 | 0.979 ± 0.0228 | 0.402 ± 0.0036 | 0.604 ± 0.0018 | 2.086 ± 0.0058 | **0.568 ± 0.0030** | 0.283 ± 0.0003 | 1.557 ± 0.0014 | 0.556 ± 0.0028 |
| NicheFlow | GVFM | 0.596 ± 0.0027 | 0.991 ± 0.0137 | 0.406 ± 0.0025 | 0.574 ± 0.0015 | 2.220 ± 0.0107 | 0.594 ± 0.0046 | 0.268 ± 0.0003 | 1.661 ± 0.0033 | **0.531 ± 0.0010** |
| NicheFlow | GLVFM | **0.664 ± 0.0014** | **0.883 ± 0.0094** | **0.398 ± 0.0023** | **0.628 ± 0.0013** | 2.079 ± 0.0043 | 0.576 ± 0.0055 | **0.285 ± 0.0003** | 1.554 ± 0.0021 | 0.532 ± 0.0009 |

cross-attention to condition on the encoder's outputs. The cross-attention mechanism allows each target point to attend to all source points.

**Output projection.** The decoder outputs are linearly projected to yield posterior mean estimates $(\bar{\boldsymbol{f}}_t^\theta, \bar{\boldsymbol{r}}_t^\theta)$ for expression and coordinates.

## 5 Experiments

We propose quantitative and qualitative evaluations of our algorithm. Quantitatively, we test whether source-conditioned samples generated by our model preserve the biological structure and shape of future tissue states. Qualitatively, we demonstrate that our approach accurately captures compositional shifts in substructural components and developmental trajectories across time.

### 5.1 Quantitative evaluation

Our first research question is to assess the impact of two core modeling choices in NicheFlow: **(i)** learning trajectories over spatial microenvironments, rather than independently for each cell and **(ii)** restricting source and target point clouds to spatially co-localized neighborhoods of cells, instead of sampling them randomly across the slide.

We use NicheFlow and baseline FM approaches that do not incorporate **(i)** and **(ii)** to simulate spatial trajectories conditioned on early time point observations. Assuming that spatial arrangements and the biological composition at later stages evolve from earlier slides, the global correspondence between predicted and true slides indicates the quality of the generative trajectory.

#### 5.1.1 Training setup

**Datasets.** We assess model performance across three spatiotemporal datasets: **(i)** Mouse embryo-genesis [20, 8] and **(ii)** the axolotl brain development [42], two Stereo-seq datasets profiling the spatially-resolved cellular development of a mouse embryo and axolotl brain across three (E9.5, E10.5 and E11.5) and five time points, respectively. We also consider the **(iii)** mouse brain ageing dataset [43], profiled with MERFISH [44] across twenty time points (see Apps. F.1 and F.2).

**Dataset construction.** For each dataset and time point $s \in \mathcal{S}$, we construct a set of cellular microenvironments by applying the fixed-radius neighborhood definition introduced in Sec. 4.1. Each microenvironment $\mathcal{M}_i^s$ is centered at cell $i$ and contains all cells within a fixed radius $r$. This results in:

$$\boldsymbol{\mathcal{M}}^s := \{\mathcal{M}_i^s \mid i = 1, \ldots, N_s\},$$

where $N_s$ is the number of cells in the tissue at time $s$, and each $\mathcal{M}_i^s$ is an attributed point cloud encoding both spatial and gene expression information. We standardize coordinates for cross-time comparability and reduce the normalized gene expression to its top 50 Principal Components (PC).

**Batching.** We train NicheFlow with mini-batches of source and target cellular point clouds. To ensure spatial diversity during training, we sample individual batches uniformly from within discrete

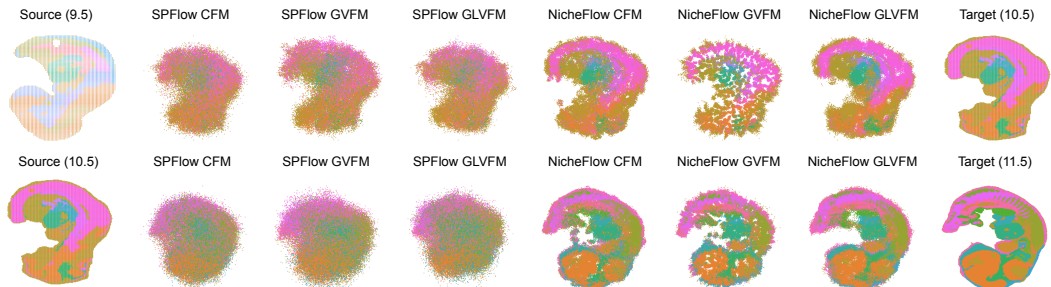

Figure 2: Qualitative comparison of generated samples on the embryonic development dataset (9.5-11.5 days). We show source and target samples alongside predictions from SPFlow and NicheFlow with different objectives.

regions of the slides computed with $K$-Means clustering over the 2D coordinates (see Fig. 13 for a visualization with different $K$ values). From these regions, we collect $M$ source and target microenvironments and resample $N \leq M$ matching pairs from the entropic OT coupling $(\mathcal{M}^0, \mathcal{M}^1) \sim \pi^\star_{\epsilon,\lambda}$ as described in Sec. 3.1, where $\mathcal{M}^0$ and $\mathcal{M}^1$ denote the sampled sets (see Sec. 4.2 for details on our OT coupling).

**Evaluation data.** For consistent and reproducible evaluation, we discretize each tissue into a fixed 2D grid and define evaluation microenvironments as fixed-radius neighborhoods around the nearest cells to each grid point. This guarantees full spatial coverage and ensures deterministic comparison across methods. See App. F.5 for details.

**Multiple time-point.** NicheFlow predicts piecewise trajectories between subsequent time points. Instead of learning one flow for each couple of subsequent slides, we train a single model with additional conditioning on source and target labels (see App. F.6).

### 5.1.2 Quantitative evaluation metrics

**Spatial structure.** We quantify coordinate generation accuracy using two asymmetric distance metrics. The *point-to-shape distance* (PSD) measures how far predicted coordinates deviate from the true structure by computing the mean squared distance from each generated point to its nearest ground truth counterpart. In contrast, the *shape-to-point distance* (SPD) evaluates how well the generated points cover the target region by averaging the squared distance from each ground truth point to its nearest generated point (see App. F.3 for a mathematical formulation of the metrics).

**Cell-type organization.** To assess how well the model reconstructs the spatial organization of different cell types, we use a *1-nearest-neighbor* (1NN) classification setup. Since the model generates only gene expression profiles and spatial coordinates, we assign cell type labels to generated cells using a classifier trained on ground truth gene expression data (see App. F.4). Each predicted cell is then matched to its nearest real cell, and we report the weighted F1 score (1NN-F1).

### 5.1.3 Models and results

**Baselines.** We compare against what we call *SPFlow* (Single-Point Flow), a standard FM-based model that predicts temporal trajectories across slides at a single-cell level using an MLP-based velocity field. We also consider *RPCFlow* (Random Point Cloud Flow), which has the same backbone as NicheFlow, but conditions on randomly sampled point clouds instead of radius neighborhoods. Additionally, we include *LUNA* [36], a diffusion model for spatial reconstruction from dissociated cells. Note that LUNA does not model temporal dynamics and only generates coordinates from noise with their respective biological annotations. Therefore, we use it as a reference for spatial generation accuracy via the 1NN-F1 metric rather than a proper baseline.

**Ablations.** We assess different training objectives by comparing standard Conditional Flow Matching (CFM) [26] with two variational formulations modeling posteriors over coordinates and features: Gaussian-only (GVFM) and Gaussian-Laplace (GLVFM). The former uses Gaussian posteriors for coordinates and features. The latter uses the factorized formulation in Sec. 4.3. For the point-cloud-based methods, we use a fixed value of $\lambda = 0.1$ and sampled batches of 64 regions chosen from

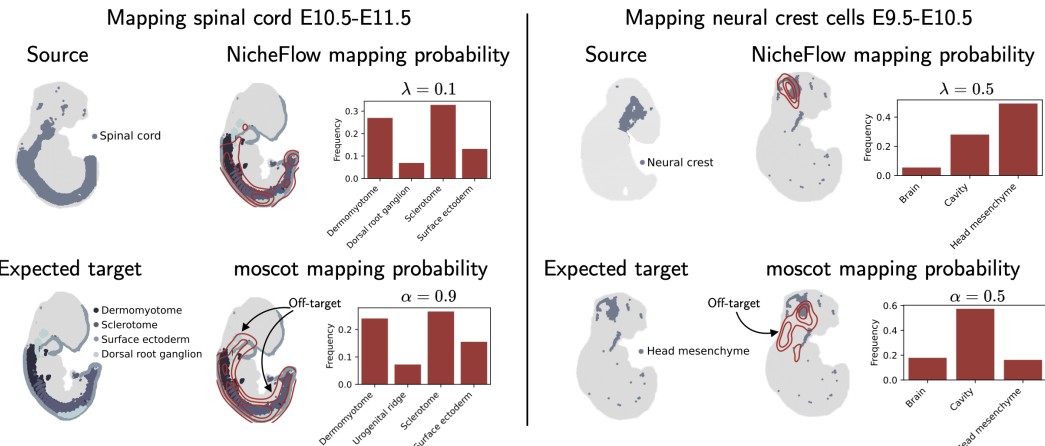

Figure 3: Left and right panels show the mapping of spinal cord (E10.5 to E11.5) and head neural crest cells (E9.5 to E10.5). In each panel, the left column shows source cells and expected targets, and the right column shows density contours of the most likely mapped regions. Bar plots display transition probabilities to the most likely descendant cell types. For NicheFlow, contours represent the proportion of samples in generated point clouds assigned to real cell coordinates across 10 samples.

the ablations in App. D.5 and D.6. We present results with a single value for $\lambda$ as this experiment compares different models on reconstructing fixed tissue structures over time, for which we enforce spatial preservation (see Sec. 4.2). However, we also include model comparisons across multiple values of $\lambda$ in Tab. 3 for completion.

**Results.** Our quantitative evaluation (Tab. 1) demonstrates that *NicheFlow* trained with the GLVFM objective consistently achieves strong performance across both spatial and semantic metrics. It outperforms all baselines in reconstructing spatial structure (PSD, SPD) and cell-type organization (1NN-F1) on developmental datasets, while remaining competitive on ageing data. These results highlight the importance of structured microenvironment modeling for capturing the spatiotemporal dynamics of complex tissues. We complement our quantitative results visually in Fig. 2 and Fig. 6 and 10 in the Appendix, where we show that SPFlow fails to capture tissue-level organization, producing blurry and spatially incoherent samples. In contrast, NicheFlow generates predictions that preserve spatial structure and cell-type organization, despite learning only from local microenvironments. In App. D.4 and App. D.7, we additionally demonstrate that NicheFlow produces conditionally consistent outputs with the source, while RPCFlow generates very diffused mappings across the slide, failing to preserve spatial consistency across time.

## 5.2    Qualitative evaluation and biological analysis

We explore the capabilities of NicheFlow on the spatial trajectory inference task through qualitative and biological assessments. Specifically, we focus on validating whether our model captures compositional changes within fixed spatial structures and developmental trajectories.

**Experimental setup.** We train the model as described in Sec. 5.1.1. Using the mouse embryonic dataset [8], we select specific microenvironments as source niches for which we want to study the trajectory over time. Specifically, we propose two scenarios for the application of NicheFlow depending on the choice of the OT parameter $\lambda$ (see Sec. 4.2):

1. *Compositional changes in fixed structures across time.* We choose the evolution of the *spinal cord* of the embryo from E10.5 to E11.5 as an example and set a low value of $\lambda$ to prioritize the preservation of the spatial location in the trajectory.
2. *Spatial and cellular development of immature cells.* Developing cells may displace to different areas of the embryo, requiring a higher value of $\lambda$ to account for gene expression. As a case study, we inspect how neural crest cells in the head evolve into mesenchymal and cranial structures.

**Baseline.** We compare NicheFlow with the spatiotemporal framework in moscot [20], which models spatial trajectories at the single-cell level. In contrast to NicheFlow, moscot integrates spatial

coordinates directly into the OT formulation using a Fused Gromov-Wasserstein cost [45], where the hyperparameter $\alpha$ controls the trade-off between spatial and feature-based distances (see App. F.8). A high $\alpha$ increases the influence of spatial distances, whereas in our framework this role is played by the hyperparameter $\lambda$ (see Sec. 4.2). We provide more details on the selection of the hyperparameter $\alpha$ for our experiments in App. D.2. Notably, moscot relies on exact OT and learns a transition matrix between source and target samples. As such, it is not a generative model over point clouds like NicheFlow. However, given the overlap in downstream tasks, we consider the comparison relevant.

**Evaluation and results.** For both scenarios (1) and (2), we have prior knowledge of the ground truth regions that the source microenvironments are expected to occupy at later developmental stages, as well as their corresponding biological compositions. For both moscot and NicheFlow, we assess whether the transported mass of source samples concentrates within the correct anatomical region at the target time point, and whether the predicted descendant cell types are biologically consistent (see App. F.8). Results are summarized in Fig. 3. When modeling the evolution of the spinal cord, moscot assigns considerable mass to unrelated regions such as the urogenital ridge and branchial arches, whereas NicheFlow correctly maps source niches to the maturing spinal cord. Similarly, NicheFlow captures the differentiation of neural crest cells into mesenchymal and cranial tissues within the head region, while moscot exhibits substantial off-target leakage towards lower regions.

## 6 Conclusions and Discussion

We introduce NicheFlow, a point-cloud-based generative model designed to capture the spatiotemporal dynamics of cellular niches in time-resolved spatial transcriptomics data. Unlike methods that model single-cell trajectories independently, NicheFlow implicitly captures spatial correlations between cells by learning trajectories on variably sized local neighborhoods. To this end, we combine OT with a new version of VFM that factorizes features and coordinates into distinct posteriors from different distribution families. We showed that NicheFlow outperforms standard FM approaches at reconstructing spatial context from previous time points and improves the mapping of biological structures in time over established exact OT approaches. With the expected increase in the volume and quality of spatial data, modeling coordinated cellular state translations with generative models is a promising avenue for generalization beyond spatiotemporal inference. We envision that models like NicheFlow will enable spatial perturbation prediction and modality translation tasks requiring principled parameterized maps beyond discrete OT to extrapolate and drive biological hypotheses.

## Acknowledgments

A.P. is supported by the Helmholtz Association under the joint research school Munich School for Data Science (MUDS). A.P., F.G., S.G. and F.J.T. also acknowledge support from the German Federal Ministry of Education and Research (BMFTR) through grant numbers 031L0289A and 031L0289C. F.J.T. acknowledges support from the Helmholtz Association's Initiative and Networking Fund via the CausalCellDynamics project (grant number Interlabs-0029) and the European Union (ERC, DeepCell, grant number 101054957). The authors of this work take full responsibility for its content.

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

# A  Broader impacts

This work addresses fundamental challenges in spatial transcriptomics by modeling complex spatial and compositional changes in developing tissues. We demonstrate how efficient representations of high-dimensional spatial cellular data can advance the understanding of developmental trajectories and microenvironment dynamics. We anticipate releasing NicheFlow as an open-source, user-friendly tool to enable broad application in spatial biology studies. Given its use with biological data, NicheFlow may also be applied in sensitive contexts involving clinical or patient information.

# B  Limitations

Our approach relies on fixed OT feature weighting by a parameter $\lambda$ during training (see Sec. 4.2), limiting flexibility at inference and potentially constraining certain biological analyses. Moreover, radius-based niche definitions may also be sub-optimal for small or irregularly shaped microenvironments, where the radius captures excessive spatial context and does not allow fine-grained modeling of the functional region's evolution.

The model assumes that spatial slides can be aligned with respect to each other in time and requires normalization-based pre-processing. Future work will be directed towards rotational and translational invariant spatial constraints and the incorporation of cell-to-cell communication priors in the neighborhood definition. While the learned flow models cell population dynamics, it does not explicitly capture biological events such as division or death.

Finally, this study focuses on in-distribution testing and does not consider generalization to unseen slides or full anatomical regions excluded from the training process. To achieve prediction in unseen settings, we foresee the need for technical replicates of the same slide across time points, as the model cannot extrapolate spatial arrangements without prior exposure to the associated region. We leave these analyses to future work when spatio-temporal measurements across multiple replicates become increasingly available.

# C  Mixed-factorized Variational Flow Matching

## C.1  Theoretical aspects of Variational Flow Matching

Variational Flow Matching (VFM) [25] relies on the observation that one can write the time-resolved marginal vector field $u_t(\boldsymbol{x})$ in FM as the expected conditional field $u_t(\boldsymbol{x} \mid \boldsymbol{x}_1)$ under the posterior $p_t(\boldsymbol{x}_1 \mid \boldsymbol{x})$ as:

$$u_t(\boldsymbol{x}) = \mathbb{E}_{p_t(\boldsymbol{x}_1 \mid \boldsymbol{x})} \left[ u_t(\boldsymbol{x} \mid \boldsymbol{x}_1) \right] . \tag{15}$$

Since $u_t(\boldsymbol{x} \mid \boldsymbol{x}_1)$ has a closed form and $u_t(\boldsymbol{x})$ is all that we need to generate the probability path $p_t$ from noise to data, this opens the door to a new interpretation of the objective as a variational inference problem, where we approximate $p_t(\boldsymbol{x}_1 \mid \boldsymbol{x})$ with a variational posterior $q_t^\theta(\boldsymbol{x}_1 \mid \boldsymbol{x})$. In other words, one can optimize the following objective:

$$\mathcal{L}_{\text{VFM}}(\theta) = -\mathbb{E}_{t \sim \mathcal{U}[0,1], \boldsymbol{x}_1 \sim p_1(\boldsymbol{x}_1), \boldsymbol{x} \sim p_t(\boldsymbol{x} \mid \boldsymbol{x}_1)} \left[ \log q_t^\theta(\boldsymbol{x}_1 \mid \boldsymbol{x}) \right] ,$$

where $p_1(\boldsymbol{x})$ is the data distribution and $p_t(\boldsymbol{x} \mid \boldsymbol{x}_1)$ a straight probability path. When $u_t(\boldsymbol{x} \mid \boldsymbol{x}_1)$ is linear in $\boldsymbol{x}_1$, this model formulation acquires convenient properties listed below.

**Mean parameterization.** The expected conditional field under the posterior only depends on the posterior mean:

$$\mathbb{E}_{p_t(\boldsymbol{x}_1 \mid \boldsymbol{x})} \left[ u_t(\boldsymbol{x} \mid \boldsymbol{x}_1) \right] = u_t(\boldsymbol{x}_1 \mid \mathbb{E}_{p_t(\boldsymbol{x}_1 \mid \boldsymbol{x})} \left[ \boldsymbol{x}_1 \right]) ,$$

suggesting that it is sufficient to parameterize the posterior mean to simulate data under the marginal flow. The posterior mean can be regressed against real samples $\boldsymbol{x}_1$ during training.

**Equivalence between posterior and approximate posterior formulation.** From the previous point, it follows that the expectation of the conditional field is the same under the true and approximate posterior, as long as their first moments match.

**Efficient simulation.** Given a parameterized posterior mean $\mu_t^\theta$, simulating the generative field in Eq. (15) is efficient under the linearity condition. For example, in the standard FM setting with

straight paths [23], the marginal generative field becomes:

$$u_t(\boldsymbol{x}) = \mathbb{E}_{q_t^\theta(\boldsymbol{x}_1 \mid \boldsymbol{x})} \left[ u_t(\boldsymbol{x} \mid \boldsymbol{x}_1) \right] \tag{16}$$

$$= u_t \left( \boldsymbol{x} \mid \mathbb{E}_{q_t^\theta(\boldsymbol{x}_1 \mid \boldsymbol{x})} \left[ \boldsymbol{x}_1 \right] \right) \tag{17}$$

$$= \frac{\mu_t^\theta(\boldsymbol{x}) - \boldsymbol{x}}{1 - t} . \tag{18}$$

which can be easily simulated in the range $t \in [0, 1]$.

## C.2 Factorized posterior

Similar to Eijkelboom et al. [25], in our work, we use a fully factorized posterior, where individual dimensions can follow different families of distributions with finite moments (see Sec. 4.3). Notably, a factorized approximate posterior over $\boldsymbol{x}_1$ is allowed as a choice for $q_t^\theta$, since the only requirement to simulate $u_t(\boldsymbol{x})$ is for $q_t^\theta(\boldsymbol{x}_1 \mid \boldsymbol{x})$ to match the expectation of $p_t(\boldsymbol{x}_1 \mid \boldsymbol{x})$ over $\boldsymbol{x}_1$, irrespectively of higher moments or correlations between factors.

In this regard, it is useful to consider the following proposition.

**Proposition 1.** *Let $\boldsymbol{x}_1 \in \mathbb{R}^D$ be a $D$-dimensional target data point, $p_t(\boldsymbol{x}_1 \mid \boldsymbol{x})$ the posterior probability path conditioned on a noisy point $\boldsymbol{x} \sim p_t(\boldsymbol{x})$, and $u_t(\boldsymbol{x} \mid \boldsymbol{x}_1)$ the conditional velocity field. Assume that $u_t(\boldsymbol{x} \mid \boldsymbol{x}_1)$ is linear in $\boldsymbol{x}_1$. Then, for any dimension $d \in \{1, \ldots, D\}$, the following holds:*

$$\mathbb{E}_{p_t(\boldsymbol{x}_1 \mid \boldsymbol{x})}[x_1^d] = \mathbb{E}_{p_t(x_1^d \mid \boldsymbol{x})}[x_1^d] \tag{19}$$

$$u_t(x^d) = u_t \left( x^d \mid \mathbb{E}_{p_t(x_1^d \mid \boldsymbol{x})}[x_1^d] \right) , \tag{20}$$

*where $x^d$ refers to the $d^{\text{th}}$ scalar dimension of the vector $\boldsymbol{x}$.*

*Proof.* We begin by proving Eq. (19) using marginalization:

$$\mathbb{E}_{p_t(\boldsymbol{x}_1 \mid \boldsymbol{x})}[x_1^d] = \int x_1^d \, p_t(\boldsymbol{x}_1 \mid \boldsymbol{x}) \, \mathrm{d}\boldsymbol{x}_1$$

$$= \int x_1^d \left( \int p_t(\boldsymbol{x}_1 \mid \boldsymbol{x}) \, \mathrm{d}\boldsymbol{x}_1^{\backslash d} \right) \mathrm{d}x_1^d$$

$$= \int x_1^d \, p_t(x_1^d \mid \boldsymbol{x}) \, \mathrm{d}x_1^d . \tag{21}$$

Next, we prove Eq. (20). Under the assumption that the conditional velocity field $u_t(\boldsymbol{x} \mid \boldsymbol{x}_1)$ is linear in $\boldsymbol{x}_1$, we have:

$$u_t(x^d) = \mathbb{E}_{p_t(\boldsymbol{x}_1 \mid \boldsymbol{x})} \left[ u_t(x^d \mid \boldsymbol{x}_1) \right]$$

$$\overset{(1)}{=} \mathbb{E}_{p_t(\boldsymbol{x}_1 \mid \boldsymbol{x})} \left[ u_t(x^d \mid x_1^d) \right]$$

$$= \mathbb{E}_{p_t(\boldsymbol{x}_1 \mid \boldsymbol{x})} \left[ \frac{x_1^d - x^d}{1 - t} \right]$$

$$= \frac{\mathbb{E}_{p_t(\boldsymbol{x}_1 \mid \boldsymbol{x})}[x_1^d] - x^d}{1 - t}$$

$$\overset{Eq. (21)}{=} \frac{\mathbb{E}_{p_t(x_1^d \mid \boldsymbol{x})}[x_1^d] - x^d}{1 - t}$$

$$= u_t \left( x^d \mid \mathbb{E}_{p_t(x_1^d \mid \boldsymbol{x})}[x_1^d] \right) . \tag{22}$$

Here, step (1) follows from the linearity assumption, which ensures that the conditional velocity at $x^d$ depends only on $x_1^d$.

In other words, the expected value under the posterior at an individual feature $d$ does not depend on the other features and has an influence only on the $d$-th dimension of the conditional vector field. This flexibility allows each dimension's approximate posterior to be chosen from a potentially different distributional family, as long as the first moment exists and is correctly parameterized.

## C.3 Marginal field derivation in source-conditioned VFM

When applying source conditioning to VFM, the marginal conditional vector field given a source $\boldsymbol{x}_0$ is:

$$u_t(\boldsymbol{x} \mid \boldsymbol{x}_0) = \int u_t(\boldsymbol{x} \mid \boldsymbol{x}_1) \frac{p_t(\boldsymbol{x} \mid \boldsymbol{x}_1) \, \pi(\boldsymbol{x}_1 \mid \boldsymbol{x}_0)}{p_t(\boldsymbol{x} \mid \boldsymbol{x}_0)} \, \mathrm{d}\boldsymbol{x}_1 \tag{23}$$

where the $p_t(\boldsymbol{x} \mid \boldsymbol{x}_1)$ is a probability path interpolating observations $\boldsymbol{x}_1$ with noise. Note that we omit $\boldsymbol{x}_0$ from the probability path and conditional velocity as they are fully determined by $\boldsymbol{x}_1$ under linear conditional probability paths. Furthermore, we can rewrite the marginal as an expectation:

$$\int u_t(\boldsymbol{x} \mid \boldsymbol{x}_1) \frac{p_t(\boldsymbol{x} \mid \boldsymbol{x}_1) \, \pi(\boldsymbol{x}_1 \mid \boldsymbol{x}_0)}{p_t(\boldsymbol{x} \mid \boldsymbol{x}_0)} \, \mathrm{d}\boldsymbol{x}_1 = \mathbb{E}_{p_t(\boldsymbol{x}_1 \mid \boldsymbol{x}, \boldsymbol{x}_0)} \left[ u_t(\boldsymbol{x} \mid \boldsymbol{x}_1) \right] \,, \tag{24}$$

where we used that $p_t(\boldsymbol{x} \mid \boldsymbol{x}_1) = p_t(\boldsymbol{x} \mid \boldsymbol{x}_0, \boldsymbol{x}_1)$.

## C.4 Gaussian and Laplacian Hybrid VFM

We define a hybrid Variational Flow Matching (VFM) model using a fully factorized variational distribution over individual points in the target microenvironment $\mathcal{M}^1$. Following the mean-field assumption, the variational distribution factorizes over spatial and feature dimensions:

$$q_t^\theta(\mathcal{M}^1 \mid \mathcal{M}, \mathcal{M}^0) = \prod_{(\boldsymbol{c}_1, \boldsymbol{x}_1) \in \mathcal{M}^1} q_t^\theta(\boldsymbol{c}_1, \boldsymbol{x}_1 \mid \mathcal{M}, \mathcal{M}^0) \tag{25}$$

$$= \prod_{(\boldsymbol{c}_1, \boldsymbol{x}_1) \in \mathcal{M}^1} \left( \prod_{k=1}^{2} f_t^\theta(c_1^k \mid \mathcal{M}, \mathcal{M}^0) \cdot \prod_{d=1}^{D} r_t^\theta(x_1^d \mid \mathcal{M}, \mathcal{M}^0) \right) . \tag{26}$$

In line with Eijkelboom et al. [25], using FM with straight paths enables us to efficiently simulate the marginal generating field using the *first moment* of the posterior distribution. In other words, for a fully factorized posterior, we only need to parameterize a mean predictor. In our setting, the mean prediction is a neural network $\mu_t^\theta$ as a time-condition function of a noisy microenvironment $\mathcal{M}$ and a source $\mathcal{M}^0$ with outputs:

$$(\bar{\boldsymbol{f}}_t^\theta, \bar{\boldsymbol{r}}_t^\theta) = \mu_t^\theta(\mathcal{M}, \mathcal{M}^0) \,,$$

where $\bar{f}_t^{\theta,k}$ and $\bar{r}_t^{\theta,d}$ are the expected values for the $k^{\text{th}}$ coordinate and $d^{\text{th}}$ cell feature. Then, we choose a parameterization for the variational factors at time $t \in [0,1]$ as follows:

$$x_1^d \sim \mathcal{N}(\bar{r}_t^{\theta,d}, 1), \tag{27}$$

$$c_1^k \sim \text{Laplace}(\bar{f}_t^{\theta,k}, 1), \tag{28}$$

Substituting into the negative log-likelihood yields:

$$-\log(q_t^\theta(\mathcal{M}^1 \mid \mathcal{M}, \mathcal{M}^0)) \tag{29}$$

$$= -\log \left( \prod_{(\boldsymbol{c}_1, \boldsymbol{x}_1) \in \mathcal{M}^1} \left( \prod_{k=1}^{2} f_t^\theta(c_1^k \mid \mathcal{M}, \mathcal{M}^0) \cdot \prod_{d=1}^{D} r_t^\theta(x_1^d \mid \mathcal{M}, \mathcal{M}^0) \right) \right) \tag{30}$$

$$= \sum_{(\boldsymbol{c}_1, \boldsymbol{x}_1) \in \mathcal{M}^1} \left( \sum_{k=1}^{2} \left( \log 2 + |c_1^k - \bar{f}_t^{\theta,k}| \right) + \sum_{d=1}^{D} \left( \frac{1}{2} \log(2\pi) + \frac{1}{2} \left( x_1^d - \bar{r}_t^{\theta,d} \right)^2 \right) \right)$$

$$= \sum_{(\boldsymbol{c}_1, \boldsymbol{x}_1) \in \mathcal{M}^1} \left( \|\boldsymbol{c}_1 - \bar{\boldsymbol{f}}_t^\theta\|_1 + \frac{1}{2} \|\boldsymbol{x}_1 - \bar{\boldsymbol{r}}_t^\theta\|_2^2 \right) + \text{const w.r.t. } \theta \tag{31}$$

This results in a loss consisting of an $\ell_1$ error on spatial coordinates and a mean squared error on gene expression features, consistent with the hybrid variational design.

# D   Additional results

## D.1   Additional comparisons with moscot on embryonic development

We propose a similar analysis as presented in Sec. 5.2.

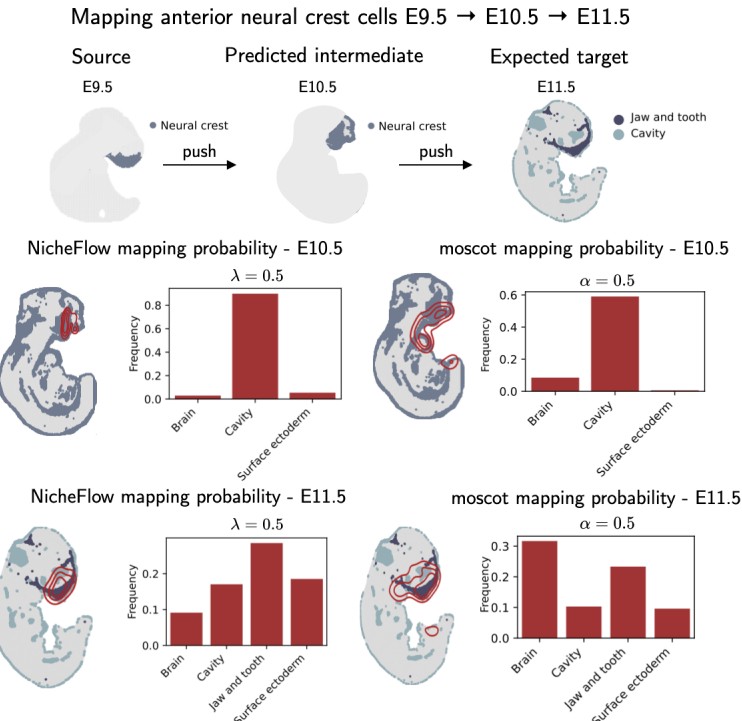

Figure 4: Comparison of NicheFlow and moscot on the prediction of the anterior neural crest cells' fate. For both models, we take source facial neural crest cells at E9.5, push them to time point E10.5, and show the compositional and density predictions in the middle panel. Then, the predictions at 10.5 are used as a source for a second trajectory prediction operation from 10.5 to 11.5, for which we inspect again the cell density over the target slide and the cell type probabilities.

In Fig. 4, we compare the ability of NicheFlow to predict an entire spatial structure trajectory by pushing an initial source cloud through all the developmental stages. To this end, the trajectory of an initial point cloud is first pushed to the next time point, and the model's prediction is used as a source for predicting the subsequent time point. An accurate niche trajectory reconstruction signifies that our model can be used to sequentially predict microenvironment evolution by treating its intermediate predictions as inputs, corroborating their accurate reflection of real point clouds.

In Fig. 4, we show that pushing anterior neural crest cells twice from E9.5 to E11.5 through the flow generates realistic target point clouds with a cell composition reflecting the expected cranial structure, mostly made of cavity cells, jaw and teeth (arising at E11.5 for the first time) and surface ectoderm. Doing the same with moscot oversamples regions outside of the cranial structure, thereby incorrectly mapping most of the neural crest density to brain cells.

In Fig. 5, we also show that NicheFlow is more accurate than moscot at transporting mass from defined organs like the liver across development. More specifically, while density leaks from the liver to the GI tract in the mapping produced by moscot, the prediction computed by NicheFlow more accurately retrieves the liver structure at the later time point. Together with previous evidence, our results underscore the importance of accounting for spatial correlations between cells during OT-based trajectory inference to buffer out the noise resulting from single-cell-based predictions.

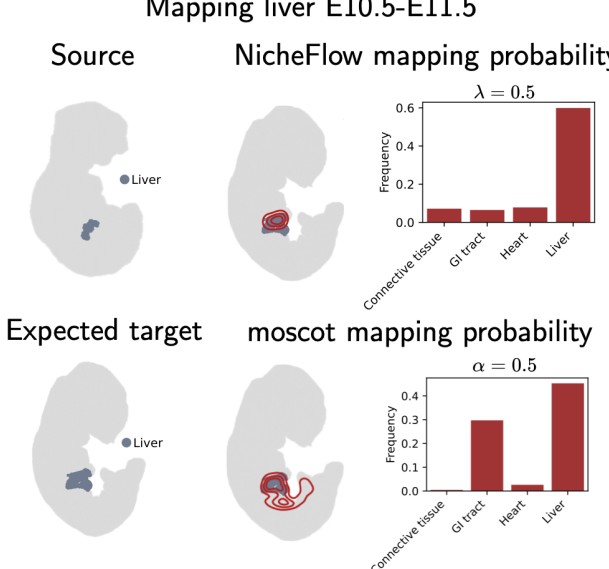

Figure 5: Comparison between moscot and NicheFlow on mapping the liver structure from E10.5 to E11.5. The liver at time E10.5 is used as a source for trajectory prediction using the different models. The left column shows the source and expected target regions highlighted on the respective E10.5 and E11.5 embryos. The middle column displays the density of the prediction obtained by transporting niches from the source to the target slide. On the right, the aggregated cell type proportions according to the density in the middle column (see App. F.8).

## D.2 $\alpha$ parameter sweep in moscot

The comparison with moscot assesses how well each model captures compositional changes in anatomical structures or migratory patterns, depending on the use case. Qualitatively, we found that the spatial component in the OT problem considered by moscot and regulated by the hyperparameter $\alpha$ is comparable to our spatial term $\alpha$ across the 0.1–0.9 range in the Fig. 3 analysis.

In Tab. 2, we report the proportion of source density mapped to the correct cell type across $\alpha$ values, to support our choice for the results in Sec. 5.2. For the spinal cord (Fig. 3, left), values above 0.75 yielded better qualitative and quantitative results. For neural crest cells (Fig. 3, right), $\alpha = 0.5$ performs best.

More specifically, when mapping fixed structures over time, values below 0.75 caused excessive density dispersion outside the anatomical region. For migration, no value led to generally accurate transitions, though $\alpha = 0.5$ mapped the highest density to the expected cell type.

Table 2: Effect of the parameter $\alpha$ balancing spatial and cell state preservation in moscot. The results in the table indicate the percentage of source density mapped to the correct cell type from the source anatomical structure (the higher, the better).

| Tissue | $\alpha = 0.1$ | $\alpha = 0.25$ | $\alpha = 0.5$ | $\alpha = 0.75$ | $\alpha = 0.9$ |
|---|---|---|---|---|---|
| Spinal Cord | 0.146 | 0.190 | 0.712 | 0.729 | 0.725 |
| Neural Crests | 0.159 | 0.160 | 0.162 | 0.155 | 0.112 |

## D.3 Additional experiments on the axolotl brain development and aging datasets

We provide additional visualizations of the generated samples on the axolotl brain development dataset, presented in Fig. 6. As can be seen from the figure, NicheFlow correctly retrieves the spatial and anatomical characteristics of the brain, including hemisphere formation and cavity.

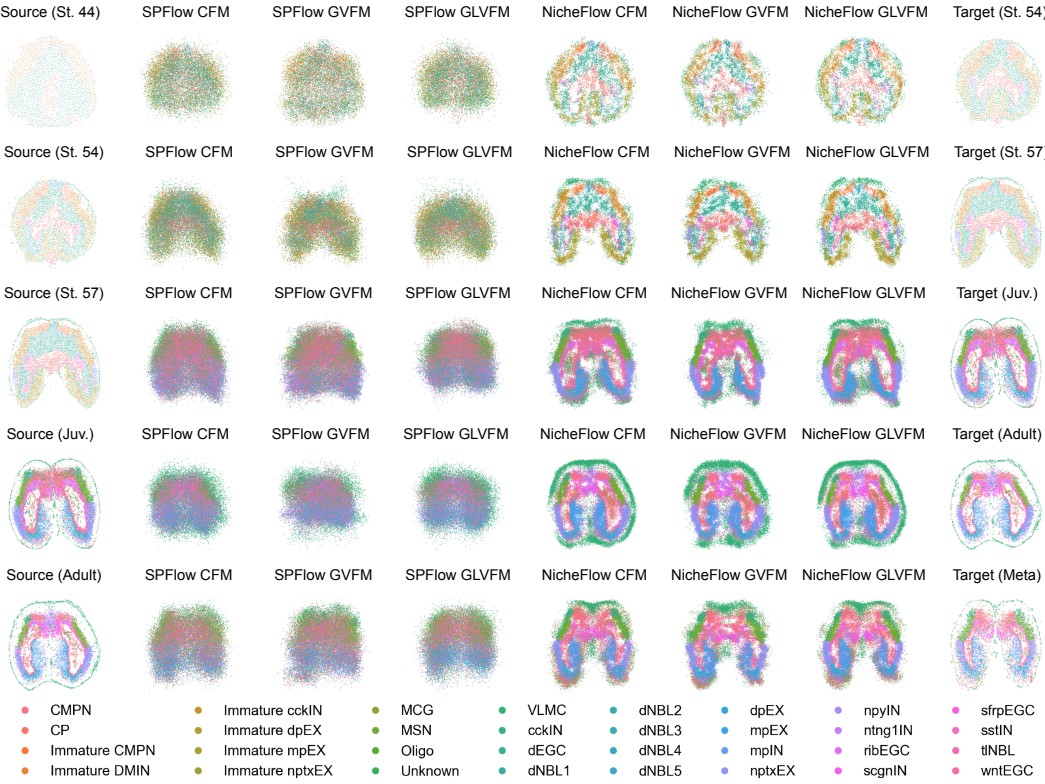

Figure 6: Qualitative comparison of generated samples on the axolotl brain development dataset (Stage 44, 54, 57, Juvenile, Adult, Meta). We show source and target samples alongside predictions from SPFlow and NicheFlow with different objectives. NicheFlow captures the spatial structure and cell-type organization more faithfully across developmental stages.

To further support our result, we propose a more in-depth qualitative analysis of the NicheFlow application on the axolotl brain development dataset. More specifically, in Fig. 7 we demonstrate that our model predicts the formation of crucial anatomical structures like the left and right lobes both spatially and compositionally. This vouches for flexibility in NichFlow's performance, which extends to non-trivial topology changes and simultaneously accounts for accurate cell state and coordinate generation in time. Similar results can be observed in Fig. 8, where we showcase the correct prediction of the formation of a left lobe cavity, predicting trajectories from an immature brain region.

Moreover, in Fig. 9 we predict the compositional and structural time evolution of the left dorsal pallium in the axolotl brain development. Following Wei et al. [42], we know that early time points populate the dorsal pallium of immature cell types like ependymoglial cells (EGC), neuroblasts (NBL), and immature neurons. In the left dorsal pallium, these disappear at the juvenile stages (the $3^{rd}$) and lead to differentiation into mature neurons (nptxEX) and later EGC (WntEGC, sfrpEGC). This fixed structural development was accurately predicted by NicheFlow when pushing left dorsal pallium cells forward across the trajectory.

Finally, similar to Fig. 2 and Fig. 6, in Fig. 10, we qualitatively show that NicheFlow with microenvironment sampling strategy and mixed-factorized VFM is the best approach for reconstructing mouse brain trajectories in time.

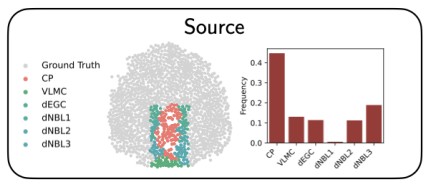

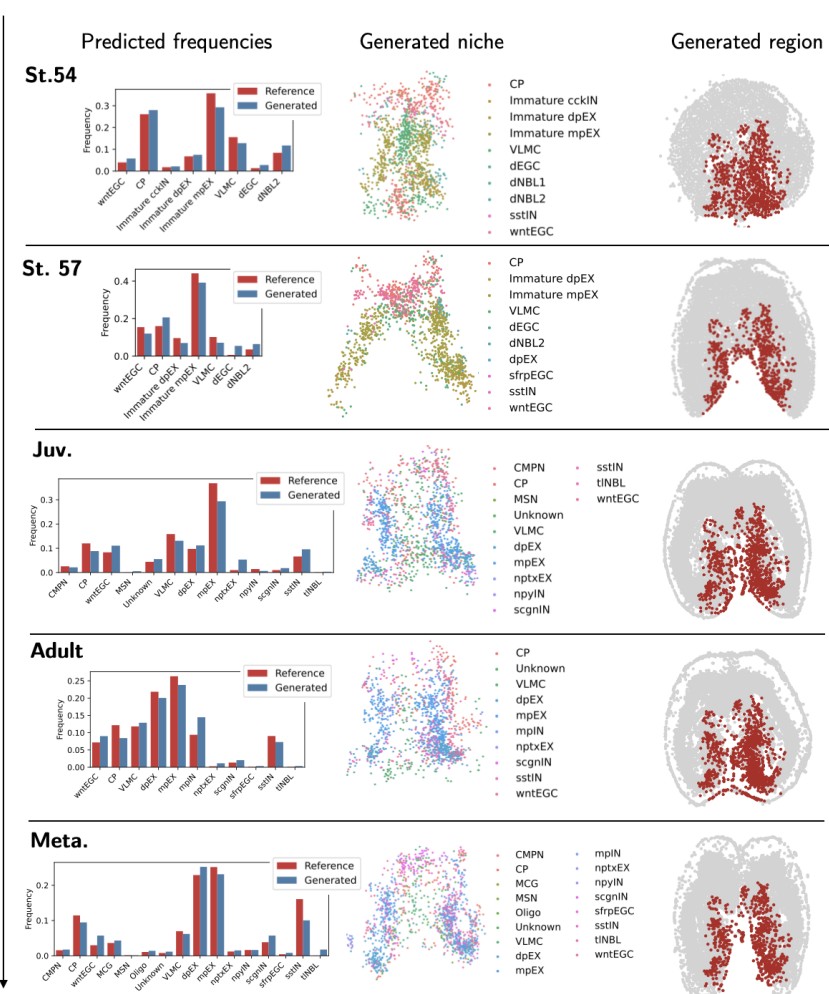

Figure 7: Prediction of hemisphere formation on the axolotl brain development. We evaluate the ability of NicheFlow to generate the anatomical splitting of central brain structure upon the formation of the right and left brain regions. The top panel shows the reference source region and cell type composition. For each stage (St.54, St.57, Juvenile, Adult, and Meta), we show: (1) predicted vs. reference cell type frequencies, (2) the generated niche visualized via 2D embedding colored by cell type, and (3) spatial projection of the generated region onto the anatomical reference (right). We set $\lambda = 0.5$.

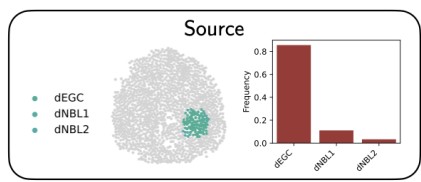

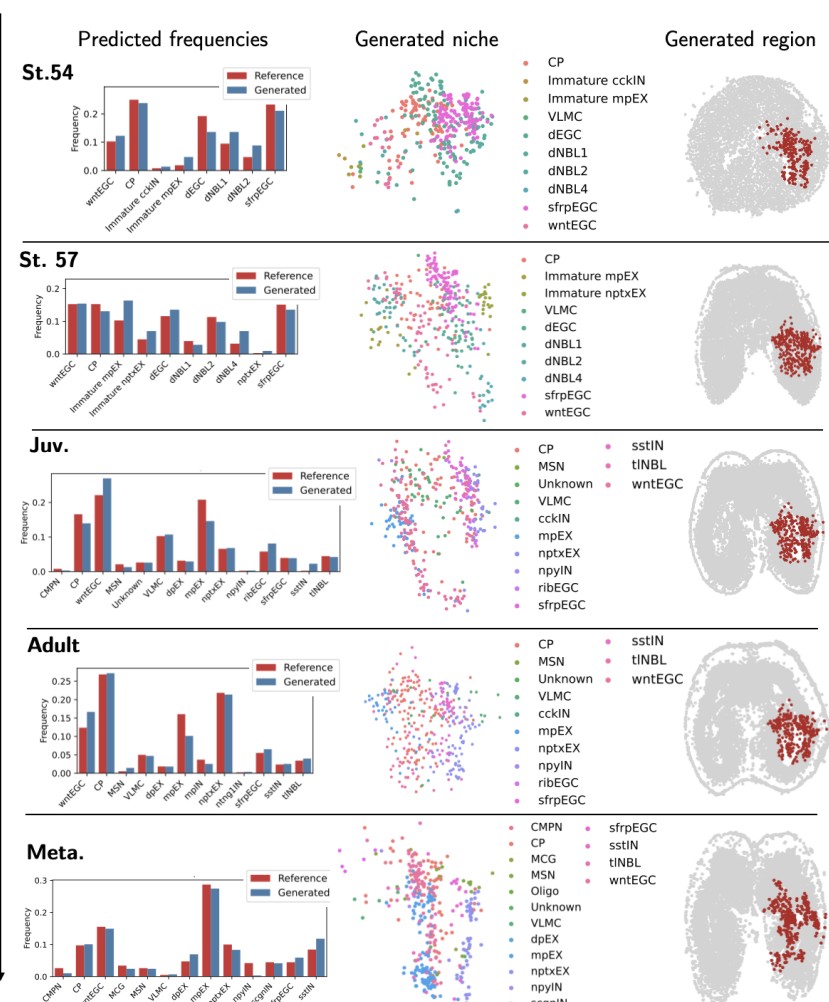

Figure 8: Prediction of cavity formation in the left brain lobe during axolotl development. We assess the ability of NicheFlow to model the emergence of a cavity structure within the left lobe of the axolotl brain. The top panel shows the reference source region and its cell type composition. For each developmental stage (St.54, St.57, Juvenile, Adult, and Meta), we display: (1) predicted versus reference cell type frequencies, (2) the generated niche visualized in 2D embedding space, colored by cell type, and (3) spatial projection of the generated region onto the anatomical brain reference (right). The progression illustrates the model's ability to recapitulate the asymmetric cavity formation localized to the left lobe. We set $\lambda = 0.5$.

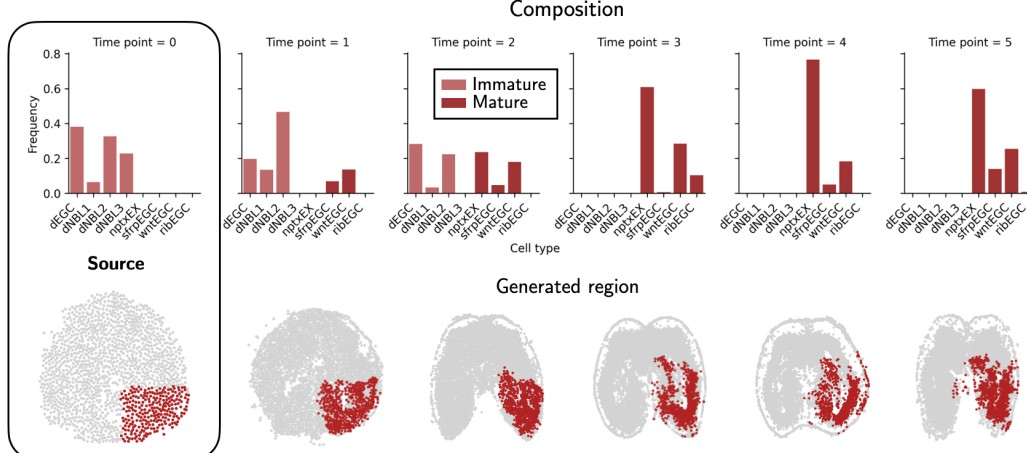

Figure 9: Structural and compositional prediction of the left dorsal pallium during the axolotl brain development. We use the left dorsal pallium region at the St.44 developmental stadium (highlighted on the left) and predict its trajectory over time. In the top row, we show the proportion of different cell types in the predicted region, colored as immature (light red) and mature (dark red). At the bottom, we show the structural prediction for the left dorsal pallium overlaid on the true slide. We set $\lambda = 0.5$.

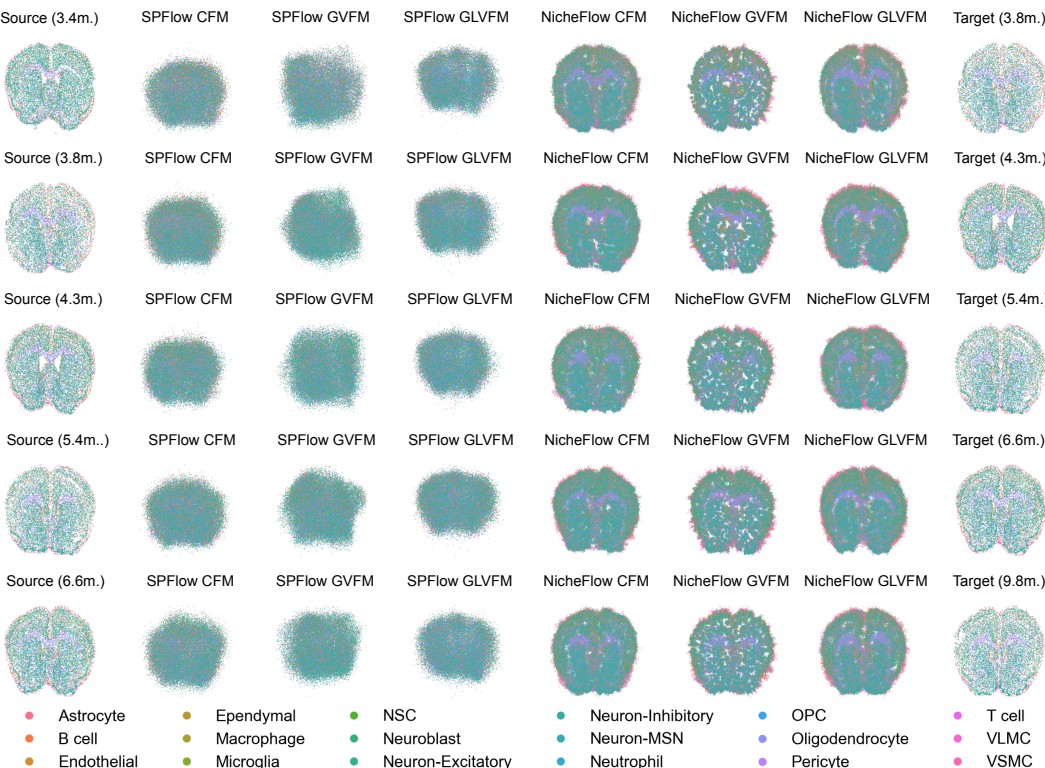

Figure 10: Qualitative comparison of generated samples on the mouse brain aging dataset (3.4, 3.8, 4.3, 5.4, 6.6, 9.8 months). We show source and target samples alongside predictions from SPFlow and NicheFlow with different objectives. NicheFlow captures the spatial structure and cell-type organization more faithfully across developmental stages.

## D.4    Conditional generation

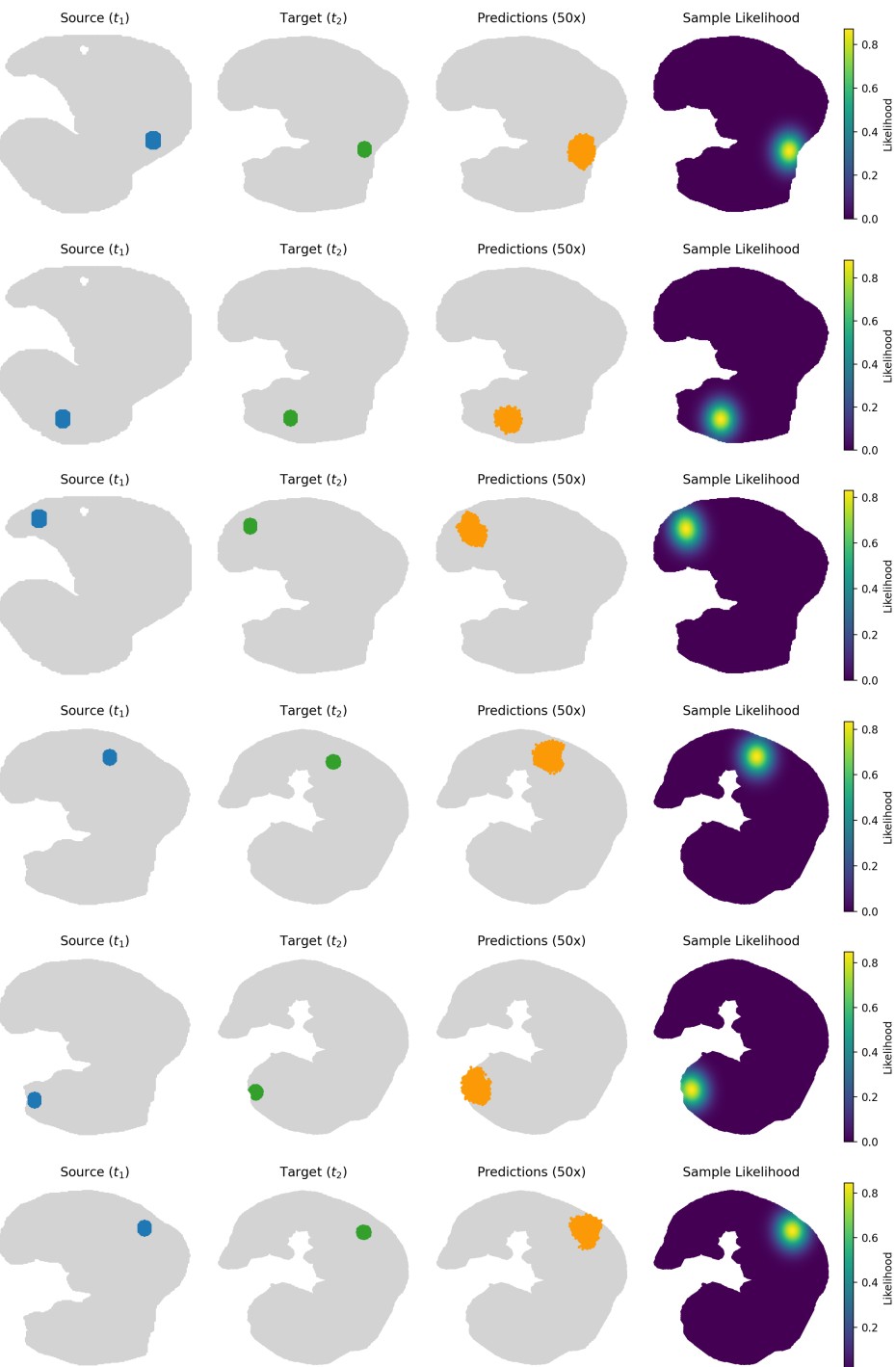

Figure 11: Qualitative evaluation of conditional generation with NicheFlow on the embryonic development dataset. Each row corresponds to a different source microenvironment at time $t_1$ (blue), shown alongside the ground truth target microenvironment at time $t_2$ (green). The third column displays 50 samples (orange) generated conditionally by the model, while the fourth column visualizes a kernel density estimate of the sample likelihood over the spatial domain.

To assess whether the model accurately conditions on input microenvironments, we visualize the spatial distribution of generated samples given a fixed source microenvironment. Figure 11 illustrates such cases on the embryonic development dataset, showing the input region at time $t_1$, the corresponding target at time $t_2$, and 50 independently generated samples from NicheFlow. Directly computing the likelihood of ground truth cell coordinates under our generative model is intractable, as it would require evaluating the density of an implicitly defined distribution over point clouds. Instead, we approximate the spatial likelihood using kernel density estimation (KDE) over Monte Carlo samples drawn from the model. Given a set of generated coordinates $\{\hat{c}_i\}_{i=1}^{N}$, we estimate the likelihood at a ground truth location $c$ as:

$$\hat{p}(c) = \frac{1}{N} \sum_{i=1}^{N} \exp\left( -\frac{\|c - \hat{c}_i\|^2}{2\sigma^2} \right), \tag{32}$$

where $\sigma$ is a fixed bandwidth parameter. The resulting KDE heatmap visualizes the spatial concentration of samples, allowing us to qualitatively assess whether the model produces consistent predictions conditioned on the source microenvironment.

### D.5   OT Ablation Study

We begin by emphasizing that the optimal choice of the OT feature-weighting parameter $\lambda$ in Eq. (12) depends on the downstream application. This parameter determines the relative importance assigned to gene expression versus spatial coordinates during OT. For developmental processes such as organogenesis or regeneration, higher values of $\lambda$ prioritize transcriptional similarity, facilitating the reconstruction of continuous differentiation trajectories and capturing fate-driven transitions, which may span large spatial distances. In contrast, for applications that aim to monitor changes within a fixed spatial region, such as shifts in cell-type composition over time, a lower $\lambda$ is more appropriate. In such cases, emphasizing spatial locality helps avoid spurious long-range transport assignments caused by molecular noise.

Table 3: Ablation study on the feature-coordinate trade-off parameter $\lambda$ in Eq. (12) across mouse embryonic development (MED), axolotl brain development (ABD), and mouse brain aging (MBA). We report 1NN-F1, PSD, and SPD for each training objective (CFM, GVFM, GLVFM) and model variant (RPCFlow, NicheFlow). All results are averaged over five evaluation runs. Bold indicates the best value per dataset and metric.

| | | | MED | | | ABD | | | MBA | | |
|---|---|---|---|---|---|---|---|---|---|---|---|
| Model | Obj. | $\lambda$ | 1NN-F1 ↑ | PSD ↓ | SPD ↓ | 1NN-F1 ↑ | PSD ↓ | SPD ↓ | 1NN-F1 ↑ | PSD ↓ | SPD ↓ |
| RPCFlow | CFM | 0.10 | 0.546 ± 0.0012 | 0.981 ± 0.0024 | 0.564 ± 0.0015 | 0.524 ± 0.0020 | 2.051 ± 0.0039 | 1.015 ± 0.0036 | 0.271 ± 0.0004 | 1.543 ± 0.0016 | 0.810 ± 0.0010 |
| RPCFlow | CFM | 0.25 | 0.545 ± 0.0004 | 0.958 ± 0.0040 | 0.570 ± 0.0011 | 0.511 ± 0.0015 | 2.079 ± 0.0053 | 1.024 ± 0.0046 | 0.273 ± 0.0005 | **1.535 ± 0.0009** | 0.818 ± 0.0013 |
| RPCFlow | CFM | 0.50 | 0.554 ± 0.0009 | 0.988 ± 0.0031 | 0.562 ± 0.0011 | 0.507 ± 0.0011 | 2.077 ± 0.0065 | 1.023 ± 0.0032 | 0.273 ± 0.0006 | 1.546 ± 0.0010 | 0.810 ± 0.0013 |
| RPCFlow | CFM | 0.75 | 0.537 ± 0.0007 | 0.981 ± 0.0020 | 0.595 ± 0.0022 | 0.517 ± 0.0005 | 2.033 ± 0.0072 | 1.026 ± 0.0050 | 0.275 ± 0.0003 | 1.553 ± 0.0008 | 0.801 ± 0.0010 |
| RPCFlow | GLVFM | 0.10 | 0.586 ± 0.0016 | 0.979 ± 0.0021 | 0.586 ± 0.0012 | 0.554 ± 0.0007 | 2.053 ± 0.0044 | 1.038 ± 0.0025 | 0.265 ± 0.0004 | 1.723 ± 0.0015 | 0.779 ± 0.0011 |
| RPCFlow | GLVFM | 0.25 | 0.593 ± 0.0003 | 0.924 ± 0.0019 | 0.575 ± 0.0017 | 0.555 ± 0.0013 | 2.076 ± 0.0057 | 1.036 ± 0.0024 | 0.267 ± 0.0003 | 1.728 ± 0.0016 | 0.777 ± 0.0005 |
| RPCFlow | GLVFM | 0.50 | 0.586 ± 0.0013 | 0.934 ± 0.0019 | 0.569 ± 0.0014 | 0.551 ± 0.0008 | 2.038 ± 0.0045 | 1.032 ± 0.0034 | 0.269 ± 0.0002 | 1.715 ± 0.0014 | 0.783 ± 0.0006 |
| RPCFlow | GLVFM | 0.75 | 0.593 ± 0.0009 | 0.948 ± 0.0035 | 0.570 ± 0.0011 | 0.561 ± 0.0008 | **2.014 ± 0.0056** | 1.035 ± 0.0047 | 0.268 ± 0.0005 | 1.675 ± 0.0019 | 0.784 ± 0.0013 |
| RPCFlow | GVFM | 0.10 | 0.503 ± 0.0013 | 1.155 ± 0.0044 | 0.578 ± 0.0007 | 0.477 ± 0.0008 | 2.260 ± 0.0077 | 1.036 ± 0.0031 | 0.249 ± 0.0003 | 1.753 ± 0.0020 | 0.784 ± 0.0010 |
| RPCFlow | GVFM | 0.25 | 0.520 ± 0.0010 | 1.223 ± 0.0044 | 0.566 ± 0.0011 | 0.478 ± 0.0014 | 2.364 ± 0.0065 | 1.030 ± 0.0042 | 0.246 ± 0.0005 | 1.710 ± 0.0013 | 0.787 ± 0.0014 |
| RPCFlow | GVFM | 0.50 | 0.521 ± 0.0012 | 1.185 ± 0.0032 | 0.569 ± 0.0009 | 0.480 ± 0.0008 | 2.360 ± 0.0036 | 1.025 ± 0.0015 | 0.245 ± 0.0004 | 1.756 ± 0.0025 | 0.774 ± 0.0007 |
| RPCFlow | GVFM | 0.75 | 0.514 ± 0.0013 | 1.202 ± 0.0014 | 0.573 ± 0.0008 | 0.471 ± 0.0012 | 2.476 ± 0.0082 | 1.037 ± 0.0036 | 0.248 ± 0.0003 | 1.831 ± 0.0015 | 0.780 ± 0.0012 |
| NicheFlow | CFM | 0.10 | 0.609 ± 0.0030 | 0.979 ± 0.0228 | 0.402 ± 0.0036 | 0.604 ± 0.0018 | 2.086 ± 0.0058 | 0.568 ± 0.0030 | 0.283 ± 0.0003 | 1.557 ± 0.0014 | 0.556 ± 0.0028 |
| NicheFlow | CFM | 0.25 | 0.569 ± 0.0031 | 0.973 ± 0.0074 | 0.425 ± 0.0062 | 0.586 ± 0.0013 | 2.106 ± 0.0072 | **0.565 ± 0.0039** | 0.281 ± 0.0006 | 1.546 ± 0.0029 | 0.612 ± 0.0032 |
| NicheFlow | CFM | 0.50 | 0.551 ± 0.0009 | 1.051 ± 0.0496 | 0.471 ± 0.0110 | 0.585 ± 0.0013 | 2.089 ± 0.0066 | 0.591 ± 0.0026 | 0.283 ± 0.0002 | 1.588 ± 0.0033 | 0.604 ± 0.0061 |
| NicheFlow | CFM | 0.75 | 0.519 ± 0.0038 | 1.103 ± 0.0323 | 0.515 ± 0.0101 | 0.571 ± 0.0012 | 2.126 ± 0.0110 | 0.592 ± 0.0044 | 0.278 ± 0.0003 | 1.566 ± 0.0027 | 0.616 ± 0.0041 |
| NicheFlow | GVFM | 0.10 | 0.596 ± 0.0027 | 0.991 ± 0.0137 | 0.406 ± 0.0025 | 0.574 ± 0.0015 | 2.220 ± 0.0107 | 0.594 ± 0.0046 | 0.268 ± 0.0003 | 1.661 ± 0.0033 | **0.531 ± 0.0010** |
| NicheFlow | GVFM | 0.25 | 0.563 ± 0.0027 | 1.051 ± 0.0117 | 0.491 ± 0.0105 | 0.571 ± 0.0009 | 2.343 ± 0.0136 | 0.619 ± 0.0061 | 0.265 ± 0.0006 | 1.599 ± 0.0032 | 0.590 ± 0.0018 |
| NicheFlow | GVFM | 0.50 | 0.533 ± 0.0053 | 1.034 ± 0.0452 | 0.800 ± 0.0401 | 0.556 ± 0.0016 | 2.283 ± 0.0126 | 0.742 ± 0.0134 | 0.269 ± 0.0005 | 1.607 ± 0.0029 | 0.605 ± 0.0022 |
| NicheFlow | GVFM | 0.75 | 0.526 ± 0.0028 | 1.121 ± 0.0112 | 0.870 ± 0.0336 | 0.556 ± 0.0013 | 2.166 ± 0.0091 | 0.684 ± 0.0128 | 0.263 ± 0.0004 | 1.613 ± 0.0031 | 0.731 ± 0.0040 |
| NicheFlow | GLVFM | 0.10 | **0.664 ± 0.0014** | 0.883 ± 0.0094 | 0.398 ± 0.0023 | **0.628 ± 0.0013** | 2.079 ± 0.0043 | 0.576 ± 0.0055 | **0.285 ± 0.0003** | 1.554 ± 0.0021 | 0.532 ± 0.0009 |
| NicheFlow | GLVFM | 0.25 | 0.629 ± 0.0033 | 0.923 ± 0.0109 | **0.394 ± 0.0035** | 0.618 ± 0.0014 | 2.102 ± 0.0028 | 0.577 ± 0.0051 | 0.284 ± 0.0004 | 1.599 ± 0.0035 | 0.549 ± 0.0012 |
| NicheFlow | GLVFM | 0.50 | 0.610 ± 0.0036 | 0.909 ± 0.0107 | 0.417 ± 0.0060 | 0.610 ± 0.0008 | 2.136 ± 0.0025 | 0.579 ± 0.0014 | 0.283 ± 0.0005 | 1.600 ± 0.0039 | 0.546 ± 0.0009 |
| NicheFlow | GLVFM | 0.75 | 0.592 ± 0.0019 | **0.879 ± 0.0054** | 0.472 ± 0.0150 | 0.603 ± 0.0014 | 2.106 ± 0.0060 | 0.592 ± 0.0073 | 0.281 ± 0.0004 | 1.573 ± 0.0008 | 0.595 ± 0.0026 |

In Tab. 3, we sweep the value of $\lambda$ across multiple models and training strategies on the same task as in Sec. 5.1, showing consistent results across settings as reported in Tab. 1.

Furthermore, Figure 12 visualizes the impact of varying $\lambda$ on the OT plans described in Sec. 4.2, where intermediate values yield more coherent and biologically plausible couplings.

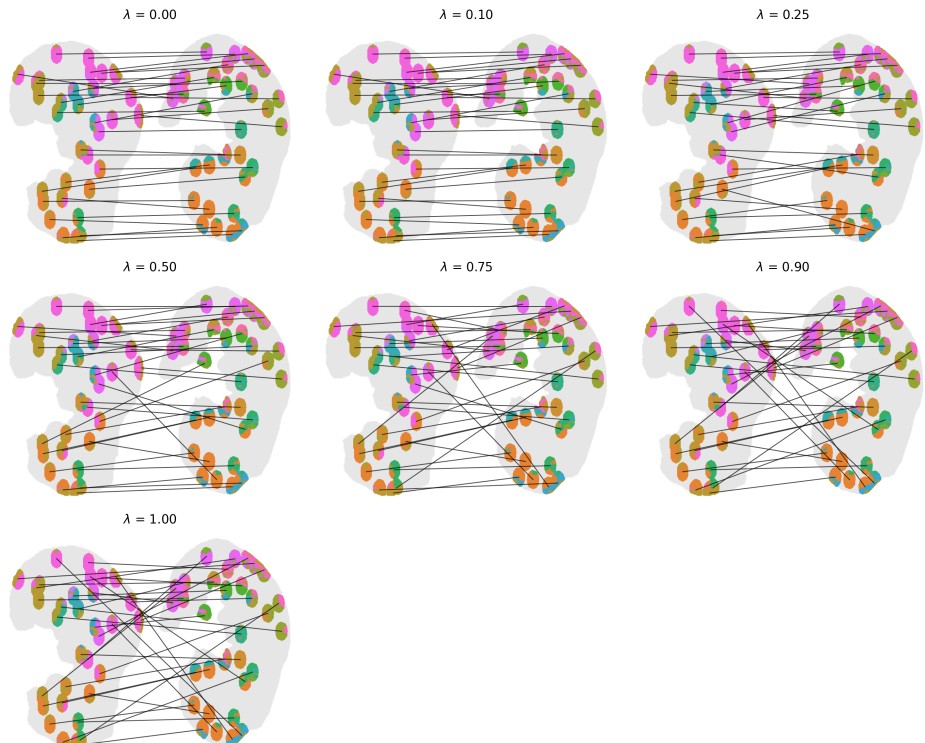

Figure 12: Visualization of OT couplings computed under varying values of the pooling parameter $\lambda$ in eq. (12), which balances spatial coordinates and gene expression in microenvironment matching. Lower $\lambda$ values prioritize spatial proximity, resulting in more dispersed and less structured alignments, while intermediate values yield tighter, biologically consistent mappings. Very high $\lambda$ settings ignore spatial context and may lead to implausible long-range matches.

## D.6 $K$-Means regions ablation study

To ensure diverse and spatially distributed sampling during training, we partition each tissue section into $K$ spatial regions using $K$-Means clustering over the 2D cell coordinates (see Sec. 5.1.1). At each training step, microenvironments are sampled uniformly from within these regions, encouraging broad spatial coverage and preventing oversampling of densely populated areas. In this ablation study, we investigate how varying the number of spatial regions, $K$, affects model performance.

Table 4: Ablation study of the number of spatial regions $K$ defined over the datasets. We evaluate NicheFlow with the GLVFM objective across three datasets: mouse embryonic development (MED), axolotl brain development (ABD), and mouse brain aging (MBA). Results are reported as mean ± standard deviation over five evaluation runs.

| $K$ | MED | | | ABD | | | MBA | | |
|---|---|---|---|---|---|---|---|---|---|
| | 1NN-F1 $\uparrow$ | PSD $\downarrow (10^2)$ | SPD $\downarrow (10^2)$ | 1NN-F1 $\uparrow$ | PSD $\downarrow (10^2)$ | SPD $\downarrow (10^2)$ | 1NN-F1 $\uparrow$ | PSD $\downarrow (10^2)$ | SPD $\downarrow (10^2)$ |
| 8 | $0.640_{\pm 0.0043}$ | $\mathbf{0.876_{\pm 0.0055}}$ | $0.441_{\pm 0.0058}$ | $0.617_{\pm 0.0017}$ | $1.954_{\pm 0.0057}$ | $0.631_{\pm 0.0073}$ | $0.283_{\pm 0.0005}$ | $1.561_{\pm 0.0018}$ | $0.571_{\pm 0.0019}$ |
| 16 | $0.661_{\pm 0.0033}$ | $0.881_{\pm 0.0068}$ | $0.393_{\pm 0.0056}$ | $\mathbf{0.633_{\pm 0.0008}}$ | $\mathbf{1.936_{\pm 0.0027}}$ | $0.628_{\pm 0.0104}$ | $0.283_{\pm 0.0003}$ | $1.564_{\pm 0.0035}$ | $0.538_{\pm 0.0021}$ |
| 32 | $0.659_{\pm 0.0025}$ | $0.899_{\pm 0.0120}$ | $\mathbf{0.391_{\pm 0.0029}}$ | $0.622_{\pm 0.0005}$ | $1.968_{\pm 0.0036}$ | $0.640_{\pm 0.0124}$ | $0.279_{\pm 0.0005}$ | $1.600_{\pm 0.0040}$ | $0.537_{\pm 0.0015}$ |
| 64 | $\mathbf{0.664_{\pm 0.0014}}$ | $0.883_{\pm 0.0094}$ | $0.398_{\pm 0.0023}$ | $0.628_{\pm 0.0013}$ | $2.079_{\pm 0.0043}$ | $\mathbf{0.576_{\pm 0.0055}}$ | $\mathbf{0.285_{\pm 0.0003}}$ | $\mathbf{1.554_{\pm 0.0021}}$ | $\mathbf{0.532_{\pm 0.0009}}$ |

As shown in Figure 13, increasing $K$ leads to increasingly fine-grained spatial partitions. While moderate values of $K$ help improve spatial resolution, excessively high values (e.g., $K = 128$ or 256) result in overly small and fragmented regions. This can cause significant overlap between sampled microenvironments and reduce sampling diversity. Moreover, in sparsely populated tissue sections, high $K$ values may yield regions with insufficient cells, degrading both representativeness and stability.

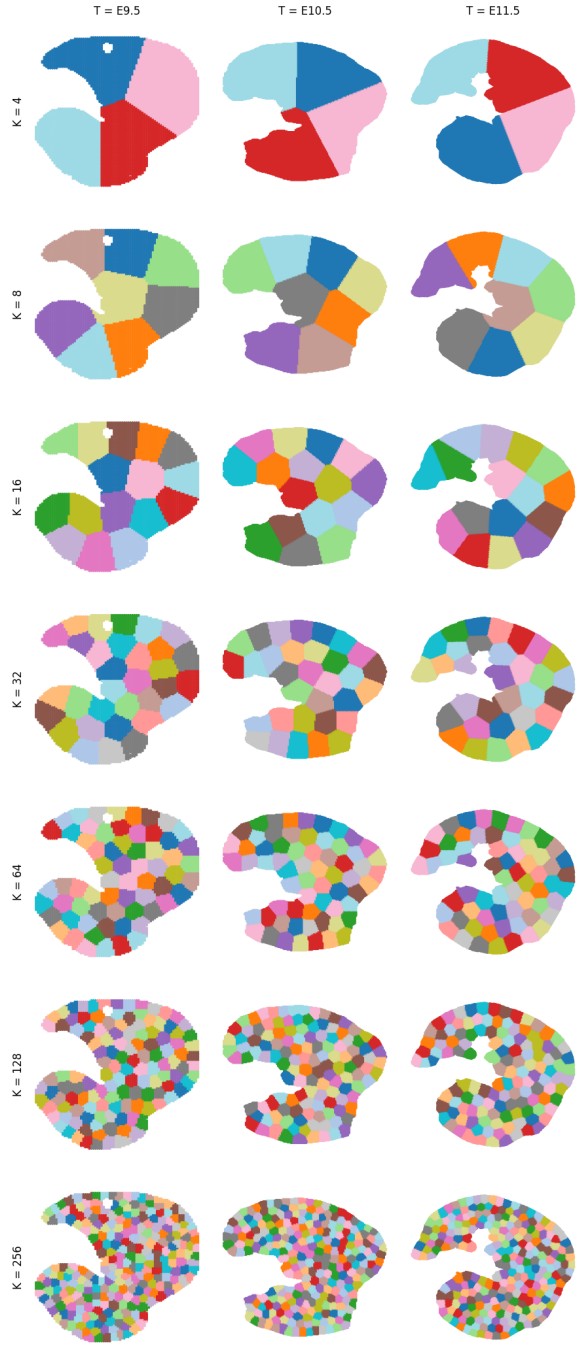

Figure 13: Visualization of spatial partitions obtained via KMeans clustering with $K = \{4, 8, 16, 32, 64, 128, 256\}$ on the embryonic development dataset. For low $K$, each region covers large heterogeneous areas; for high $K$, regions become small, dense, and highly overlapping, potentially degrading the diversity and utility of sampled microenvironments.

Conversely, using too few regions (e.g., $K = 4$) results in broad spatial partitions that may encompass multiple heterogeneous tissue compartments. This undermines the locality assumptions of our model and increases intra-region variability, which can impair the model's ability to learn.

Given the trade-offs outlined above, we focus our evaluation on $K \in \{8, 16, 32, 64\}$, which spans a range of granularities that preserve both spatial interpretability and sampling robustness. Tab. 4 sum-

marizes the quantitative results for this ablation across three datasets: mouse embryonic development (MED), axolotl brain development (ABD), and mouse brain aging (MBA). We observe that using $K = 64$ consistently yields strong performance, achieving the highest 1NN-F1 scores on both the MED and MBA datasets, while also performing competitively on ABD. These findings indicate that $K = 64$ offers an effective balance between spatial resolution and stability, and we adopt it as the default configuration in our main experiments.

### D.7 Justifying the choice of NicheFlow over RPCFlow

Although RPCFlow achieves competitive performance on spatial metrics such as PSD and SPD, it lacks the essential capability of meaningful conditional generation. In RPCFlow, conditioning is performed using randomly sampled point clouds from the spatial regions without explicit microenvironment structure. As shown in Fig. 14, when conditioned on such random sources, RPCFlow tends to reconstruct the entire target tissue, rather than capturing local dynamics driven by the source input. This undermines its ability to model spatiotemporal evolution in a biologically grounded manner.

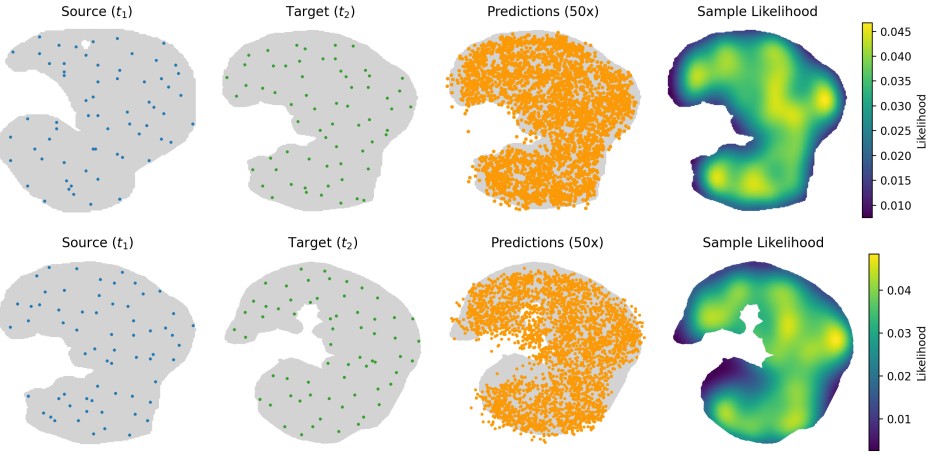

Figure 14: RPCFlow fails to perform meaningful conditional generation: the model generates a diffuse reconstruction resembling the entire target tissue rather than conditioning on the provided source points. Each row shows a different source-target pair.

In contrast, NicheFlow is explicitly designed to push localized microenvironments through time. By conditioning on fixed-radius neighborhoods centered around specific spatial positions, the model learns how cellular contexts evolve, preserving both spatial coherence and transcriptional identity. Figure 15 visualizes this distinction: while NicheFlow consistently generates well-localized predictions aligned with the input microenvironment, RPCFlow fails to retain spatial specificity, often diffusing the prediction across broader regions.

From a biological perspective, predicting the fate of a local tissue region over time is far more relevant than mapping random point sets. Microenvironments encode structured cellular contexts, such as signaling interactions or niche-specific cell states, that are crucial for downstream analysis (e.g., lineage fate prediction, microenvironment-based intervention simulation). Because RPCFlow lacks this interpretability and fails to enable such downstream tasks, it cannot serve as a practical generative model in biological settings.

In summary, while RPCFlow may appear performant under some aggregate spatial metrics, only NicheFlow enables localized, conditionally consistent generation of evolving tissue regions. This makes it not only superior for evaluation but also for practical use in biological modeling.

### D.8 Structure-aware evaluation

While we refer to our primary evaluation metrics as PSD and SPD, they correspond to the two asymmetric directions of the Chamfer Distance (CD) - a widely used metric in point cloud generation [46]. PSD measures how closely the predicted points adhere to the ground truth structure (fidelity), while SPD captures how well the prediction covers the full extent of the target (coverage). We

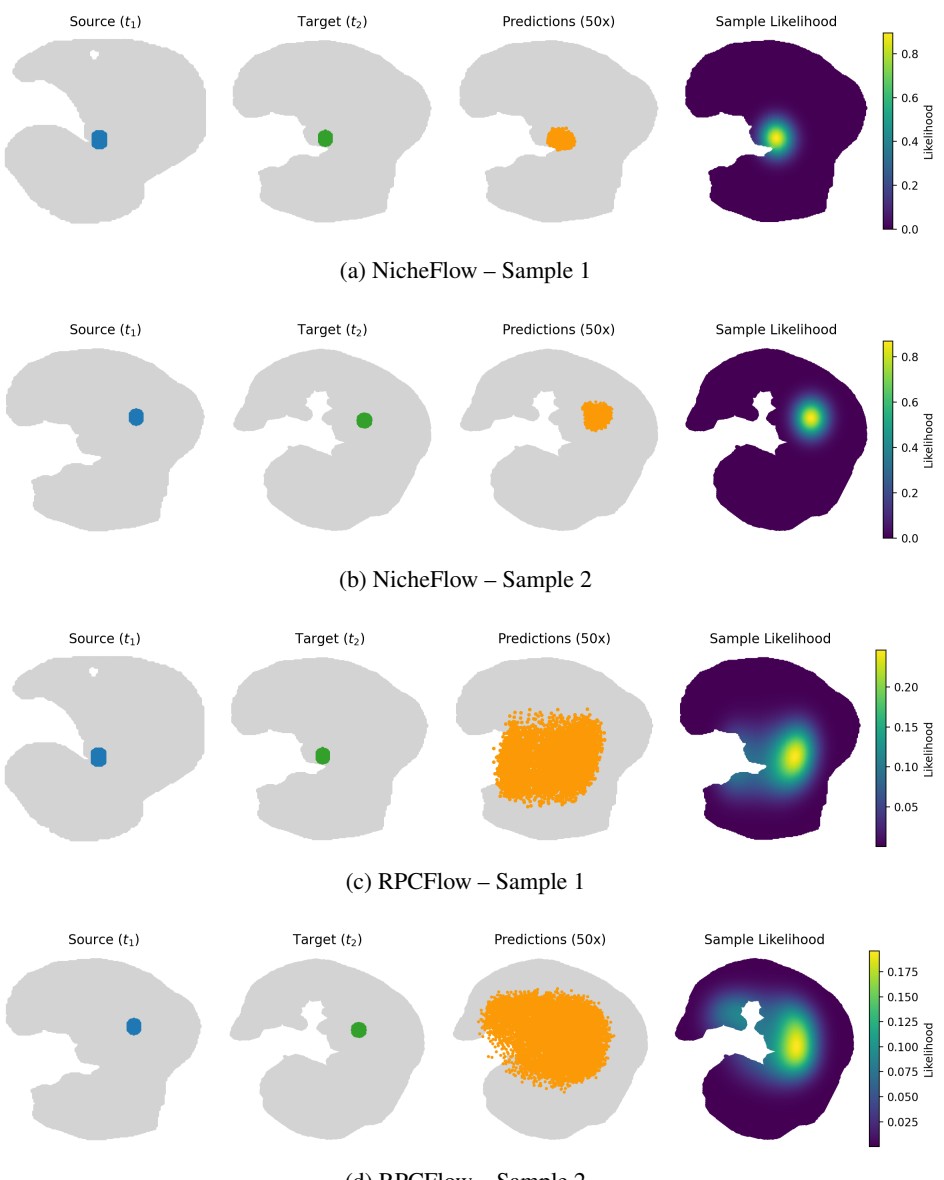

Figure 15: Comparison between NicheFlow and RPCFlow in a fixed-source microenvironment setting. Each row shows input (source), ground truth target, model prediction, and KDE-estimated likelihood. While NicheFlow (top) produces well-localized samples consistent with the input, RPCFlow (bottom) generates less structured and spatially inaccurate predictions.

compute both metrics over collections of microenvironments tiled across the tissue, providing an indirect assessment of global morphological accuracy.

To directly assess the preservation of internal structure within generated microenvironments, we additionally evaluate Gromov-Wasserstein (GW) and Fused Gromov-Wasserstein (FGW) distances [47, 40]. These metrics compare the intra-point pairwise distances in predicted and ground truth point clouds, capturing latent structural relationships. FGW further incorporates transcriptomic similarity, offering a holistic measure of structural and feature-based alignment.

Given the computational cost of GW and FGW, we restrict their evaluation to microenvironment-level generations. For each source niche, we sample predictions from the trained model and compute GW/FGW against the corresponding real microenvironment, averaging results across multiple runs.

The source–target pairs are obtained by computing OT between all microenvironments at time $t$ and $t + 1$, yielding a one-to-one coupling used for conditioning and evaluation.

As SPFlow generates individual points and RPCFlow produces randomly sampled, unstructured clouds, these models lack coherent internal structure and are not amenable to structural comparison via GW or FGW. We therefore report these metrics only for models that generate full microenvironments.

The results in Tab. 5 show that NicheFlow, particularly when trained with the GLVFM objective, consistently achieves lower GW and FGW distances compared to baseline models. These findings underscore the structural fidelity of our predictions and further validate the modeling advantages of microenvironment-aware generative training.

Table 5: Gromov–Wasserstein (GW) and Fused Gromov–Wasserstein (FGW) distances between generated and real microenvironments on mouse embryonic development (MED), axolotl brain development (ABD), and mouse brain aging (MBA). All models are trained with a fixed $\lambda = 0.1$. Results are averaged over five generation runs. Bold indicates the best (lowest) value per column.

| Model | Objective | MED | | ABD | | MBA | |
|---|---|---|---|---|---|---|---|
| | | GW $(10^2)$ | FGW | GW $(10^2)$ | FGW | GW $(10^2)$ | FGW |
| NicheFlow | CFM | $0.315_{\pm 0.003}$ | $3.273_{\pm 0.003}$ | $0.783_{\pm 0.002}$ | $3.543_{\pm 0.004}$ | $0.315_{\pm 0.001}$ | $3.531_{\pm 0.000}$ |
| NicheFlow | GVFM | $0.463_{\pm 0.004}$ | $3.167_{\pm 0.007}$ | $0.797_{\pm 0.006}$ | $\mathbf{3.414_{\pm 0.003}}$ | $0.339_{\pm 0.000}$ | $3.387_{\pm 0.000}$ |
| NicheFlow | GLVFM | $\mathbf{0.224_{\pm 0.001}}$ | $\mathbf{3.147_{\pm 0.007}}$ | $\mathbf{0.720_{\pm 0.004}}$ | $3.420_{\pm 0.003}$ | $\mathbf{0.262_{\pm 0.000}}$ | $\mathbf{3.367_{\pm 0.000}}$ |

## D.9 Wasserstein metrics

To complement the local structural assessments with GW and FGW, we additionally include global evaluation metrics that are widely adopted in spatial transcriptomics [48–50]. Specifically, we report the 1-Wasserstein ($\mathcal{W}_1$) and 2-Wasserstein ($\mathcal{W}_2$) distances between predicted and real cell distributions, computed separately over spatial coordinates and gene expression features. These metrics are calculated at the level of individual cell types and averaged across all timepoints and generated samples. This provides a more global perspective on how well the model reconstructs tissue-wide spatial and transcriptional distributions.

Table 6: Comparison of Wasserstein distances between generated and real tissue structures on the mouse embryonic development (MED) dataset. We compute $\mathcal{W}_1$ and $\mathcal{W}_2$ distances separately for spatial coordinates (Pos.) and gene expression features (Genes), averaged across all generated cell types and timepoints. All models are trained with a fixed value of $\lambda = 0.1$, and results are averaged over five generation runs. Bold indicates the best (lowest) value per column.

| Model | Obj. | $\mathcal{W}_1$ Pos. | $\mathcal{W}_1$ Genes | $\mathcal{W}_2$ Pos. | $\mathcal{W}_2$ Genes |
|---|---|---|---|---|---|
| SPFlow | CFM | $0.634_{\pm 0.004}$ | $6.545_{\pm 0.004}$ | $0.773_{\pm 0.005}$ | $5.585_{\pm 0.005}$ |
| SPFlow | GVFM | $0.612_{\pm 0.005}$ | $6.517_{\pm 0.002}$ | $0.743_{\pm 0.005}$ | $5.609_{\pm 0.113}$ |
| SPFlow | GLVFM | $0.613_{\pm 0.001}$ | $6.457_{\pm 0.003}$ | $0.738_{\pm 0.001}$ | $5.546_{\pm 0.108}$ |
| RPCFlow | CFM | $0.290_{\pm 0.006}$ | $6.303_{\pm 0.004}$ | $0.460_{\pm 0.007}$ | $5.386_{\pm 0.004}$ |
| RPCFlow | GVFM | $0.258_{\pm 0.004}$ | $6.134_{\pm 0.004}$ | $0.398_{\pm 0.006}$ | $5.292_{\pm 0.114}$ |
| RPCFlow | GLVFM | $0.221_{\pm 0.003}$ | $6.087_{\pm 0.002}$ | $0.365_{\pm 0.007}$ | $5.244_{\pm 0.113}$ |
| NicheFlow | CFM | $0.253_{\pm 0.002}$ | $6.177_{\pm 0.001}$ | $0.420_{\pm 0.003}$ | $5.314_{\pm 0.111}$ |
| NicheFlow | GVFM | $0.222_{\pm 0.004}$ | $5.985_{\pm 0.005}$ | $0.352_{\pm 0.006}$ | $5.173_{\pm 0.122}$ |
| NicheFlow | GLVFM | $\mathbf{0.212_{\pm 0.004}}$ | $\mathbf{5.930_{\pm 0.003}}$ | $\mathbf{0.342_{\pm 0.007}}$ | $\mathbf{5.031_{\pm 0.003}}$ |

We evaluate all model variants and training objectives across the three datasets (MED, ABD, MBA) and report the results in Tabs. 6 to 8. NicheFlow with the GLVFM objective consistently achieves the lowest Wasserstein distances across nearly all dimensions in the embryonic development dataset (MED), outperforming both baseline models and alternative objectives.

In the ABD and MBA datasets, which exhibit more gradual spatial evolution or less distinct cell-type boundaries, we observe that RPCFlow sometimes matches or marginally outperforms NicheFlow in isolated metrics. However, as we discussed in App. D.7, RPCFlow is not a conditionally consistent model: it does not produce meaningful trajectories given a source microenvironment and lacks

Table 7: Comparison of Wasserstein distances between generated and real tissue structures on the axolotl brain development (ABD) dataset. We report $\mathcal{W}_1$ and $\mathcal{W}_2$ distances separately for spatial positions (Pos.) and gene expression features (Genes), averaged across all generated cell types and timepoints. All models use $\lambda = 0.1$, and results are averaged over five generation runs. Bold indicates the best (lowest) value per column.

| Model | Obj. | $\mathcal{W}_1$ Pos. | $\mathcal{W}_1$ Genes | $\mathcal{W}_2$ Pos. | $\mathcal{W}_2$ Genes |
|---|---|---|---|---|---|
| SPFlow | CFM | $0.474 \pm 0.003$ | $6.711 \pm 0.002$ | $0.573 \pm 0.003$ | $6.096 \pm 0.081$ |
| SPFlow | GVFM | $0.538 \pm 0.002$ | $6.681 \pm 0.003$ | $0.639 \pm 0.003$ | $6.209 \pm 0.099$ |
| SPFlow | GLVFM | $0.489 \pm 0.002$ | $6.533 \pm 0.002$ | $0.586 \pm 0.003$ | $6.090 \pm 0.166$ |
| RPCFlow | CFM | $\mathbf{0.234 \pm 0.003}$ | $6.363 \pm 0.004$ | $0.374 \pm 0.005$ | $6.031 \pm 0.089$ |
| RPCFlow | GVFM | $0.247 \pm 0.002$ | $6.093 \pm 0.006$ | $\mathbf{0.369 \pm 0.005}$ | $5.921 \pm 0.051$ |
| RPCFlow | GLVFM | $0.254 \pm 0.002$ | $\mathbf{6.070 \pm 0.004}$ | $0.406 \pm 0.003$ | $\mathbf{5.884 \pm 0.042}$ |
| NicheFlow | CFM | $0.267 \pm 0.002$ | $6.347 \pm 0.003$ | $0.428 \pm 0.005$ | $6.107 \pm 0.003$ |
| NicheFlow | GVFM | $0.284 \pm 0.004$ | $6.081 \pm 0.004$ | $0.448 \pm 0.006$ | $5.917 \pm 0.036$ |
| NicheFlow | GLVFM | $0.279 \pm 0.006$ | $6.085 \pm 0.003$ | $0.444 \pm 0.011$ | $5.923 \pm 0.034$ |

Table 8: Comparison of Wasserstein distances between generated and real tissue structures on the mouse brain aging (MBA) dataset. We compute $\mathcal{W}_1$ and $\mathcal{W}_2$ distances for spatial coordinates (Pos.) and gene expression features (Genes), averaging per cell type and timepoint. All models are trained with $\lambda = 0.1$, and results are averaged across five generation runs. Bold marks the lowest value in each column.

| Model | Obj. | $\mathcal{W}_1$ Pos. | $\mathcal{W}_1$ Genes | $\mathcal{W}_2$ Pos. | $\mathcal{W}_2$ Genes |
|---|---|---|---|---|---|
| SPFlow | CFM | $0.515 \pm 0.001$ | $7.544 \pm 0.004$ | $0.600 \pm 0.002$ | $5.237 \pm 0.056$ |
| SPFlow | GVFM | $0.564 \pm 0.001$ | $7.461 \pm 0.004$ | $0.646 \pm 0.001$ | $4.961 \pm 0.114$ |
| SPFlow | GLVFM | $0.552 \pm 0.002$ | $7.431 \pm 0.003$ | $0.632 \pm 0.002$ | $\mathbf{4.117 \pm 0.196}$ |
| RPCFlow | CFM | $\mathbf{0.385 \pm 0.003}$ | $7.318 \pm 0.006$ | $\mathbf{0.480 \pm 0.003}$ | $6.659 \pm 0.098$ |
| RPCFlow | GVFM | $0.416 \pm 0.003$ | $7.202 \pm 0.003$ | $0.510 \pm 0.003$ | $6.346 \pm 0.061$ |
| NicheFlow | CFM | $0.416 \pm 0.008$ | $7.321 \pm 0.012$ | $0.510 \pm 0.008$ | $6.487 \pm 0.084$ |
| NicheFlow | GVFM | $0.412 \pm 0.004$ | $\mathbf{7.102 \pm 0.001}$ | $0.503 \pm 0.004$ | $6.403 \pm 0.087$ |
| NicheFlow | GLVFM | $0.431 \pm 0.008$ | $7.121 \pm 0.008$ | $0.527 \pm 0.007$ | $6.298 \pm 0.029$ |

biological interpretability. Therefore, even when RPCFlow achieves slightly lower Wasserstein distances in aggregate, these gains are not actionable in practice, as the model cannot be used to infer biologically meaningful transitions or trace the evolution of specific spatial regions.

Taken together, Wasserstein metrics offer a complementary, global validation of the structural and semantic accuracy of NicheFlow. They reinforce the quantitative evidence already provided by Chamfer-based metrics (PSD/SPD) and Gromov-based structural evaluations (GW/FGW), confirming that our model accurately captures the tissue-scale organization of both spatial and gene expression patterns.

# E  Algorithms

Here, we present the algorithmic procedures underlying NicheFlow. To streamline the exposition, we first define compact notation for representing noisy microenvironments and their interpolations.

We denote a single noisy microenvironment (simplifying Eq. (11)) as:

$$\mathcal{M}^z \sim \mathcal{N}(\mathbf{0}, \mathbf{I}_{D+2})^{1 \times k} = \{(\boldsymbol{c}_i^z, \boldsymbol{z}_i) \mid \boldsymbol{c}_i^z \sim \mathcal{N}(\mathbf{0}, \mathbf{I}_2),\ \boldsymbol{z}_i \sim \mathcal{N}(\mathbf{0}, \mathbf{I}_D),\ \forall i = 1, \dots, k\}, \quad (33)$$

where $k$ denotes the number of spatial points per microenvironment, $\boldsymbol{c}_i^z$ are 2D cell coordinates, and $\boldsymbol{z}_i$ are associated feature vectors in $\mathbb{R}^D$. For a collection of $N$ microenvironments, we define the corresponding set of noisy microenvironments

$$\boldsymbol{\mathcal{M}^z} \sim \mathcal{N}(\mathbf{0}, \mathbf{I}_{D+2})^{N \times k} = \left\{ \mathcal{M}_i^z \sim \mathcal{N}(\mathbf{0}, \mathbf{I}_{D+2})^{1 \times k} \mid \forall i = 1, \dots, N \right\}. \quad (34)$$

Given a noisy sample $\mathcal{M}^z$ and a corresponding ground-truth microenvironment $\mathcal{M}^1$, we define their linear interpolation at time $t \in [0, 1]$ as: $\mathcal{M}^t = (1 - t)\mathcal{M}^z + t\mathcal{M}^1$, where the interpolation is

performed element-wise across the matrix rows. For batched data, this generalizes to:

$$\boldsymbol{\mathcal{M}}^t = (1-t)\boldsymbol{\mathcal{M}}^z + t\boldsymbol{\mathcal{M}}^1. \tag{35}$$

---

**Algorithm 1** SAMPLEANDINTERPOLATE

---

**Input:** Number of samples $N$, feature dimension $D$, microenvironment size $k$, OT plan $\pi_{\epsilon,\lambda}^*$
**Output:** Source ($\boldsymbol{\mathcal{M}}^0$), target ($\boldsymbol{\mathcal{M}}^1$), interpolated ($\boldsymbol{\mathcal{M}}^t$), and noisy ($\boldsymbol{\mathcal{M}}^z$) microenvironments
  1: $(\boldsymbol{\mathcal{M}}^0, \boldsymbol{\mathcal{M}}^1) \leftarrow$ Sample from $K$-Means regions           ▷ Initial microenvironment pairs
  2: $(\boldsymbol{\mathcal{M}}^0, \boldsymbol{\mathcal{M}}^1) \leftarrow \pi_{\epsilon,\lambda}^*(\boldsymbol{\mathcal{M}}^0, \boldsymbol{\mathcal{M}}^1)$           ▷ Resample with OT plan
  3: $\boldsymbol{\mathcal{M}}^z \sim \mathcal{N}(\mathbf{0}, \mathbf{I}_{D+2})^{N \times k}$           ▷ Noisy initial states
  4: $t \sim \mathcal{U}(0, 1)$           ▷ Random interpolation time
  5: $\boldsymbol{\mathcal{M}}^t \leftarrow (1-t)\boldsymbol{\mathcal{M}}^z + t\boldsymbol{\mathcal{M}}^1$           ▷ Linearly interpolated states

---

## E.1 OT Conditional Flow Matching

The OT Conditional Flow Matching (OT-CFM) algorithm consists of a training and a generation phase. During training, the model learns a time-dependent vector field $u_t^\theta$ conditioned on a source microenvironment $\mathcal{M}^0$, which maps a noisy microenvironment $\mathcal{M}^z$—sampled from Gaussian noise—to its corresponding target microenvironment $\mathcal{M}^1$. The supervision is provided via source-target pairs $(\mathcal{M}^0, \mathcal{M}^1)$ obtained through EOT over pooled microenvironment representations. At generation time, the learned vector field is integrated starting from $\mathcal{M}^z$, conditioned on a given source $\mathcal{M}^0$, to generate the predicted target microenvironment $\mathcal{M}^1$.

We optimize the following loss:

$$\mathcal{L}(\boldsymbol{\mathcal{M}}^0, \boldsymbol{\mathcal{M}}^1, \boldsymbol{\mathcal{M}}^t, \boldsymbol{\mathcal{M}}^z; \theta) = \frac{1}{2} \sum_{\substack{\mathcal{M}^0 \in \boldsymbol{\mathcal{M}}^0 \\ \mathcal{M}^z \in \boldsymbol{\mathcal{M}}^z \\ \mathcal{M}^t \in \boldsymbol{\mathcal{M}}^t \\ \mathcal{M}^1 \in \boldsymbol{\mathcal{M}}^1}} \left\| u_t^\theta(\mathcal{M}^t, \mathcal{M}^0) - (\mathcal{M}^1 - \mathcal{M}^z) \right\|_2^2 \tag{36}$$

The full pseudocode for both phases is provided in Algs. 2 and 3.

---

**Algorithm 2** OT CFM — Training

---

**Input:** Number of samples $N$, feature dimension $D$, microenvironment size $k$, OT plan $\pi_{\epsilon,\lambda}^*$, conditional velocity field $u_t^\theta$
**Output:** Trained parameters $\theta$ of $u_t^\theta$
  1: $(\boldsymbol{\mathcal{M}}^0, \boldsymbol{\mathcal{M}}^1, \boldsymbol{\mathcal{M}}^t, \boldsymbol{\mathcal{M}}^z) \leftarrow$ SAMPLEANDINTERPOLATE$(N, D, k, \pi_{\epsilon,\lambda}^*)$    ▷ Microenvironments (Alg. 1)
  2: $\theta \leftarrow \nabla_\theta \mathcal{L}(\boldsymbol{\mathcal{M}}^0, \boldsymbol{\mathcal{M}}^1, \boldsymbol{\mathcal{M}}^t, \boldsymbol{\mathcal{M}}^z; \theta)$           ▷ Compute loss (Eq. (36)) & update parameters $\theta$

---

**Algorithm 3** OT CFM — Generation

---

**Input:** Source microenvironment $\mathcal{M}^0$, learned conditional velocity field $u_t^\theta$
**Output:** Generated microenvironment $\mathcal{M}^1$
  1: $\mathcal{M}^z \sim \mathcal{N}(\mathbf{0}, \mathbf{I}_{D+2})^{1 \times k}$           ▷ Sample noisy sample
  2: $\mathcal{M}^1 \leftarrow \mathcal{M}^z + \int_0^1 u_t^\theta(\mathcal{M}^t, \mathcal{M}^0) \, dt$           ▷ Solve ODE

---

## E.2 OT Gaussian Variational Flow Matching

The OT Gaussian Variational Flow Matching (OT-GVFM) algorithm adopts a variational perspective on Flow Matching. Instead of directly learning a time-dependent conditional velocity field, the model learns a factorized variational posterior $q_t^\theta(\mathcal{M}^1 \mid \mathcal{M}^t, \mathcal{M}^0)$ over target microenvironments $\mathcal{M}^1$, conditioned on an interpolated microenvironment $\mathcal{M}^t$ and a source microenvironment $\mathcal{M}^0$. The predicted velocity field is then computed as the difference between the posterior mean $\mu_t^\theta(\mathcal{M}^t, \mathcal{M}^0)$ and the current state $\mathcal{M}^1$.

The training objective minimizes the discrepancy between ground-truth targets and the predicted posterior means $(\bar{f}_t^\theta, \bar{r}_t^\theta)$:

$$\mathcal{L}(\mathcal{M}^0, \mathcal{M}^1, \mathcal{M}^t; \mu_t^\theta) = \frac{1}{2} \sum_{\substack{\mathcal{M}^0 \in \mathcal{M}^0 \\ \mathcal{M}^t \in \mathcal{M}^t \\ \mathcal{M}^1 \in \mathcal{M}^1}} \sum_{\substack{(c_1, x_1) \in \mathcal{M}^1 \\ (\bar{f}_t^\theta, \bar{r}_t^\theta) \in \mu_t^\theta(\mathcal{M}^t, \mathcal{M}^0)}} \left( \|c_1 - \bar{f}_t^\theta\|_2^2 + \|x_1 - \bar{r}_t^\theta\|_2^2 \right). \quad (37)$$

At generation time, trajectories are generated by integrating the vector field $\mu_t^\theta(\mathcal{M}^t, \mathcal{M}^0) - \mathcal{M}^t$, starting from a noise-sampled microenvironment $\mathcal{M}^z$ and conditioned on a given source $\mathcal{M}^0$. To ensure stability near $t = 1$, the vector field is scaled by a time-dependent denominator, as in prior VFM formulations.

The pseudocode for both phases is provided in Algs. 4 and 5.

---

**Algorithm 4** OT-GVFM — Training

---

**Input:** Number of samples $N$, feature dimension $D$, microenvironment size $k$, OT plan $\pi_{\epsilon,\lambda}^*$, source-conditioned posterior mean predictor $\mu_t^\theta$
**Output:** Trained parameters $\theta$ of $\mu_t^\theta$
1: $(\mathcal{M}^0, \mathcal{M}^1, \mathcal{M}^t, \mathcal{M}^z) \leftarrow \text{SAMPLEANDINTERPOLATE}(N, D, k, \pi_{\epsilon,\lambda}^*)$  ▷ Microenvironments (Alg. 1)
2: $\theta \leftarrow \nabla_\theta \mathcal{L}(\mathcal{M}^0, \mathcal{M}^1, \mathcal{M}^t; \mu_t^\theta)$  ▷ Compute loss (Eq. (37)) & update parameters $\theta$

---

**Algorithm 5** OT-GVFM — Generation

---

**Input:** Source microenvironment $\mathcal{M}^0$, learned source-conditioned posterior mean predictor $\mu_t^\theta$
**Output:** Generated microenvironment $\mathcal{M}^1$
1: $\mathcal{M}^z \sim \mathcal{N}(\mathbf{0}, \mathbf{I}_{D+2})^{1 \times k}$  ▷ Sample noisy sample
2: $\mathcal{M}^1 \leftarrow \mathcal{M}^z + \int_0^1 \frac{\mu_t^\theta(\mathcal{M}^t, \mathcal{M}^0) - \mathcal{M}^t}{1 - t + \epsilon} dt$  ▷ Solve ODE

---

### E.3 NicheFlow: OT Gaussian-Laplace Variational Flow Matching

NicheFlow extends OT-GVFM by modifying the variational posterior: it assumes a Gaussian distribution for gene expression features and a Laplace distribution for spatial coordinates. This change leads to a hybrid loss that combines an $L^2$ loss on gene expression and an $L^1$ loss on spatial locations:

$$\mathcal{L}(\mathcal{M}^0, \mathcal{M}^1, \mathcal{M}^t; \theta) = \sum_{\substack{\mathcal{M}^0 \in \mathcal{M}^0 \\ \mathcal{M}^t \in \mathcal{M}^t \\ \mathcal{M}^1 \in \mathcal{M}^1}} \sum_{\substack{(c_1, x_1) \in \mathcal{M}^1 \\ (\bar{f}_t^\theta, \bar{r}_t^\theta) \in \mu_t^\theta(\mathcal{M}^t, \mathcal{M}^0)}} \left( \|c_1 - \bar{f}_t^\theta\|_1 + \frac{1}{2}\|x_1 - \bar{r}_t^\theta\|_2^2 \right) \quad (38)$$

At generation time, the model integrates the velocity field defined by the difference between the predicted mean $\mu_t^\theta(\mathcal{M}^t, \mathcal{M}^0)$ and the current state $\mathcal{M}^1$, starting from noise and conditioned on the source $\mathcal{M}^0$, identical to the OT-GVFM procedure.

## F  Experimental setup

### F.1  Dataset description

We evaluate our model on three publicly available, time-resolved spatial transcriptomics datasets spanning development and aging processes. Each dataset provides single-cell resolution profiles with matched spatial coordinates and curated cell type annotations. A detailed summary of the organism, tissue, number of time points, cell types, total number of cells, and acquisition technology for each dataset is provided in Table 9.

### F.2  Dataset and microenvironments preprocessing

**Dataset preprocessing.**  All datasets used in our study underwent a preprocessing procedure appropriate for spatial transcriptomic analysis, involving total count normalization, logarithmic

Table 9: Overview of the time-resolved spatial transcriptomics datasets used in our experiments. MED: Mouse Embryonic Development, ABD: Axolotl Brain Development, MBA: Mouse Brain Aging. Each dataset varies in organism, tissue type, number of timepoints, cell types, total number of cells, and spatial transcriptomics technology.

| Dataset | Organism | Tissue | Timepoints | Cell Types | Cells | Technology |
|---------|----------|--------|------------|------------|-------|------------|
| MED | Mouse | Whole embryo | 3 | 24 | 54k | Stereo-seq |
| ABD | Axolotl | Brain | 6 | 33 | 36k | Stereo-seq |
| MBA | Mouse | Brain | 20 | 18 | 1.5M | MERFISH |

transformation, and principal component analysis (PCA). For the mouse embryonic development [20] and axolotl brain development [42] datasets, total count normalization, and logarithmic transformation had already been applied; we, therefore, performed PCA ourselves on the transformed data. For the mouse brain aging dataset [43], we applied all three steps: we first normalized raw count matrices so that each cell had the same total expression, followed by a natural logarithm transformation of the form $\log(X + 1)$ to stabilize variance and mitigate the influence of large values. We then computed PCA on the log-transformed data.

To reduce computational overhead due to high cell counts in the aging dataset, we subsampled the data by a factor of 0.2.

Finally, we standardized PCA components across all cells to ensure consistent scaling across time points. Spatial coordinates were standardized independently for each time point by subtracting the per-time-point mean and dividing by the standard deviation. This standardization preserves the relative spatial configuration while accounting for scale and position differences over developmental time.

**Microenvironments preprocessing.** To facilitate efficient training and enable consistent microenvironment construction, we precompute all fixed-radius neighborhoods for each dataset using a radius $r$ chosen based on spatial resolution. To reduce variability in the number of neighbors and improve batching efficiency, we fix the number of nodes per microenvironment to the most frequent neighbor count observed within each slide. This standardization ensures structural comparability across microenvironments while significantly reducing computational overhead during training, as costly radius or nearest-neighbor queries are avoided at runtime.

### F.3 Quantitative evaluation metrics

We assess spatial fidelity using two complementary asymmetric distance measures. The *point-to-shape distance* (PSD) captures how much predicted cell positions diverge from the actual tissue layout, computed as the average squared distance from each simulated point to its nearest neighbor in the ground truth. Conversely, the *shape-to-point distance* (SPD) quantifies how comprehensively the predicted distribution spans the target structure by averaging the squared distance from each ground truth point to its closest generated counterpart.

Let $\mathcal{G}^t$ denote the set of generated coordinates and $\mathcal{R}^t$ the set of ground truth coordinates at time $t$. Define $\text{NN}^t_{\text{ref}}(c_i)$ as the nearest neighbor of $c_i \in \mathcal{G}^t$ in $\mathcal{R}^t$, and $\text{NN}^t_{\text{gen}}(c_i)$ as the nearest neighbor of $c_j \in \mathcal{R}^t$ in $\mathcal{G}^t$. Then, the two metrics are given by:

$$\text{PSD} = \frac{1}{|\mathcal{G}|} \sum_{\mathcal{G}^t \in \mathcal{G}} \sum_{c_i \in \mathcal{G}^t} \|c_i - \text{NN}^t_{\text{ref}}(c_i)\|_2^2, \tag{39}$$

$$\text{SPD} = \frac{1}{|\mathcal{R}|} \sum_{\mathcal{R}^t \in \mathcal{R}} \sum_{c_j \in \mathcal{R}^t} \|c_j - \text{NN}^t_{\text{gen}}(c_j)\|_2^2. \tag{40}$$

where $\mathcal{G} := \cup_{t \in \mathcal{T}} \mathcal{G}^t$ and $\mathcal{R} := \cup_{t \in \mathcal{T}} \mathcal{R}^t$.

### F.4 Cell-type classification for evaluation

To evaluate cell-type fidelity of generated microenvironments, we train a supervised classifier to assign cell type labels based on gene expression profiles. We apply the same preprocessing steps

used for training our generative models: total count normalization, log-transformation, and PCA (see App. F.2). The resulting low-dimensional embeddings are used as input features for a simple multilayer perceptron (MLP), trained to predict discrete cell type labels.

The classifier consists of a two-layer feedforward network with ReLU activations and a final linear projection to the number of cell types. It is trained using cross-entropy loss and optimized with the AdamW optimizer. We report performance using the weighted F1-score.

We obtain strong classification results across all datasets. On the mouse embryonic development dataset, the classifier achieves a weighted F1-score of 0.85; on axolotl brain development, 0.80; and on the aging dataset, 0.97. These results correlate with the number of input genes and the variance retained in the PCA-reduced space. The aging dataset contains only 300 genes, and 50 principal components explain sufficient variance to accurately distinguish most cell types. In contrast, the embryonic development dataset contains approximately 2,000 genes, and the axolotl brain development dataset includes over 12,700 genes, making classification more challenging due to higher gene expression variability.

### F.5 Discretized microenvironments

To ensure consistent and reproducible evaluation across methods and datasets, we construct a fixed set of evaluation microenvironments by discretizing the spatial domain of each tissue section. For each time point, we define a regular 2D grid over the tissue and select the closest cell to each grid point as the centroid of a microenvironment. Each centroid is then used to construct a fixed-radius neighborhood, following the microenvironment definition in Section 4.1. This procedure ensures full spatial coverage by verifying that every cell belongs to at least one microenvironment.

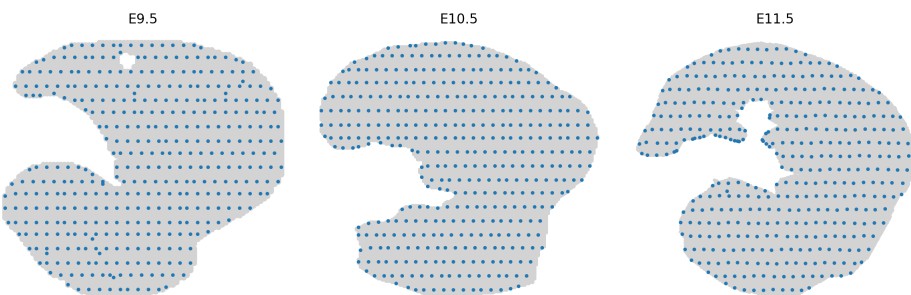

Figure 16: Discretized grid of microenvironments for the mouse embryonic development dataset. Each blue point denotes a centroid around which a microenvironment is constructed. To ensure consistent coverage across tissue sections, additional centroids are randomly sampled such that each slide contains the same number of microenvironments.

Figure 16 illustrates this discretization for the mouse embryonic development dataset. Each blue dot corresponds to a selected centroid. In cases where the number of grid-based centroids falls below a target threshold, additional centroids are randomly sampled to match a fixed total count per slide. This augmentation ensures that all slides contribute equally to the evaluation and prevents bias from sparse regions.

We apply the same discretization procedure to all three datasets used in our experiments: mouse embryogenesis, axolotl brain development, and mouse brain aging. By standardizing the evaluation regions spatially and deterministically, we eliminate the need for stochastic region sampling during evaluation, which would otherwise lead to nondeterministic and irreproducible results.

### F.6 Microenvironment Transformer

To model the spatiotemporal evolution of local cellular neighborhoods, we design a permutation-equivariant transformer-based architecture tailored to structured point cloud data. Our Microenvironment Transformer processes local microenvironments—sets of cells with gene expression features and spatial coordinates—and predicts time-dependent outputs such as velocity fields or future states.

The model operates on a source $\mathcal{M}^0$ and noisy $\mathcal{M}^z$ microenvironments with per-cell features $x_i \in \mathbb{R}^D$ and 2D coordinates $c_i \in \mathbb{R}^2$. The architecture consists of the following components:

1. **Input Embeddings:**
   (a) **Feature Embedding:** A linear transformation is applied to the input features $x_i$ of each cell.
   (b) **Coordinate Embedding:** Spatial coordinates $c_i$ are linearly projected and concatenated to the feature embedding.
   (c) **Time Embedding:** For the noisy microenvironment only, time $t \in [0, 1]$ is encoded using sinusoidal functions $\cos(\omega t)$ and $\sin(\omega t)$, followed by a linear projection and concatenation with the input embedding.

2. **Transformer Encoder (Source Microenvironment):**
   (a) **Self-Attention:** A stack of transformer encoder blocks with multi-head self-attention processes the embedded source microenvironment.
   (b) **No Time Embedding:** Time information is *not* provided to the encoder, as it encodes the source $\mathcal{M}^0$.
   (c) **Residual Feedforward:** Each block contains a feedforward subnetwork with LeakyReLU activation and a residual connection.
   (d) **Layer Normalization:** Applied after both the attention and feedforward layers.
   (e) **Masking:** Binary masks are used to ignore padding in variable-length microenvironments.

3. **Transformer Decoder (Noisy Microenvironment):**
   (a) **Time Embedding:** Temporal context is injected into the decoder by embedding the time $t$ and concatenating it to the target point embedding.
   (b) **Cross-Attention:** Decoder layers apply cross-attention between the noisy microenvironment and the encoded source microenvironment.
   (c) **Self-Attention and Feedforward:** Each decoder block includes standard self-attention and residual feedforward layers.
   (d) **Layer Normalization and Masking:** As with the encoder, normalization, and masking are applied throughout.

4. **Final Output Projection:**
   (a) **Prediction Head:** A linear layer maps the decoder outputs to the desired dimensionality.

This architecture allows for flexible and expressive modeling of temporal dynamics in cellular point clouds. By encoding only the source and decoding the temporally conditioned target, the model supports variational and flow-based training objectives with explicit temporal conditioning.

### F.7  Hyperparameters and Computational Costs

**Model hyperparameters.**    For all experiments, we use the same configuration for the Microenvironment Transformer architecture. The full set of hyperparameters is as follows:

- **Input feature dimension:** 50 PCA-based gene expression features concatenated with a one-hot encoding of the time-point, resulting in a total dimensionality of $50 + |\mathcal{T}|$, where $|\mathcal{T}|$ is the number of slides (timepoints) in the dataset.
- **Input coordinate dimension:** 2
- **Embedding dimension:** 128
- **MLP hidden dimension:** 256
- **Number of attention heads:** 4
- **Number of encoder layers:** 2
- **Number of decoder layers:** 2
- **Dropout rate:** 0.1
- **Output dimension:** 52 (gene expressions features + coordinates)

**OT and mini-batching.** To ensure spatial diversity and computational tractability, we uniformly sample 256 source–target microenvironment pairs from the $K$ spatial clusters obtained via $K$-Means. We then compute the entropic OT plan between these sampled pairs and resample 64 source-target pairs from this plan to define a single training instance. During training, we process 16 such instances per batch.

**Optimization.** All models are trained using the AdamW optimizer with a learning rate of $2 \cdot 10^{-4}$ and a weight decay of $1 \cdot 10^{-5}$. We train each model until convergence.

**Computational cost.** All models were trained on a single NVIDIA GeForce GTX 1080 Ti GPU with 11GB of memory. Depending on the dataset and training objective (e.g., CFM or VFM), training takes approximately 12–16 hours per model.

### F.8 Comparison with moscot

Here, we describe how we conduct the comparisons with moscot, as shown in Fig. 3, Fig. 4, and Fig. 5. For both NicheFlow and moscot, we select a source microenvironment that we want to track over time. In the case of NicheFlow, this corresponds to an aggregate of point clouds. For moscot, it refers to a group of single cells spatially located within the region of interest. The same set of cells is used for both methods.

**NicheFlow.** To generate contour plots over the spatial slide, we push forward the selected region and assign the generated points to their nearest real neighbors based on spatial coordinates. We then compute a probability value for each real position by normalizing the number of assigned generated points. In other words, the more generated points that are close to a given real point, the higher the probability assigned to that location in the contour plot. Cell type proportions are computed as the aggregated frequencies of the generated cell types across the slide. Each plot considers 10 independent generation runs from the same niche.

**moscot.** This baseline is not a generative model, but rather a standard discrete OT framework using a Fused Gromov-Wasserstein cost. As such, it does not generate new features or coordinates. Instead, it outputs a transition matrix that assigns matching probabilities between each source slide cell and each target slide cell. We use these transition probabilities to compute contour plots over the target slide and to aggregate cell type probabilities for the bar plots. For the latter, moscot provides a custom method called `cell_transition()`.

For the Appendix figures Fig. 4,Fig. 7, Fig. 8, and Fig. 9, we propagate the initial source region across multiple time steps. In NicheFlow, this is achieved by using the simulated point cloud from step $t$ to predict the next state at $t + 1$, and then feeding this output as the input for the following trajectory step from $t + 1$ to $t + 2$, and so on. Ground truth points are not used as intermediate sources during this process. For moscot, pushing the source across time points can be automatically done by setting non-subsequent time points in the `cell_transition()` function.

