# OpenReview forum: "Modeling Microenvironment Trajectories on Spatial Transcriptomics with NicheFlow"
_NeurIPS.cc/2025/Conference — NeurIPS 2025 poster_

### Official Review · Reviewer_yBc1 · 2025-06-13

**Clarity:** 1
**Significance:** 2
**Originality:** 3
**Rating:** 3
**Confidence:** 3

**Summary:**

In this manuscript the authors propose a flow matching framework to model the temporal evolution of spatially resolved microenvironments from time-resolved spatial transcriptomics data.

They develop a method denoted as NicheFlow and apply it to several spatiotemporal datasets involving embryonic and brain development.

**Questions:**

P2 Lines 39-44 Are the methods mentioned here applied to spatio-temporal datasets, or just to spatial datasets ?

P2 Line 61: independently optimizes.... why is it important to perform this optimization independently ?

P2 Line 75: in modeling cellular data [25-28]. How are references [25-27] relevant to this task ?

P3 L90 identify likely descendants.. sentence is unclear.

P4 L150 NicheFlow predicts the corresponding tissue structure... Why is this interesting from a biological / scientific perspective?

P5 top: it is a bit unclear what dynamics can be learned by looking at only two consecutive time points. It is also unclear if
the two time points one measures exactly the same cells (did the cells not move / where they tracked / no cell divisions occurred) ?

P5 Eq. 12 - it seems a bit simplistic to simply average the gene expression of neighborhood cells? What are typical processes
in the dynamical evolution of niches and are these easily captured by such averaging ?

P8 Which reference is SPFlow ?

P12 are refs 34 and 38 the same?

**Ethical Concerns:**

["NO or VERY MINOR ethics concerns only"]

**Final Justification:**

Based on all referee reports, and the author replies, in my view, in its current form (or with small revisions) this manuscript is much more suitable for publication in a computational biology venue. As I am not an expert in this field, I increased my rating to borderline reject.

**Limitations:**

The authors do not discuss the limitations of their approach in the main text, but they do mention several of them in the supplementary, appendix B.

**Quality:**

2

**Strengths And Weaknesses:**

The main strength of the manuscript is the development of NicheFlow, a method to model the temporal evolution not of individual cells,
but of niches.

That said, the manuscript has several major weaknesses, as I outline below.

Clarity of Exposition and Fit to NeurIPS: The manuscript is not very accessible to a non-expert, and not sufficiently self contained.
To me it seems that the target audience of this manuscript are researchers in computational biology.
As such, in my opinion, the manuscript would be more suitable for publication in such venues rather than in a ML conference.
Also, the organization of the paper could be improved. For example, section 3 with a background on flow matching is a bit out of context
given that the precise formulation of what is measured and what are the tasks of interest appear only later.

Quality and Significance: While I am not an expert in this field, to the best of my understanding, the reason for collecting
spatial-temporal biological datasets is to allow to gain novel insights into temporal biological processes, such as development
and disease progression.
In this manuscript, the authors present their method of modeling the evolution of Niches.
However, there is no sufficient biological background, no concrete biological questions considered.
The manuscript would benefit if it included an in-depth example where new insight is gained using the proposed model
on a concrete biological problem of major importance.

Originality and Novelty: The model proposed is to the best of my knowledge novel.

The evaluation section would benefit from a specific example where the authors would illustrate how NicheFlow allows
gaining novel insight into some dynamical biological process.

Additional comments:
P2 L56 prediction of spatial coordinates and gene expression... It may be good to provide some concrete biological examples
where this would be useful / important.

P2 L66 NicheFlow enables principled learning of dynamics... Again, it may be good present a concrete biological example
with specific scientific findings enabled by NicheFlow.

---

> ### Author Rebuttal · Authors · 2025-07-30
>
> We would like to thank reviewer yBc1 for reviewing our paper. We value the criticism and feedback as an opportunity to improve the clarity of our submission, and hope that our answers will contribute to a more positive evaluation of our work.
>
> > Fit to NeurIPS
>
> We respectfully disagree with the assessment that our manuscript does not fit NeurIPS due to its application domain. Applications of ML to computational biology have become increasingly representative at major venues, including papers on spatial transcriptomics accepted at NeurIPS, ICML and ICLR (some examples: [1-3]). Moreover, NeurIPS explicitly lists life sciences as one of the key areas under its Call for Papers, reinforcing its commitment to applied domains.
>
> Our work does not simply apply an existing algorithm but introduces a new methodological perspective on generative Flow Matching (FM), motivated by an applied setting. Specifically, we extend Variational Flow Matching (VFM) to approximate a source-conditioned Entropic Optimal Transport (EOT) model and introduce a factorized posterior to model structured data such as attributed point clouds.
>
> We describe these novel contributions in the introduction (L51-68).
>
> > Background organization
>
> Our work builds on different aspects of FM that have not previously been explored together: EOT couplings across incomparable spaces, and a variational perspective on the generated space.
>
> We believe it is important to formally introduce these concepts in a general mathematical setting. This structure allows us to tie the ideas together in a principled way later in the paper, and moves from general background to model desiderata and finally to our point-cloud-based formulation.
>
> We introduce the mathematical formulation of spatial niches after the general background, as presenting it earlier would require framing the entire background in point cloud terms, which is specific to NicheFlow and better suited to the contribution sections.
>
> We propose adding a new section before subsection 3.1 to introduce how biological trajectories are typically modeled using OT.
>
> > Biological introduction and scope
>
> In the introduction, we highlight the importance of **considering cells in their environments**, describe static and dynamic spatial transcriptomics, identify key gaps in existing models, and state our research question explicitly (L45–46). This establishes both the biological context and our main goal. We frame the problem as a **general trajectory modeling task** over microenvironments, aiming to develop an algorithm applicable across **diverse biological systems with spatiotemporal axes**.
>
> This broad scope motivates our submission to NeurIPS and is reflected by our experiments with generation and biological trajectory inference on development and ageing using widely available data from benchmarks [4].
>
> We will clarify the relevance of our task in the final version with examples from tumour progression and brain development.
>
> > Biological analysis and findings
>
> Sec. 5.2 and the Appendix present predicted dynamics and developmental hypotheses enabled by our model:
>
> * Fig. 3, right panel: we predict that immature cell groups emigrate in the upper cranial region **and** that they differentiate into head mesenchymal cells, simultaneously. This demonstrates our model's combined spatial and transcriptomic insight, unlike standard single-cell RNA-seq methods.
> * App. Fig. 11: We draw similar knowledge studying how facial neural crest cells evolve into jaw and teeth structures, this time predicting across multiple timepoints.
> * App. Fig. 13-14: We show dynamic formation of anatomical regions during the axolotl brain development, capturing spatial and compositional changes.
> * App. Fig. 15: We demonstrate our model predicts a decrease in stemness in the left dorsal pallium.
>
> > L56 - gene expression and coordinates
>
> We refer to our previous answer for the biological use cases presented in our paper. Predicting both the location and gene expression of cells is important, as it provides an understanding of both **tissue composition** and **organization**. Different from standard scRNA-seq that only specifies how gene expression evolves across cell types, here we study how earlier cell states evolve and where they are positioned anatomically, adding a new layer of information to standard trajectory inference (e.g., prediction of both fate and migration of immature cells in Fig. 3, right). Predicting spatial and state transitions is also clinically relevant, where future model applications could, for example, study metastatic progression in cancer over time.
>
> > P2 L39-44: referenced methods
>
> The models cited in these lines are applied to spatial transcriptomics and infer trajectories at the single-cell level. In this paragraph, we highlight how most current approaches do not consider the trajectories of entire microenvironments of cells, even though the dynamics of tissue restructuring during temporal processes are strongly determined by the coordinated evolution of cells.
>
> > P2 Line 61: optimization
>
> We thank the reviewer for pointing this out. Our model **jointly** trains on both spatial coordinates and gene expression features, but the loss function is factorized into two components, one for spatial reconstruction and another for biological fidelity, each with its likelihood and distribution (Laplace for coordinates, Gaussian for gene expression).
>
> We will revise the sentence as follows:
>
> > "...that jointly trains on spatial and cell state dynamics using a factorized loss, modeling spatial reconstruction and biological fidelity with tailored distributional assumptions."
>
> > P2 Line 75
>
> All these references point to Flow Matching papers that were applied to learning cellular trajectory from earlier to later time points, which is a common benchmark in OT-based generative models. All these methods predict how cell states evolve into one another, but not how they organize in the anatomical space (L75-76).
>
> > P3 Line 90: descendants
>
> Exact-solution OT, as implemented in moscot estimates a transition matrix containing the probability of each target cell to be a descendant of each ancestor cell. We propose to make the sentence clearer as follows:
>
> > ... use discrete optimal transport to link cells across time and infer the evolution of cell states from spatially-resolved gene expression.
>
> > P4 L150 tissue structure
>
> Predicting tissue structure from a source microenvironment is biologically significant as tissue organization reflects biological function, with spatial changes indicating migration, proliferation, or reorganization during development or disease.
>
> Additionally, spatial reorganization has been crucial for critical phenomena such as immune cell infiltration and tumour microenvironment remodeling in cancer [5], fibrosis progression and matrix remodeling [6].
>
> In this context, our model provides a generative tool to investigate dynamics at the neighborhood level, offering insights that go beyond single-cell trajectories by modeling how spatially localized groups of cells evolve as structured units.
>
> > P5 top: type of dynamics
>
> In scRNA-seq and its spatial variant, each cell is destroyed during sequencing. Thus, each slide or sample is an independent measurement. This is a standard setting in single-cell transcriptomics and a key reason why OT has emerged as a framework for trajectory inference across timepoints. Once an appropriate optimality criterion is defined, OT can be used to infer spatio-temporal dynamics on independent snapshots. In other words, computational methods are used to reconstruct a trajectory from successive measurements, even though longitudinal tracking of individual cells is not experimentally possible. In our work, each whole slide, as defined in Eq. (8), contains a different number of cells. We will add a sentence to make this aspect clearer in Section 4.1.
>
> > P5 eq 12: Averaging
>
> We act on the transcriptional and spatial means to match point clouds across batches. We highlight that this defines a criterion to pair up point clouds across batches, but the dynamical model still learns to flow to all points of a target point cloud, attending all the points and coordinates of a source microenvironment. We agree that using the mean in combination with very large point clouds may mask the signal, but in a more localized setting (see App. Fig. 7), other established studies have demonstrated that cellular microenvironments are well-approximated by the statistics of a Gaussian around a central cell [7]. This supports our matching criterion.
>
> > SPFlow
>
> SPFlow is our internal baseline, introduced in Sec. 5.1.3, which models trajectories at the single-cell level rather than the microenvironment level. While not from a published paper, it was designed to highlight the benefits of modeling point cloud dynamics over individual-cell dynamics. We will add a clarifying sentence in Sec. 5.1.3 stating that SPFlow is introduced in this work.
>
> > refs 34 and 38?
>
> Thank you for pointing this out. We will correct the typo in our revised version.
>
> > Limitations
>
> Discussed in Appendix B and referenced at L227.
>
> **Ref.**
>
> [1] Xie et al., "Spatially resolved gene expression prediction from histology images via bi-modal contrastive learning", NeurIPS 2023
>
> [2] Oh et al., "Global Context-aware Representation Learning for Spatially Resolved Transcriptomics", ICML 2025
>
> [3] Zhu et al., "Diffusion Generative Modeling for Spatially Resolved Gene Expression Inference from Histology Images", ICML 2023
>
> [4] Halmos et al., "DeST-OT: Alignment of Spatiotemporal Transcriptomics Data", Cell Sys. 2025
>
> [5] Zhao et al., "Spatial transcriptomics at subspot resolution with BayesSpace", Nat. Biotech. 2021
>
> [6] Raredon et al., "Single-cell connectomic analysis of adult mammalian lungs", Science Advances 2021
>
> [7] Haviv et al., "The covariance environment defines cellular niches for spatial inference", Nat. Biotechnology 2025

---

> > ### Comment · Reviewer_yBc1 · 2025-08-03
> >
> > I thanks the authors for their responses. I would like to clarify that I am well aware of multiple computational-biology papers published in ML venues. The point raised in my review is that, as this manuscript was written, in my opinion it is more suitable for publication in other venues. Indeed in their replies the authors clarify their ML contribution.
> > Beyond this, I have no further questions at this point.

---

> > > ### Author Response · Authors · 2025-08-03
> > > **Thank you for reading our rebuttal**
> > >
> > > Dear Reviewer yBc1,
> > >
> > > Thank you very much for reading our rebuttal and providing additional feedback. We apologise if we misinterpreted parts of your criticism, but we are glad our responses helped contextualize our ML contribution.
> > >
> > > In our rebuttal, alongside our ML contribution, we addressed how our work can favour biological analysis with examples from the paper, together with additional biological explanations on task relevance and clarifications on the paper’s structure. These were points of criticism by the reviewer.
> > >
> > > In light of this, we would be grateful if the reviewer could kindly let us know whether this additional context contributed to a more favourable overall assessment of the work.
> > >
> > > Thank you again for dedicating time to reviewing our paper and reading our rebuttal.
> > >
> > > Best regards,
> > >
> > > The Authors

---

> > > > ### Comment · Reviewer_yBc1 · 2025-08-05
> > > >
> > > > Dear authors,
> > > > In light of your responses I have slightly increased my score. I do believe that the manuscript needs to undergo significant changes to highlight the novelties on the ML side and make it more accessible to the wide audience of the conference.

---

> > > > > ### Author Response · Authors · 2025-08-06
> > > > > **Thank you for revieweing our work**
> > > > >
> > > > > Dear Reviewer yBc1,
> > > > >
> > > > > Thank you very much for providing additional clarifications and increasing your score. As outlined in our rebuttal text, we will try to make our paper more accessible to a wider audience by:
> > > > >
> > > > > * Further elaborating in the introduction on the biological relevance of trajectory inference in space, with application examples like tumour progression or tissue restructuring in development.
> > > > > * Adding an intuitive introduction on the biological task of developmental trajectory inference to highlight the significance of our biological results.
> > > > > * Including a subsection at the beginning of the background, formalizing how Flow Matching connects to the trajectory inference task in time-resolved single-cell data.
> > > > >
> > > > > We will also highlight our ML novelty more explicitly in the introduction and methods, proposing our model as the first to connect entropic OT to a variational perspective on Flow Matching for point cloud trajectories.
> > > > >
> > > > > We believe our current paper structure can accommodate these additions smoothly.
> > > > >
> > > > > Thank you again for the time dedicated to our work and for your constructive feedback.
> > > > >
> > > > > The Authors.

---

### Official Review · Reviewer_fydQ · 2025-06-25

**Clarity:** 3
**Significance:** 3
**Originality:** 3
**Rating:** 5
**Confidence:** 4

**Summary:**

This paper introduces NichFLOW, a flow-based generative model designed for modeling time-series spatial transcriptomics data. The authors propose incorporating microenvironments, variational flow matching, and conditional generative modeling to learn correspondences between spatially adjacent cell slices in terms of spatial position and gene expression. The method is validated on several datasets. Overall, the idea is interesting, as while static optimal transport (OT) methods have been widely used in this context, flow matching-based methods are less considered.

**Questions:**

1.  Can the authors clarify whether NichFLOW can perform interpolation (e.g., interpolate time point 1 from time points 0 and 2) or similar tasks? If so, how were these capabilities evaluated, and could additional experiments demonstrate these advantages?


2.  Could the authors provide insights into the key differences or advantages of NichFLOW over other methods to better position its contribution?

3. The evaluation metrics focus on cell type performance but lack gene-level assessments, such as W1 or W2 distances for gene expression vectors, spatial positions, or combined metrics.  Would adding such metrics provide a clearer understanding of the method’s generative capabilities? Additionally, why were neural ODE-based methods like TrajectoryNet, MIOFlow, DeepRUOT not included as baselines?

4. The method does not account for cell growth, death, which may impact the accuracy of dynamics, especially in developmental datasets. Why was an unbalanced OT (UOT) or unbalanced Fused Gromov-Wasserstein OT framework, such as those in MOSCOT, not considered to model these effects, and could its inclusion improve the method’s applicability to such datasets?

Overall, I view this work positively, and if the authors can address the weaknesses outlined, incorporate additional experiments, baselines (e.g., TrajectoryNet, MIOFlow, DeepRUOT), gene-level metrics (e.g., W1, W2 distances), and comprehensive discussions, I will consider raising my score.

**Ethical Concerns:**

["NO or VERY MINOR ethics concerns only"]

**Final Justification:**

I appreciate the authors' thorough and thoughtful response. The further discussions on existing literatures would enhance readers’ understanding of the proposed approaches and their broader implications. I believe the authors have effectively addressed my concerns, and I think this work is an impactful contribution. So I am very happy to raise my score to 5.

**Limitations:**

yes

**Quality:**

3

**Strengths And Weaknesses:**

Strengths:

1. The paper is generally well-written, includes extensive experiments, and presents a comprehensive study.

2. The work builds on some concepts from prior studies, such as GENOT and VFM, and introduces the concept of microenvironments to handle time-series spatial transcriptomics data. The integration of these ideas in this context is new.

3.   Reconstructing dynamics from time-series spatial transcriptomics data is an important problem and has gained increasing attention.

Weakness:

1. The paper employs flow matching as a conditional generative model, which appears limited in capturing the continuous dynamics of cells across time points; however, this can be achieved by many dynamical-based optimal transport (OT) methods. Specifically, the method seems less suited for interpolation tasks, such as predicting data at time point 1 given data at times 0 and 2, unless relying on simple linear interpolation. A major weakness to me is that NichFLOW appears to achieve functionality similar to static OT methods (such as WaddingOT, Moscot, etc). The paper does not clearly show whether NichFLOW can accomplish tasks that static OT methods cannot, limiting its broader applications in modeling complex spatiotemporal cellular dynamics.


2. The paper lacks a comprehensive discussion comparing the proposed method with other works working on the spatio-temporal ST data. Including such a comparison, e.g., with [1,2], could better clarify the similarities and differences between NichFLOW and other methods.


3. While the evaluation includes metrics for cell type prediction, it lacks detailed gene-level assessments, such as Wasserstein-1 (W1) and Wasserstein-2 (W2) distances between generated and true gene expression vectors, spatial positions, or their combined distributions. These metrics could better elucidate the method’s generative performance. Additionally, the paper does not include neural ODE-based methods (e.g., TrajectoryNet [3], DeepRUOT[4]) as baselines for comparison.


4. The method does not account for cell growth or death, as acknowledged in the limitations. Given that some tested datasets involve developmental processes, neglecting cell proliferation effects may lead to inaccurate dynamics. The paper could benefit from exploring unbalanced OT (UOT) frameworks, such as those used in WaddingOT/MOSCOT, which could be a straightforward way to address this issue.

Reference:

[1] DeST-OT: Alignment of Spatiotemporal Transcriptomics Data, Peter Halmos, Xinhao Liu, Julian Gold, Feng Chen, Li Ding, Benjamin J. Raphael

[2] Inferring cell trajectories of spatial transcriptomics via optimal transport analysis, Xunan Shen, Lulu Zuo, Zhongfei Ye, et. al.

[3] TrajectoryNet: A Dynamic Optimal Transport Network for Modeling Cellular Dynamics, Alexander Tong, Jessie Huang, Guy Wolf, David van Dijk, Smita Krishnaswamy

[4] Learning stochastic dynamics from snapshots through regularized unbalanced optimal transport, Zhenyi Zhang, Tiejun Li, Peijie Zhou

---

> ### Author Rebuttal · Authors · 2025-07-30
>
> We thank Reviewer fydQ for their constructive feedback and positive assessment of our method and manuscript. Below, we address the reviewer’s suggestions and remaining concerns point by point.
>
> > Dynamic with conditional model
>
> Our approach uses source-conditioned CFM to approximate a continuous Entropic Optimal Transport (EOT) coupling between source and target distributions [1]. This makes our method a principled predictor for the next time point, **provided the OT assumption holds.**
>
> Unlike previous scRNA-seq methods that evolve individual cells, we model the joint evolution of spatially defined point clouds, treating each microenvironment as a **rigid object**. Since spatial slides can differ in cell density, matching source and target point clouds becomes an **OT problem across incomparable spaces** with different cell counts.
>
> A standard ODE model like the cited ones would evolve each point individually and would be inadequate in the point cloud setting. Instead, we use the continuous EOT coupling of a source-conditioned CFM model to approximate how a source point cloud evolves over time, capturing both compositional and spatial changes in cellular neighborhoods. As shown in **Fig. 13-15** (Appendix), our model generates spatially resolved point clouds from an initial state at $t=0$.
>
>
> > Functional similarity to static OT approaches.
>
> Our method is intrinsically more flexible than moscot and WaddingtonOT, as it provides a generative model approximating a neural Monge map. NicheFlow predicts how a cellular point cloud evolves from $t$ to $t+1$ in both space and gene expression. In contrast, static OT methods only compute the probability that each earlier cell evolves into each later cell, and do not generate later time points.
>
> For example, see Fig. 13-15 in the Appendix, where we predict the evolution of a source spatial region to a target point cloud across time, using the generated neighbourhood at $t+1$ as a source to predict the dynamics until $t+2$. All the colored points are purely generated. In other words, our model can be used for discrete time point interpolation and cell state generation, which cannot be achieved by static OT.
>
> The reason for our comparison with moscot is that it is the most established method performing spatiotemporal trajectory predictions across time, and we believe it is important to show the community how our model represents a promising alternative in an applied setting. With an extra page of results, we will add some results on generative interpolations from the Appendix to the main text.
>
> > Comparison with references.
>
> We thank the reviewer for highlighting these relevant works and will cite them in our revised manuscript. Below, we briefly summarize the key differences between our approach and the suggested models, DeST-OT and SpaTrack.
>
> DeST-OT aligns spatial slides using semi-relaxed optimal transport, incorporating a loss that preserves transcriptomic and spatial proximity between ancestors and descendants. It also introduces a growth vector to estimate the expected number of descendants per source cell.
>
> SpaTrack uses Fused Gromov-Wasserstein OT, balancing transcriptomic and spatial differences based on spatial autocorrelation of features.
>
> Our work is comparable to SpaTrack and DeST-OT in terms of biological settings: they all aim to capture temporal information in time-resolved spatial transcriptomics. However, there are crucial differences:
>
> * Both suggested methods use discrete OT to probabilistically assign each source cell to an existing target cell, similar to moscot. In contrast, our approach parameterizes a continuouos map in the Monge sense (as in [1-4]), predicting how cell distributions (or point clouds, in our case) evolve over time via a flow.
> * Our model trains an approximate coupling in batches, making NicheFlow scalable to large spatial slides and datasets (e.g., the 4.2 million-cell brain ageing dataset we used). Moreover, parameterized deep models also enable future generalization to large treatment studies or cross-species tasks.
> * The main difference is that NicheFlow models the evolution of spatial microenvironments (neighborhoods), not individual cells. It generates how a spatial region changes across slides, while prior methods focus on discovering single-cell descendants from ancestors. By adjusting neighborhood size and shape, NicheFlow enables both localized and large-scale tissue analysis.
>
> In summary, with NicheFlow we propose two new perspectives on spatio-temporal modeling: the use of a continuous flow-based generative model to learn spatial trajectories and the formulation of the trajectory problem at the level of point clouds.
>
> > W1 and 2 metrics
>
> We agree that incorporating gene-level and spatial assessments using Wasserstein distances can offer additional insight into the generative capabilities of the models. In response, we have now computed $\mathcal{W}_1$ and $\mathcal{W}_2$ distances for both spatial coordinates (“Pos.”) and gene expression vectors (“Genes”) for all models presented in Table 1.
>
> To ensure scalability, we compute Wasserstein distances per cell type and report the average across types. Shown below are the best-performing configurations for each model on the mouse embryonic development dataset; additional results will be provided in the appendix due to space constraints.
>
> | Model | Objective | $\mathcal{W}_1$ Pos. | $\mathcal{W}_1$ Genes | $\mathcal{W}_2$ Pos. | $\mathcal{W}_2$ Genes |
> |---|---|---|---|---|---|
> | SPFlow | GLVFM | 0.613 ± 0.001 | 6.457 ± 0.003 | 0.738 ± 0.001 | 5.546 ± 0.108 |
> | RPCFlow | GLVFM | 0.221 ± 0.003 | 6.087 ± 0.002 | 0.365 ± 0.007 | 5.244 ± 0.113 |
> | NicheFlow | GLVFM | **0.212 ± 0.004** | **5.930 ± 0.003** | **0.342 ± 0.007** | **5.031 ± 0.003** |
>
> These results reinforce our findings: NicheFlow with the GLVFM objective outperforms other models across all metrics, including the new Wasserstein evaluations, confirming its ability to generate realistic spatial and transcriptional patterns.
>
> > Proposed baselines: MIOFlow, TrajectoryNet and DeepRUOT?
>
> These methods are designed for single-cell transcriptomic trajectory inference and do not model spatial structure or microenvironment-level dynamics. TrajectoryNet, MIOFlow, and DeepRUOT learn continuous trajectories in gene expression space (or PCA reductions thereof) rather than generating spatially resolved point clouds conditioned on tissue context.
>
> Our work addresses a different problem: modeling how spatial neighborhoods of cells evolve jointly in both space and gene expression composition. Adapting these models to **predict coordinates** cell state simultaneously, as well as generating an increased number of cells in time, would alter their original design and put them at a disadvantage. Intuitively, the lack of spatial information in the prediction prevents the computation of the SPD and PSD spatial metrics on the outputs of the proposed models.
>
> We will clarify these aspects in the revised manuscript and cite the papers.
>
> > The use of unbalanced OT.
>
> While unbalanced OT is a promising direction, integrating it into NicheFlow is non-trivial. Our model tracks the evolution of cell groups as point clouds, not individual cells. Unlike single-cell frameworks, where growth reflects mass conservation, growth at the point cloud level is harder to define. Since our OT formulation models how cell neighborhoods evolve jointly in space, it does not rely on explicit growth modeling or enforce mass conservation at the individual-cell level.
>
> Instead, our model captures changes in cell density by using a conditional generative approach to approximate the EOT coupling, allowing mappings between source and target clouds of differing sizes. The number of sampled cells at each time point defines the noisy cloud size during training, enabling the model to learn spatially resolved density shifts, as proliferation in development or expansion in disease.
>
> > Interpolation capabilities.
>
> Great question. In principle, NicheFlow can interpolate unseen time points. Given a densely sampled developmental process, one could condition generation on a continuous target time and simulate evolution from a source. If the samples sufficiently approximate a continuum, this extends to unseen slides.
>
> Unfortunately, this approach faces key data **limitations:**
>
> - Current spatial datasets are sparsely sampled in time and do not approximate a continuum. Intermediate slides are often **anatomically distinct** from both their predecessors and successors, making interpolation a fully out-of-distribution (OOD) task.
>
> - One could test interpolation using replicates (e.g., training on one embryo and predicting held-out slides from another sampled at the same timepoints), but to our knowledge, such replicated spatiotemporal datasets are not available.
>
> - Crucially, some intermediate slides contain **unique cell types or anatomical structures** not observed at other time points. In spatial transcriptomics, where transcriptional profiles are constrained by spatial context, this can lead to biologically infeasible predictions.
>
> In other words, while interpolation is algorithmically possible, its biological feasibility depends on denser temporal sampling or the availability of replicates, which are currently lacking in existing datasets.
>
> **Ref**
>
> [1] Klein et al. "GENOT: Entropic (Gromov) Wasserstein flow matching with applications to single-cell genomics." Advances in Neural Information Processing Systems 37 (2024): 103897-103944.
>
> [2] Tong et al. "Conditional flow matching: Simulation-free dynamic optimal transport." TMLR (2023).
>
> [3] Atanackovic et al. "Meta flow matching: Integrating vector fields on the wasserstein manifold." ICLR (2024).
>
> [4] Wang et al. "Joint Velocity-Growth Flow Matching for Single-Cell Dynamics Modeling." arXiv preprint arXiv:2505.13413 (2025).

---

> > ### Comment · Reviewer_fydQ · 2025-08-05
> >
> > I appreciate the authors' thorough and thoughtful response. The further discussions on existing literatures would enhance readers’ understanding of the proposed approaches and their broader implications. I believe the authors have effectively addressed my concerns, and I think this work is an impactful contribution. So I am very happy to raise my score to 5.

---

> > > ### Author Response · Authors · 2025-08-05
> > > **Thank you!**
> > >
> > > Dear Reviewer fydQ,
> > >
> > > Thank you very much for reading our rebuttal and for providing valuable feedback. We will incorporate the discussion of prior work, as well as the additional experiment presented in the rebuttal, into the submission.
> > >
> > > We are grateful for the positive feedback and for your kind contribution to improving our work.
> > >
> > > Best regards,
> > >
> > > The authors

---

### Official Review · Reviewer_yXEk · 2025-06-28

**Clarity:** 4
**Significance:** 4
**Originality:** 4
**Rating:** 6
**Confidence:** 5

**Summary:**

This paper introduces NicheFlow, a generative model designed to capture the spatiotemporal evolution of cellular microenvironments in spatial transcriptomics (ST) data. Unlike existing cell-centric approachStes, NicheFlow models local neighborhoods (“niches”) as point clouds of cells, each comprising gene expression and spatial coordinates, and learns their dynamics over time using a combination of: Optimal Transport (define soft couplings between niches across timepoints), Variational Flow Matching (simulate niche trajectories via learned velocity fields), factorized probabilistic model over expression and spatial modalities, and a Transformer-based point cloud architecture. Empirical evaluations on three datasets (mouse embryonic development, axolotl brain development, and mouse brain aging) demonstrate that NicheFlow outperforms both traditional FM-based models and the widely used MOSCOT framework in terms of spatial and semantic accuracy. Qualitative results highlight NicheFlow’s ability to capture known biological differentiation trajectories.

**Questions:**

- Can $\lambda$ be learned or adapted across time or tissue regions? This seems essential given its major role in balancing spatial vs transcriptomic cues.
- How sensitive is the model to the choice of neighborhood radius $r$? Could learned neighborhoods or graph-based neighborhood definitions be more effective?
- Could this model be extended to continuous-time predictions or interpolation across multiple timepoints?

**Ethical Concerns:**

["NO or VERY MINOR ethics concerns only"]

**Final Justification:**

I thank the authors for their complete response. Their responses address my concerns, therefore I would like to increase my ratings from 5 to 6.

**Limitations:**

While NicheFlow models gene expression trajectories over time, it does so directly in the observed (or PCA-projected) expression space. However, gene expression data is known to lie on a lower-dimensional manifold shaped by biological processes like differentiation or zonation. Many successful models in scRNA-seq (e.g., scVI, DPT, diffusion models) explicitly exploit this structure for denoising, interpolation, and trajectory learning. By ignoring the data manifold, NicheFlow may be susceptible to noise and struggle to distinguish technical variability from biological transitions. Incorporating latent space modeling or manifold-aware trajectories could improve robustness and interpretability.

**Quality:**

4

**Strengths And Weaknesses:**

Strengths:
- Shifts from modeling single-cell trajectories to niche-level dynamics, which is more aligned with tissue-scale processes like differentiation, migration, and patterning.
- Uses point cloud representations of microenvironments, which are permutation-invariant, variable in size, and expressive of spatial structure.
- Integrates EOT and source-conditioned VFM, extending prior FM formulations to handle unpaired, noisy, and variable-sized spatiotemporal point clouds.

Weaknesses:
- NicheFlow is trained on local microenvironments only — there’s no mechanism to enforce global tissue-wide coordination, which is often important in developmental processes governed by morphogen gradients or topological constraints.
- The trade-off parameter $\lambda$ (controlling spatial vs gene similarity in the OT cost) is fixed at 0.1. There’s no exploration of $\lambda$ sensitivity or adaptation, which is important for different biological regimes (e.g., morphogenesis vs migration).
- PSD and SPD capture spatial proximity but not structural or topological similarity (e.g., niche boundaries, internal configuration, or inter-cell distances). All metrics are pointwise — no graph or structure-aware evaluation is used.
- While using the microenvironments concept is good, the key part of utilizing this concept is missing -- the cells actually interact with their neighbors. Without considering cell-cell interactions, grouping cells solely based on their spatial location has limited biological significance.

---

> ### Author Rebuttal · Authors · 2025-07-30
>
> We thank Reviewer yXEk for the thoughtful review. We appreciate the positive assessment and the insightful suggestions. Below, we address the remaining questions and points of criticism.
>
> > Global tissue coordination.
>
> While we recognize the value of global coordination, our work focuses on individual functional regions in a tissue, enabling users to study the migratory or compositional development of custom biological areas over time. This aligns with the locality paradigm in graph and point-cloud methods for cell communication, which we extend to the spatiotemporal setting.
>
> In future work, we could incorporate global context by conditioning velocity fields on global niche embeddings from long-range graph or point cloud representation models. This allows biasing the model by the influence of the global cellular context. Still, scaling point-cloud methods like NicheFlow to whole-slide data remains a computational challenge, especially at higher resolutions, and is left for future exploration.
>
> > Tradeoff parameter.
>
> We note that the role of $\lambda$ is discussed and explored throughout the main text and Appendix (L314–323):
>
> - **Sec. 5.2.** In our comparison with moscot, we test two scenarios:
>   * For studying compositional changes in fixed spatial structures, we use a low value ($\lambda=0.1$) to preserve spatial location over time.
>   * For modeling migratory processes (e.g., neural crest to head mesenchyme), we use $\lambda=0.5$, showing that adding cell-state similarity aids fate mapping across space.
>
> - **App. D.3.** The value $\lambda=0.1$ is used for the **point cloud generation** task in Tab. 1. We also provide:
>   * An ablation across $\lambda$ values (Tab. 2),
>   * A visualization of its effect on coupling distance (Fig. 7, ref. in L199).
>
> - **App. D.6.1 and D.6.2.** We use $\lambda=0.5$ for additional biological experiments, such as mapping contained organs over time and inferring anatomical changes during axolotl brain development.
>
> > Pointwise metrics
>
> PSD and SPD are standard metrics in point cloud generation and correspond to the two directions of the Chamfer distance, quantifying fidelity and coverage [1]. Importantly, we compute them over many aggregated microenvironments to reconstruct the entire tissue, so they indirectly reflect global morphology.
>
> To directly address this concern, we include Gromov–Wasserstein (GW) and Fused GW (FGW) distances [2, 3] between generated and real point clouds to compare different training paradigms for NicheFlow. These metrics compare intra-point-cloud pairwise distance matrices between source and target, capturing internal structure (GW) and gene expression similarity (FGW). Values are averaged over 5 generation runs.
>
> As computing GW/FGW on entire slides is computationally infeasible due to the number of points, we evaluate them at the generated niche level, where structural comparisons are tractable. A direct comparison with SPFlow or RPCFlow is not appropriate: SPFlow predicts single points and lacks structure, while RPCFlow produces random point clouds that vary across runs.
>
> #### MED
>
> | Mod. | Obj. | GW $(10^2)$ | FGW |
> |---|---|---|---|
> | NicheFlow | CFM | 0.315 ± 0.003 | 3.273 ± 0.003 |
> | NicheFlow | GVFM | 0.463 ± 0.004 | 3.167 ± 0.007 |
> | NicheFlow | GLVFM | **0.224 ± 0.001** | **3.147 ± 0.007** |
>
> #### ABD
>
> | Mod. | Obj.| GW $(10^2)$ | FGW |
> |---|---|---|---|
> | NicheFlow | CFM | 0.783 ± 0.002 | 3.543 ± 0.004 |
> | NicheFlow | GVFM | 0.797 ± 0.006 | **3.414 ± 0.003** |
> | NicheFlow | GLVFM | **0.720 ± 0.004** | 3.420 ± 0.003 |
>
> #### MBA
>
> | Mod. | Obj. | GW $(10^2)$ | FGW |
> |---|---|---|---|
> | NicheFlow | CFM | 0.315 ± 0.001 | 3.531 ± 0.000 |
> | NicheFlow | GVFM | 0.339 ± 0.000 | 3.387 ± 0.000 |
> | NicheFlow | GLVFM | **0.262 ± 0.000** | **3.367 ± 0.000** |
>
> NicheFlow with the GLVFM objective consistently achieves **the lowest or competitive GW and FGW distances**, confirming the value of our novel mixed-factorized posterior at preserving geometric constraints.
>
> > Cellular interactions
>
> While modeling cell-cell interactions is important, spatial proximity alone can reveal meaningful biology, e.g., how brain regions evolve with disease and treatment, or how processes like fibrosis and inflammation alter tissue structure beyond direct cell-cell signaling. Our goal is to provide a framework for modeling combined spatial and molecular shifts as a response to time.
>
> Additionally, our qualitative analyses (Fig. 3, App. Figs. 11–14) show that NicheFlow generalizes to biologically annotated regions in the mouse embryo and axolotl brain, demonstrating its flexibility to **handle more sophisticated definitions of source clouds** based on labeling and prior biological knowledge.
>
> > Adapt $\lambda$
>
> $\lambda$ controls how source and target batches are paired during training, influencing the modeled dynamics.
>
> In principle, $\lambda$ can vary over time (e.g., higher in early development to capture large-scale migrations, lower later to emphasize local changes) or space (e.g., focusing long-range cell migration only in specific regions). However, spatial variation requires biological knowledge and annotation to identify areas where such dynamics are expected, making this approach less easily automatable.
>
> Learning $\lambda$ is technically feasible, but it requires additional supervision. Simply including it in the gradients of Flow Matching lacks biological interpretability. Biologically motivated proxies (e.g., stemness) could guide learning, but at the cost of added complexity. Instead, we treat $\lambda$ as a biologically informed prior that shapes learning in an interpretable way.
>
> > Sensitivity to $r$
>
> $r$ defines the spatial resolution at which NicheFlow learns tissue dynamics and **constraints its generated outputs to it**. According to our experiments, the choice of $r$ has an influence on the resulting microenvironments, which highlights the importance of selecting a biologically meaningful radius that matches the desired level of granularity (i.e., resolution of the anatomical regions to study). In practice, $r$ can be chosen to emphasize either coarse or fine structures, and it could even **vary across timepoints** to reflect biological contexts.
>
> > Alternative neighborhoods
>
> We refer the reviewer to our response above on "Microenvironments and cellular interactions", where we discuss the flexibility of NicheFlow in handling non-trivial neighborhoods. As noted there, one could define point clouds using alternative logics, such as pruning or weighting edges based on cell–cell interactions, rather than using a fixed spatial radius. While we opted for radius-based neighborhoods due to their interpretability and prevalence in spatial transcriptomics [4], our framework is not limited to this choice and could integrate graph-based or learned groupings in future work.
>
> > Continuous-time/Multi-timepoint
>
> NicheFlow enables multi-timepoint generation by sequentially predicting future states: we predict $t{+}1$ from $t$, then use the output as a source to generate $t{+}2$, and so on, without relying on ground truth between steps (see App. D.6). This is demonstrated in App. Figs. 11, 13–15 across various developmental processes in mouse (e.g., neural crest cell propagation) and axolotl (e.g., tracking hemisphere and cavity formation).
>
> A true continuous formulation would require aligning the model’s internal flow with biological time, for instance via a Neural ODE–style vector field. This extension is challenging due to point cloud density changes and the issue of incomparable source and target spaces, but it represents an interesting direction for future work. More crucially, such a formulation would require a more densely sampled temporal dataset to approximate a continuum. This is a technological limitation of nowadays' spatiotemporal data acquisition techniques, which focus on a coarse view of a dynamical process.
>
> > Limitations
>
> Our representation of attributed point clouds enables **flexible modeling of continuous cellular states** and can be combined with predefined or learned embeddings. In this work, we focused on datasets with a single sample per time point, where time is the dominant source of variation in both gene expression and spatial organization. We thus avoided scVI or similar approaches that remove sample-specific batch effects, as they risk discarding meaningful temporal signals. When replicates are available, VAE-based models can help remove technical variation, but in our setting, training a VAE would likely capture similar biological structure as PCA, albeit via a non-linear encoder.
>
> We also highlight the generative nature of NicheFlow: it uses Conditional Flow Matching to transform noise into spatially organized cell neighborhoods, conditioned on earlier time points. The model integrates spatial geometry via a point cloud transformer, capturing both expression and spatial context for generating spatial neighborhoods.
>
> Lastly, unlike pseudotime methods (e.g., DPT) that infer latent trajectories from expression data only, our approach models how spatial cell assemblies evolve over physical time. Our method does not attempt to infer latent cell orderings from expression data, but instead focuses on modeling how local tissue organization changes over time.
>
> While manifold learning is relevant, spatially constrained versions thereof are an interesting future direction, but beyond the scope of this work.
>
> **Ref**
>
> [1] Fan, Haoqiang, et al.. "A point set generation network for 3D object reconstruction from a single image." CVPR (2017).
>
> [2] Mémoli, Facundo. "Gromov–Wasserstein distances and the metric approach to object matching." Foundations of Computational Mathematics 11.4 (2011): 417–487.
>
> [3] Vayer, Titouan, et al. "Fused Gromov-Wasserstein Distance for Structured Objects." Algorithms (2020)
>
> [4] Haviv, Doron, et al. "The covariance environment defines cellular niches for spatial inference." Nature Biotechnology 43 (2025): 269–280.

---

> ### Comment · Reviewer_yXEk · 2025-08-03
>
> I thank the authors for their complete response. Their responses address my concerns, therefore I would like to increase my ratings from 5 to 6.

---

> > ### Author Response · Authors · 2025-08-03
> > **Thank you very much**
> >
> > We sincerely thank Reviewer yXEk for reading our rebuttal and providing thoughtful feedback, which helped us improve the paper. We truly appreciate their decision to increase the score and are encouraged by their recognition of our revisions.
> >
> > Best regards,
> >
> > The Authors

---

### Official Review · Reviewer_1m53 · 2025-07-06

**Clarity:** 2
**Significance:** 2
**Originality:** 3
**Rating:** 4
**Confidence:** 4

**Summary:**

This paper proposes a framework, NicheFlow, for predicting both the physical migration of cells within the microenvironment and the associated changes in gene expression. The framework combines variational flow matching with optimal transport-based conditioning, and is evaluated on three time-resolved spatial transcriptomics datasets to assess its representational capacity and fitting accuracy.

**Questions:**

* Appendix D.3 explores how changing $\lambda$ affects prediction and metrics, but the paper does not explain how to select $\lambda$ on data not seen during training. Without a principled way to choose $\lambda$, the method may have limited practical utility.
* In Fig. 10, NicheFlow appears to better fit niche migration than RPCFlow, but could this difference be due to hyperparameter choices or distribution assumptions (e.g., Gaussian vs. Laplace), rather than model architecture itself?
* In Fig. 3, NicheFlow seems to outperform Moscot, but this might simply be due to better parameter tuning. While both $\alpha$ (in MOSCOT) and $1-\lambda$ (in NicheFlow) control the spatial weighting, their formulations differ (e.g., GW loss vs. distance), so direct comparison at the same values is not necessarily fair.

Minor Comments
* What do the names RPCFlow and SPFlow mean? Please provide explicit definitions for these models.
* Table 1 is not referenced in the main text.

I would be open to raising my score if the authors are able to address the critical concerns raised above.

**Ethical Concerns:**

["NO or VERY MINOR ethics concerns only"]

**Final Justification:**

While I appreciate the new results provided in the rebuttal, I am not fully convinced by the claim that the value of $\lambda$ can be selected based on the biological question without a principled procedure. The updated evaluation does provide a clearer picture of the sensitivity to $\lambda$, but the absence of any holdout evaluation (e.g., a validation set or cross-validation) still limits the method’s generalizability and undermines model selection.
Overall, I will raise my score slightly.

**Limitations:**

yes

**Quality:**

3

**Strengths And Weaknesses:**

**Strengths**

* The problem of jointly estimating niche migration and gene expression changes is well-motivated and biologically relevant.
* The proposed model integrates recent advances, including the variational formulation of flow matching and OT-based alignment of unpaired source and target spaces, into a cutting-edge framework.
* The paper conducts component-wise validation of the model, including ablation studies, which strengthens the credibility of the empirical findings.
* The manuscript is well-structured, and the main idea is clearly explained.

**Weaknesses**
1. Ambiguous notation: The notation is somewhat unclear and potentially confusing. The paper refers to $x_0$ and $x_1$ as source and target variables, respectively, which implies that they live in different, unaligned domains. However, this creates confusion because $x_{t=0}$ and $x_0$ may refer to different variables.
2. Lack of hold-out evaluation: The most critical issue is the lack of explanation regarding how the hold-out test set was constructed. Throughout the paper, it appears that the model is evaluated directly on the source and target slides used for training, thereby assessing fitting performance rather than generalization. Since the proposed model is generative and intended to handle unseen states, it is essential to evaluate its performance on held-out samples or slides. Without such validation, the model’s generalizability remains unclear.
3. Model design: It is not evident that the current formulation of NicheFlow is effective in capturing heterogeneous niche dynamics. As described around eq. (13), the model appears to share the same flow functions $f_t$ and residual terms $r_t$ across all samples in $\mathcal{M}^1$, which could lead to excessive averaging and loss of niche-specific information. Although Table 1 shows some improvements of NicheFlow over RPCFlow, the performance likely depends on proper tuning $\lambda$, whose selection process is not discussed. This omission makes it difficult to judge whether the observed improvements are robust or incidental.
4.  Limited evaluation validity: The evaluation in Fig. 3 appears to measure how well the model fits the training data, not its predictive performance on unseen inputs. This makes it difficult to assess the true modeling capability. Moreover, the comparison of hyperparameters such as $\alpha$ (in MOSCOT) and $1 - \lambda$ may not be fair due to differences in underlying loss formulations. A comparison under optimal settings for each method is needed.

---

> ### Author Rebuttal · Authors · 2025-07-30
>
> We thank Reviewer 1m53 for the constructive and thorough feedback. We appreciate the positive remarks on our model's structure and core idea, as well as the critical comments. Below, we address the remaining concerns.
>
> > Notation
>
> As noted, $\mathbf{x}_0$ and $\mathbf{x}_1$ explicitly refer to source and target samples. When source and target spaces are incomparable, $\mathbf{x}_0$ serves as a conditioning input, not as the ODE’s initial state, and our notation reflects it.
>
> We intentionally avoid using $\mathbf{x}_t$ for the ODE solution, instead denoting it as $\phi_t^\theta(\cdot)$ throughout. Using $\mathbf{x}_t$ would suggest $\mathbf{x}_0 = \mathbf{x}\_{t=0}$, which is ambiguous when source and target domains are incomparable. By using $\phi_t^\theta$, we clarify that $\phi_0^\theta(\mathbf{z} \mid \mathbf{x}_0) = \mathbf{z}$, with $\mathbf{x}_0$ as a condition rather than the starting point of the trajectory. This is made explicit in L138 to L140. The same convention applies to interpolations, where we introduce $g_t$ (e.g., Eq. (2), L105).
>
> > Hold-out evaluation
>
> We agree that hold-out evaluation is generally desirable, and this could be implemented by leaving out either an anatomical region or an entire slide. However, both options are challenging due to **data constraints**.
>
> Predicting the trajectory of a region on a new slide requires prior observation of the same region in another sample at the same time point. Without this, the task becomes fully out-of-distribution, which remains difficult for current generative models [1]. Existing datasets lack replicated slides per time point, and to our knowledge, none provide repeated spatiotemporal measurements. In addition, current datasets span only coarse developmental stages and cover a few time points due to experimental costs. Holding out a slide would therefore introduce large temporal gaps, making it hard for the model to infer transitions.
>
> Instead, we showcase the model as follows:
> - ML perspective: We show that our approach improves over baselines at enforcing spatial constraints for spatiotemporal prediction (Fig. 2 and Tab. 1), evaluated on a grid of **test point clouds** (Sec. 5.1.1, App. F.5).
> - Biological perspective: We highlight its use for hypothesis generation, in contrast to exact OT methods, by analyzing spatial and transcriptional cell fate and the compositional evolution of selected anatomical regions (Sec. 5.2, App. D.6).
>
> Our goal is to introduce a scalable deep generative framework for niche-based spatiotemporal analysis, addressing the limitations of exact OT methods, which do not scale to large slides. With advances in dataset generation, we expect neural OT maps to play a key role in enabling generalization and modeling in spatiotemporal and perturbation contexts.
>
> > Flow function/residual
>
> In NicheFlow, $r_t$ and $f_t$ are neural networks predicting posterior means over denoised target coordinates and cellular features (L213–222). These means are used to infer the velocity field on noisy coordinates and cell states, conditioned on a source point cloud during generation (Sec. 3.3 and Sec. 4.3). Thus, $r_t$ is not a residual, but a posterior mean predictor that enables computing the generative velocity on the coordinates.
>
> $r_t$ and $f_t$ are outputs of a point cloud transformer that attends to **all nodes and positions** in both the source and noisy point clouds with cross-attention, and predicts a velocity for each sample. Samples from $\mathcal{M}$ are passed through the same global function, but this is **not** a shared MLP over individual points, but a **permutation-invariant** point cloud transformer. This architecture computes the velocities of samples and coordinates, integrating the intra-cloud context as well as the conditioning source point cloud (Sec. 4.4 and App. F.6 for details).
>
> > Justification for Tab. 1
>
> The metrics in Tab. 1 provide a technical evaluation of the **source-conditioned point cloud generation task**, supporting our architectural and modeling choices over baselines. Since we assess how well the global point cloud structure is recovered by a flow model and training strategy, this comparison was performed using a fixed $\lambda$ to enforce spatial preservation and the same **biological semantics** across models. For the rebuttal, we followed the suggestion and evaluated RPCFlow across varying $\lambda$ values (see below).
>
> > $\lambda$-dependent improvement
>
> To address the reviewer's concern, we conducted an extensive ablation study where we varied $\lambda \in \\{0.1, 0.25, 0.5, 0.75 \\}$ for both RPCFlow and NicheFlow across all three datasets and across all three training objectives. This resulted in 72 evaluations, each reported as the mean ± std. over 5 generation runs.
>
> Due to the limited space, we present only the best-performing method for each metric on each dataset. We will add the full table to the Appendix. When NicheFlow outperforms RPCFlow on all metrics, we report the overall best RPCFlow configuration.
>
> #### MED
>
> | Mod. | Obj. | $\lambda$ | 1NN-F1 | PSD $(10^2)$ | SPD   $(10^2)$ |
> |---|---|---|---|---|---|
> | NicheFlow | GLVFM | 0.1 | **0.663±0.002** | **0.897±0.021** | **0.824±0.013** |
> | RPCFlow | GLVFM | 0.25 | 0.593±0.001 | 0.926±0.003 | 1.137±0.002 |
>
> #### ABD
>
> | Mod. | Obj. | $\lambda$ | 1NN-F1 | PSD $(10^2)$ | SPD   $(10^2)$ |
> |---|---|---|---|---|---|
> | NicheFlow | GLVFM | 0.1 | **0.629±0.001** | 2.074±0.006 | **1.235±0.006** |
> | RPCFlow | GLVFM | 0.75 | 0.561±0.001 | **2.015±0.005** | 1.546±0.005 |
>
> #### MBA
>
> | Mod. | Obj. | $\lambda$ | 1NN-F1 | PSD $(10^2)$ | SPD   $(10^2)$ |
> |---|---|---|---|---|---|
> | NicheFlow | GLVFM | 0.1 | **0.285±0.001** | 1.556±0.001 | 0.694 ± 0.002 |
> | RPCFlow | CFM | 0.75 | 0.276±0.000 | **1.554±0.001** | 0.708±0.002 |
> | RPCFlow | GVFM | 0.5 | 0.245±0.000 | 1.755±0.001| **0.684±0.001** |
>
> We conclude that RPCFlow benefits from tuning the $\lambda$ parameter. However, even with its optimal configurations, it does not outperform NicheFlow with different patterns than when $\lambda=0.1$. Biologically, different values of $\lambda$ have distinct meanings (see below).
>
> > Biology of $\lambda$
>
> While Tab. 1 evaluates point cloud generation, the choice of $\lambda$ carries strong biological significance, which we explore throughout the paper. Specifically, $\lambda$ controls the trade-off between spatial coherence and feature similarity in the OT coupling and **should be selected based on the underlying question** (L318–323):
>
> 1. High $\lambda$: suited for modeling cell type migration across anatomical regions (Fig. 3, right).
> 2. Low $\lambda$: appropriate for studying how fixed structures evolve compositionally (Fig. 3, left).
>
> Since the optimal value depends on the biological context, we emphasize a qualitative analysis of $\lambda$’s impact, rather than relying solely on quantitative metrics. We show examples where a high or low value is necessary for meaningful biological interpretation.
>
> Notably, RPCFlow is excluded from our qualitative evaluation, as Sec. D.5 (Fig. 10) shows it fails to condition predictions on the source. Despite strong generative performance, it **ignores the source microenvironment** and yields biologically implausible mappings.
>
> > Comparison with moscot
>
> Moscot solves an **exact OT problem**, i.e. it computes transition probabilities between all source and target point pairs. Since its transition matrix is fixed, the model cannot generalize to test samples. We therefore compare our model to moscot on its native task of mapping embryonic structures across developmental time. Our results show that the proposed spatially constrained generative model offers a promising alternative to this established method. Additionally, App. D.6.2 presents tasks that exact OT models like moscot cannot handle, such as generating niche-specific anatomical changes and compositional remodeling.
>
> > $\alpha$ in moscot
>
> The comparison with moscot assesses how well each model captures compositional changes in anatomical structures or migratory patterns. Qualitatively, we found that the spatial component in the FGW problem is comparable to our spatial term across the 0.1–0.9 $\alpha$ range in the Fig. 3 analysis. When mapping fixed structures over time, values below 0.75 caused excessive density dispersion outside the anatomical region. For migration, no $\alpha$ value led to generally accurate transitions, though $\alpha = 0.5$ mapped the highest density to the expected cell type.
>
> Following 1m53's suggestion, we report the proportion of source density mapped to the **correct cell type** across $\alpha$ values, to support our choice. For the spinal cord (Fig. 3 left), values above 0.75 yielded better qualitative and quantitative results. For neural crest cells (Fig. 3 right), $\alpha = 0.5$ performs best.
>
> | Tissue | α=0.1 | α=0.25 | α=0.5 | α=0.75 | α=0.9 |
> |---|---|---|---|---|---|
> | Spinal Cord | 0.146 | 0.190 | 0.712 | **0.729** | 0.725 |
> | Neural Crests | 0.159 | 0.160 | **0.162** | 0.155 | 0.112 |
>
> > RPCFlow vs. NicheFlow
>
> The difference between RPCFlow and NicheFlow is that the former is trained on point clouds sampled across the slide, while the latter uses strictly defined spatial neighborhoods. Both methods are evaluated on the same test niches, and across the same model backbones, OT strategies and posterior losses (Gassian vs. Laplacian). Moreover, we showed above that NicheFlow overcomes RPCFlow for different values of $\lambda$, making our quantitative results robust to it.
>
> > RPCFlow and SPFlow?
>
> RPCFlow refers to a Random Point Cloud Flow model, where point clouds are sampled randomly across the tissue. SPFlow stands for Single Point Flow, which models trajectories at the single-cell level rather than on microenvironments. While described in Sec. 5.1.3, we will clarify them further in the revision.
>
> > Tab. 1 ref.
>
> We reference Tab. 1 in L299.
>
> [1] Nalisnick et al. Do deep generative models know what they don't know?, ICLR (2019).

---

> > ### Author Response · Authors · 2025-08-06
> >
> > Dear Reviewer 1m53,
> >
> > We would like to thank you again for reviewing our paper and providing feedback to improve its presentation and experimental completeness.
> >
> > In our rebuttal, we addressed the concerns expressed in the reviewing phase by:
> > * Clarifying ambiguous notation and the scope of our experimental validation and model selection.
> > * Providing additional experiments varying the parameter $\lambda$ to further support the value of our model.
> > * Elaborating on the comparison with moscot and providing evidence of the validity of our chosen parameter setting.
> > * Providing answers to the remaining technical questions.
> >
> > As we approach the conclusion of the discussion period, we would greatly appreciate hearing whether the reviewer feels their concerns have been addressed. We remain fully available to provide clarifications or answer any additional questions they may have.

---

> > > ### Comment · Reviewer_1m53 · 2025-08-09
> > > **Official Comment by Reviewer 1m53**
> > >
> > > Thank you for carefully addressing my concerns and for providing additional experiments on the effects of parameters $\lambda$ and $\alpha$.
> > > While I appreciate the new results, I am not fully convinced by the claim that the value of $\lambda$ can be selected based on the biological question without a principled procedure. The updated evaluation does provide a clearer picture of the sensitivity to $\lambda$, but the absence of any holdout evaluation (e.g., a validation set or cross-validation) still limits the method’s generalizability and undermines model selection.
> > >
> > > I also note that the paper cited in the rebuttal (Nalisnick et al. [1]) studies OOD detection (identifying OOD inputs), not predictive performance on OOD inputs, which is not directly relevant to the current discussion.
> > >
> > > Overall, I will raise my score slightly.

---

### Note · Authors · 2025-08-11

We would like to thank the reviewers and the AC again for handling our submission and for their contribution to improving our paper.

Hereby, we provide a summary of the discussion outcomes and the improvements made to our manuscript in response to the feedback received.

**1m53.**

- Clarified the mathematical notation and model components.
- Elaborated on the validation approach, noting limitations in existing spatial data (replicates, sampling depth).
- Added experiments over different ranges of the hyperparameter $\lambda$ for baseline flow models, explaining how its choice depends on the biological question.
- Justified the hyperparameter choice for moscot, providing additional experiments.

**yXEk.**

- Elaborated on the relevance of our neighborhood-based modeling choices in the field of spatial biology.
- Clarified misunderstandings about the choice of $\lambda$ and elaborated on multi-timepoint predictions and hyperparameter sensitivity.
- Presented additional favourable results on geometrically-informed metrics.

**fydQ.**

- Discussed missing related work, highlighted our contribution and contrasted our model with standard static and dynamic OT approaches.
- Presented additional results on Wasserstein distance metrics in the coordinate and state spaces.
- Clarified the position of our modeling choices in the context of unbalanced OT and interpolation tasks.

**yBc1.**

- Motivated our paper's structure and proposed textual revisions to better position our contribution biologically for a general audience.
- Highlighted the biological contribution and evidence presented in our paper.
- Provided additional clarifications on baselines, relevance of our modeling choices and data characteristics.

As a result of our rebuttal responses, all reviewers increased their score by 1. We commit to include all additional experimental results, literature discussions and biological clarifications in a revised version of the manuscript.

Thank you again for your time and consideration.

---

### Decision · Program_Chairs · 2025-09-17

**Decision:**

Accept (poster)

**Comment:**

The reviewers and I have a near-unanimous positive assessment of the paper: modeling the spatiotemporal evolution of ST data is an important setting and the approach performs favorably in comparison to MOSCOT and other FM approaches. It's a relevant ML problem (arising in computational biology) with appropriate baselines. The original paper lacked some quantitative results (and held-out performance), but the results in the rebuttal address this and show favorable results.